# Neuraminidase 1 promotes renal fibrosis development in male mice

Qian-Qian Chen[1,2,7], Kang Liu[3,7], Ning Shi[4,7], Gaoxiang Ma ®[5], Peipei Wang[6], Hua-Mei Xie[5], Si-Jia Jin[4], Ting-Ting Wei[1], Xiang-Yu Yu[1], Yi Wang[5], Jun-Yuan Zhang[1], Ping Li[1], Lian-Wen Qi ®[1,4,5] ✉ & Lei Zhang[1,4] ✉

The functions of the influenza virus neuraminidase has been well documented but those of the mammalian neuraminidases remain less explored. Here, we characterize the role of neuraminidase 1 (NEU1) in unilateral ureteral obstruction (UUO) and folic acid (FA)-induced renal fibrosis mouse models. We find that NEU1 is significantly upregulated in the fibrotic kidneys of patients and mice. Functionally, tubular epithelial cell-specific NEU1 knockout inhibits epithelial-to-mesenchymal transition, inflammatory cytokines production, and collagen deposition in mice. Conversely, NEU1 overexpression exacerbates progressive renal fibrosis. Mechanistically, NEU1 interacts with TGFβ type I receptor ALK5 at the 160-200aa region and stabilizes ALK5 leading to SMAD2/3 activation. Salvianolic acid B, a component of *Salvia miltiorrhiza*, is found to strongly bind to NEU1 and effectively protect mice from renal fibrosis in a NEU1-dependent manner. Collectively, this study characterizes a promotor role for NEU1 in renal fibrosis and suggests a potential avenue of targeting NEU1 to treat kidney diseases.

Renal fibrosis is an inevitable consequence of almost all forms of progressive chronic kidney disease (CKD)[1]. Despite its high prevalence and severe morbidity and mortality, there is currently limited effective therapy that can halt or reverse renal fibrogenesis and the progression of CKD[2]. Comprehensive characterization of the complex mechanisms and signaling mediators that underpin renal fibrosis could propose new therapeutic avenues.

Kidney fibrosis is a multifactorial and progressive disease, involving diverse cell types of renal and extrarenal origin. Four major cell types are involved in CKD progression: tubular epithelial cells (TEC), myofibroblasts, endothelial cells, and immune cells[3,4]. The different types of cells play intricate, distinct, and interrelated roles. TEC constitute a major component of the kidney that respond to injuries. Previous studies have highlighted resident TEC as initiators of human kidney fibrosis rather than victims of same[5]. Injured TEC undergo

partial epithelial-to-mesenchymal transition (EMT) but still reside within the basement membrane of the tubules. They are characterized by the acquisition of mesenchymal features, and co-expression of epithelial and mesenchymal cell markers. As a consequence, damaged TEC release paracrine signals such as pro-inflammatory and pro-fibrotic factors into the renal interstitium, thus reshaping the micro-environment to promote both inflammation and fibrogenesis[6]. A diverse array of molecular determinants that reside in the TEC have been characterized to play critical roles in renal fibrosis and CKD. Some act as promotors of TEC injury and tubulointerstitial fibrosis, for instance, Snail, Twist, Wnt9a, Notch1, Hif1α, and Kim1[7–12]. Others such as Mst1/2 and Atg5 play a suppressor role[13,14].

Neuraminidases or sialidases are a family of glycoside hydrolase enzymes that catalyze the removal of sialic acid from viral and cellular glycoconjugates[15]. The most widely studied neuraminidase is on the

[1]State Key Laboratory of Natural Medicines, China Pharmaceutical University, Nanjing 210009, China. [2]School of Pharmacy, Nanjing University of Chinese Medicine, Nanjing 210023, China. [3]Department of Nephrology, Jiangsu Province Hospital, The First Affiliated Hospital of Nanjing Medical University, Nanjing 210029, China. [4]School of Traditional Chinese Pharmacy, China Pharmaceutical University, Nanjing 211198, China. [5]Clinical Metabolomics Center, China Pharmaceutical University, Nanjing 211198, China. [6]College of Food Science and Technology, Shanghai Ocean University, Shanghai 201306, China. [7]These authors contributed equally: Qian-Qian Chen, Kang Liu, Ning Shi. ✉e-mail: Qilw@cpu.edu.cn; Zhanglei@cpu.edu.cn

surface of the influenza virus, where the enzyme removes sialic acid from host receptors to facilitate viral release. In mammals, neuraminidases have been shown to be involved in diverse physiological and pathological processes. First, the mammalian neuraminidase family consists of four members (NEU1-NEU4), responsible for the initial step of degradation of glycoconjugates by removing sialic acids[16]. Among them, NEU1 and NEU4 are typically located in lysosomes. NEU1 deficiency leads to sialidosis, a disease characterized by tissue accumulation of sialo-glycopeptides and sialo-oligosaccharides[17]. Second, neuraminidases participate in post-translational modifications via desialylation, and modulate the structure and function of glycoproteins[18,19]. Third, neuraminidases can affect protein functions via protein–protein interaction. Our previous work showed that neuraminidases bind to transcriptional factor GATA4 to drive cardiac hypertrophy[20]. Fourth, we also reported that neuraminidases activation resulted in accumulation of circulating *N*-acetyl-neuraminic acid, a signaling metabolite that can trigger RhoA and Cdc42-dependent myocardial injury[21]. On account of their various functions, mammalian neuraminidases have been shown to play an emerging role in several human diseases, including autoimmune[22,23], cardiovascular diseases[20,24,25] cancers[26], neurodegenerative disorders[27], and lung diseases[28,29]. Design of the specific inhibitors targeting neuraminidases have the potential to treat these diseases[30,31]. The role of neuraminidases in CKD and TGFβ signaling, however, remain largely unexplored.

To address the existing scientific gray areas and loopholes in respect of the neuraminidases, this work aimed to study their roles in renal fibrosis. We detected the neuraminidase 1 (NEU1) expression in patients with renal fibrosis, and in mice subjected to unilateral ureteral obstruction (UUO) or administered folic acid. TEC-specific NEU1 knockout and overexpression mice were generated to characterize the role of NEU1 in renal fibrosis progression. PCR array, co-immunoprecipitation, and surface plasmon resonance (SPR) were employed to investigate the underlying mechanisms by which NEU1 promotes renal fibrosis. In addition, natural compounds were screened to bind to mammalian NEU1 and protect kidneys from injury in mice.

## Results

### NEU1 was elevated in TEC of CKD patients

We first examined expressions of the four neuraminidase members (NEU1-NEU4) by analyzing renal transcriptomics database Nephroseq (https://www.nephroseq.org/resource/login.html). In the 'Ju CKD TubInt' dataset, the mRNA of NEU1 but not NEU2-NEU4 was significantly upregulated in kidney biopsy tissues of the CKD patients ($n = 123$) compared with controls ($n = 31$) (Fig. 1a). Specifically, NEU1 was elevated in most types of CKD including IgA ($n = 25$), diabetic kidney disease ($n = 17$), lupus nephritis ($n = 32$), focal segmental glomerulosclerosis ($n = 17$), and membranous glomerulonephritis ($n = 18$), but not in minimal change disease ($n = 14$) (Supplementary Fig. 1a). The NEU1 was also highly expressed in 'Ju CKD Glom' dataset ($n = 149$ for CKD and $n = 21$ for healthy control) and GSE66494 ($n = 53$ for CKD and $n = 8$ for healthy control) (Fig. 1b, c).

Next, we analyzed 16 microdissected human kidney samples collected from patients with renal fibrosis ($n = 8$) and without renal fibrosis (no specific pathologic alterations by biopsy procedures, $n = 8$). Clinical demographics of these subjects are provided in Supplementary table 1. Patients with renal fibrosis showed severe collagen deposition and kidney injury as captured by Masson and hematoxylin-eosin (HE) staining (Fig. 1d, f, g). Immunohistochemistry (Fig. 1d, e) and immunofluorescence (Supplementary Fig. 1b, c) revealed that NEU1 protein levels were significantly higher in patients with renal fibrosis than without renal fibrosis. In addition, the levels of NEU1 showed a strong positive correlation with the score of tubular injury (Fig. 1h), the level of serum creatinine (Fig. 1i), and the blood urea nitrogen (Fig. 1j), but showed a negative correlation with the estimated glomerular filtration rate (Fig. 1k). Immunofluorescence (Fig. 1l–n) showed that the increase of NEU1 content was mainly co-localized with kidney injury molecule 1 (KIM1), a biomarker of injured epithelial cells in acute kidney injury and CKD[32,33]. Analysis of Nephroseq database also showed that NEU1 expression was negatively correlated with glomerular filtration rate (Supplementary Fig. 1d) and was mainly expressed in the renal cortex part of the kidney (Supplementary Fig. 1e).

### NEU1 was upregulated in fibrotic kidneys of mice

Next, we determined the protein levels of NEU1 in mice. Renal fibrosis model was established by unilateral ureteral obstruction (UUO) or folic acid stimulation. NEU1 mRNA (Fig. 2a) and protein (Fig. 2b–d) were significantly increased in the kidneys of the fibrotic mice in response to UUO. These elevations were time-dependent manner (Fig. 2a, b). Staining of the kidney sections showed increased localization of NEU1 in the TEC, but not in the macrophages (Fig. 2e, f). NEU1 expression was also increased in the kidneys of folic acid-induced renal fibrosis (Fig. 2b–d, g). Next, we analyzed the fibrosis indices and then correlated them with NEU1 expression. The results showed that the mRNA expressions of *Acta2*, *Vim*, *Col1a1*, *Col3a1*, and *Tgfβ* greatly increased in fibrotic kidney compared with control group (Fig. 2h, i). Of note, the level of NEU1 showed a positive correlation with these fibrosis indices (Fig. 2j–s).

### NEU1 mediated TGFβ-induced changes in HK-2 cell

Human tubular epithelial HK-2 cells were stimulated by TGFβ, a commonly used stimulator to induce fibrosis, to study the role of NEU1 in TEC injury. In response to TGFβ stimulation, NEU1 expression was increased at the mRNA (Supplementary Fig. 2a) and protein levels (Supplementary Fig. 2b). E-Cadherin (*CDH1*), a marker of epithelium cells, was upregulated in NEU1-knockdown (Supplementary Fig. 2c, d) HK-2 cells in the presence of TGFβ. The mesenchymal cell markers stimulated by TGFβ, such as Vimentin (*VIM*), Snail (*SNAI1*), and Slug (*SNAI2*) were significantly downregulated when NEU1 was knocked down (Supplementary Fig. 2e, f). Conversely, NEU1 overexpression (Supplementary Fig. 2g, h) promoted TGFβ-induced Fibronectin 1, KIM1, and EMT-associated genes transcription and protein expression (Supplementary Fig. 2i, j). Overexpressed NEU1 did not form aggregation when the cells transfected with NEU1 full-length plasmid (Supplementary Fig. 2k).

### TEC-specific NEU1 deletion inhibited renal fibrosis in mice

To investigate the role of NEU1 in kidney fibrosis, we generated mice with TEC-specific deleted *Neu1* using a Cre/loxP-dependent conditional gene-targeted approach. Mice homozygous for the *Neu1*-loxP (fl)-targeted allele (*Neu1*fl/fl) were crossed with TEC-specific Ksp1.3 Cre lines (Supplementary Fig. 3a). The mRNA level (Supplementary Fig. 3b), protein expression (Supplementary Fig. 3c, d), and enzyme activity (Supplementary Fig. 3e) confirmed the reduction of NEU1 expression in *Neu1*fl/fl mice (*Neu1*CKO) compared with *Neu1*fl/fl littermates without Cre recombinase activity (control mice). Electron microscopy showed that there were no significant morphological differences between *Neu1*CKO mice and the control mice in the podocytes, glomerular basement membrane, the blood vessel, and tubular morphology (Supplementary Fig. 3f). A minor change is that a few vacuoles (examples indicated by asterisks) were detected in the cytosol of tubular cells in *Neu1*CKO mice but not in the control mice (Supplementary Fig. 3f). Next, we performed immunohistochemical assay for LAMP1, an indicator of lysosomal storage and exocytosis[17,29]. LAMP1 expression was slightly increased but not significantly altered ($p = 0.39$) in the kidney tissues of *Neu1*CKO mice compared with control groups (Supplementary Fig. 3g).

In response to UUO (Fig. 3a), NEU1 knockout significantly improved morphology (Fig. 3b, c), reduced collagen deposition (Fig. 3b, d), inhibited tubular necrosis and tubulointerstitial

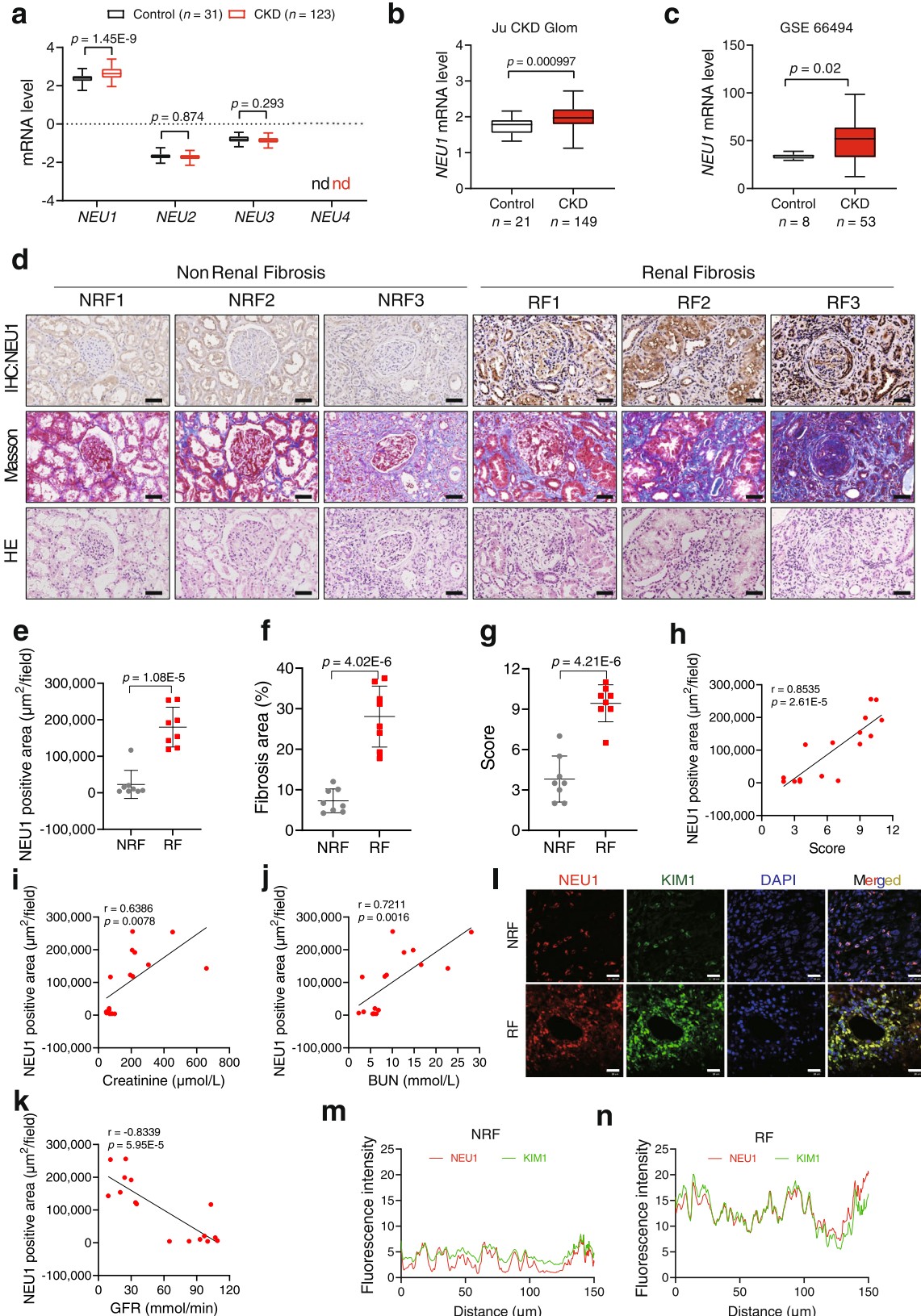

inflammation (Fig. 3b, e, f), suppressed macrophage infiltration (Fig. 3b, g) and nuclear phosphorylated NF-κB (pNF-κB) in tubular cells (Fig. 3b, h). NEU1 deficiency markedly inhibited UUO-induced KIM1 expression (Fig. 3i). Immunofluorescence showed that partial EMT was induced by UUO, as evidenced by the remnants of TEC in the basement membrane and co-expression of epithelial cell marker E-Cadherin and mesenchymal cell marker Vimentin (Fig. 3j, k). This EMT progression was inhibited by NEU1 knockdown (Fig. 3j, k). In line with this, the mRNA and protein of E-Cadherin were dramatically restored (Fig. 3l and Supplementary Fig. 4a, b), while Snail, Slug, α-SMA, and Vimentin were significantly inhibited in the kidneys of *Neu1*CKO mice (Fig. 3l and Supplementary Fig. 4a, b).

**Fig. 1 | NEU1 is significantly upregulated in kidneys from patients with CKD.**
**a** Relative NEU1, NEU2, NEU3, and NEU4 mRNA levels. Data analysis from Nephro-seq database ('Ju CKD Tublnt' dataset, median-centered log2). $n = 31$ samples in control group, $n = 123$ samples in CKD group. Unpaired $t$-test. ns, no significant difference. nd, not detected. **b** The expression of NEU1 (median-centered log2) in kidney specimens from control ($n = 21$ samples) and patients with chronic kidney disease (CKD, $n = 149$ samples). Unpaired $t$-test. Data analysis from Nephroseq database ('Ju CKD Glom' dataset). **c** The mRNA levels of NEU1 in kidney specimens of CKD ($n = 53$ samples) and control ($n = 8$ samples) in GSE66494 dataset (Probe ID: A_24_P394533). Unpaired $t$-test. **d** Tissue adjacent sections of kidney from patients with non-renal fibrosis or renal fibrosis by immunohistochemistry, Masson staining, and HE staining. Scale bar = 20 μm. NRF non-renal fibrosis, RF Renal fibrosis. $n = 8$ samples per group. **e**–**g** Quantification of NEU1 expression (**e**), fibrotic area (**f**),

and score of kidney damage (**g**) based on immunohistochemistry, Masson, or HE staining in (**d**). Data were presented as mean ± SD. $n = 8$ samples per group. Unpaired two-tailed $t$-test. IOD: integrated optical density. **h** The correlation of NEU1 expression and degree of tubular degeneration ($n = 16$, Pearson $\chi^2$ test). **i**–**k** Pearson's correlation of NEU1 with serum creatinine level (**i**), blood urea nitrogen (BUN) (**j**), and glomerular filtration rate (GFR) (**k**) ($n = 16$, Pearson $\chi^2$ test). **l**–**n** Representative images of co-immunofluorescence staining of NEU1 and KIM1 in kidney tissues of non-renal fibrosis and renal fibrotic patients (**l**). Fluorescence intensity of NEU1 and KIM1 in diagram k-up (**m**) and k-down (**n**), Image J software was used for statistics. Scale bars = 20 μm. $n = 3$ samples per group. **a**–**c** Data are presented as box-and-whisker plots, solid line inside box indicates the median, the bottom and top of box represent first and third quartiles, and the bottom and top whisker show the minimum and maximum, respectively. All tests were two-tailed.

PCR array involving 84 fibrosis-related mRNAs showed that the TGFβ family, MMP members, collagens, inflammatory cytokines, chemokines, and adhesion molecules were significantly stimulated in the kidneys of mice in response to UUO, and were evidently inhibited by NEU1 knockout (Supplementary Fig. 4c–e). Further RT-qPCR confirmed that NEU1 knockout dramatically inhibited UUO-induced gene expressions of inflammatory cytokines (*Il1β*, *Il6*, and *Il18*), chemokines (*Ccl1*, *Ccl2*, *Ccl3*, *Ccl4*, *Ccl5*, *Cxcr10*, and *Cxcr12*), tumor necrosis factor (*Tnfα*), interferons (*Ifnα*, *Ifnβ*, and *Ifnγ*), adhesion molecules (*Icam* and *Vcam*) (Supplementary Fig. 4f, g) and fibrogenic factors (*Col1a1*, *Col3a1*, *Fn1*, *Tgfb1*, and *Tnc*) (Fig. 3m).

In another renal fibrosis model induced by folic acid (Fig. 4a), *Neu1*CKO mice also showed significantly improved morphology and renal weight (Fig. 4b, c). Renal function-associated creatinine and blood urea nitrogen were considerably improved in *Neu1*CKO mice compared with littermate control (Fig. 4d, e). NEU1 knockdown markedly inhibited folic acid-induced *KIM1* expression (Fig. 4f). HE and Masson staining indicated that the kidneys of the *Neu1*CKO mice developed relatively less tubular injury and interstitial fibrosis (Fig. 4g–i). Macrophage infiltration (Fig. 4g, j), EMT marker gene expression (Fig. 4l and Supplementary Fig. 5a, b), pro-inflammatory cytokines (Fig. 4g, k and Supplementary Fig. 5c), chemokines (Supplementary Fig. 5d), and fibrogenic factors (Fig. 4m) were noticeably reversed due to NEU1 deficiency in the renal tubular epithelial cells.

## NEU1 overexpression augmented UUO-induced renal fibrosis in mice

Besides loss-of-function, a gain-of-function approach was performed using adeno-associated virus serotype 9 encoding NEU1 (AAV9-NEU1) and AAV9-Ctrl (Fig. 5a). The virus was injected in situ in the cortex of the kidney in mice (Fig. 5a). The mRNA, protein expression, and enzyme activity of NEU1 were significantly increased in the kidneys after injection of AAV9-NEU1 for 5 weeks (Fig. 5b, c and Supplementary Fig. 6a, b). Lysosomal protective protein/cathepsin A (PPCA), a chaperone of NEU1 indispensable for catalytic activation, was also increased at 55 kDa in kidney tissues of NEU1-overexpression mice (Supplementary Fig. 6c). We observed a slight elevation of the mature PPCA subunit at 32 kDa and no significant difference at 20 kDa between the two groups (Supplementary Fig. 6c). The immunofluorescence results showed that AAV9-NEU1 was successfully transduced into the TEC (Supplementary Fig. 6d). As expected, NEU1 overexpression exacerbated the UUO-induced renal shriveled (Fig. 5d, e), tubular expansion (Fig. 5d, f), and collagen deposition both in the cortex and medulla of the kidneys (Fig. 5d, g). RT-qPCR, western blots, and immunohistochemistry indicated that NEU1 overexpression augmented UUO-induced KIM1 expression (Fig. 5h) and EMT progression in the kidneys (Fig. 5i–n and Supplementary Fig. 6e, f). UUO-stimulated macrophage infiltration (Supplementary Fig. 6g), chemokines, pro-inflammatory cytokines, interferons, adhesion molecules, and fibrogenic factors expressions were also aggravated by NEU1 overexpression (Fig. 5o and Supplementary Fig. 6h, i).

## NEU1 interacted with ALK5 at the 160−200 region

In an effort to decode the underlying mechanism by which NEU1 promoted renal fibrosis, Kyoto Encyclopedia of Genes and Genomes (KEGG) pathway analysis was performed by mapping differentially expressed genes of PCR array into the website (http://www.genome.jp/kegg) to identify hub signaling pathway. The correlation analysis identified top 20 enriched KEGG pathways that are potential to be involved in NEU1-mediated renal fibrosis (Supplementary Fig. 7a). The top three enriched KEGG pathways are AGE-RAGE signaling pathway in diabetic complications, TGF-β signaling pathways, and proteoglycans in cancers (Supplementary Fig. 7a). Since TGFβ is the predominant pathogenic factor that drives EMT and fibrosis in CKD[34], we then centered on the TGF-β signaling pathway.

Upon TGFβ stimulation, the TGF-β type I (activin receptor-like kinases 1 to 7, ALK1-7) and type II receptors (AMHR2, TGFBR2, and ACVR2B) assemble a heterotetrameric receptor complex at the initial stage[35]. Subsequently, we tested the possibility of direct interaction of NEU1 with TGF-β receptors. Co-immunoprecipitation (Co-IP) assay showed that NEU1 selectively bound to ALK5 with a strong affinity, but could not bind to ALK2, ALK3, ALK6, ALK7, AMHR2, and TGFBR2 among the TGFβ receptor family (Fig. 6a, b). It appeared that NEU1 could also moderately interact with ALK1, ALK4, and ACVR2B. The interaction of NEU1 with ALK5 was confirmed using Co-IP combined with mass spectrometry (Supplementary Fig. 7b and Supplementary Data 1). Immunofluorescence showed that NEU1 was co-localized with ALK5 in human fibrotic kidneys (Fig. 6c), indicating their potential interaction. The interaction between NEU1 and ALK5 was enhanced in response to TGFβ stimulation in HK-2 cells (Fig. 6d, e).

To explore the localization of these interactions, we used bimolecular fluorescence complementation (BiFC) to detect the direct interaction between NEU1 and ALK5 in HK-2 cells. If binding happens, the VC155 fragment fused to NEU1 and the VN173 fused to ALK5 are brought into proximity and produce a fluorescent signal (Fig. 6f). Of note, fluorescent signals were much higher in TGFβ group than in control group, suggesting a strong binding of NEU1 with ALK5 upon TGFβ stimulation (Fig. 6g, h). We then used lyso-tracker, mito-traker, and CM-tracker to observe the localization of the interaction between NEU1 and ALK5. The results showed that fluorescent signals of NEU1 and ALK5 interaction were co-localized with lysosomes, mitochondrion, but less co-localized with cell plasma membrane (Supplementary Fig. 7c). Another technology, in situ proximity ligation assay (PLA), was also performed to validate the interaction of NEU1 and ALK5 in fibrotic kidney of patient. PLA used DNA hybridization and DNA amplification steps with fluorescent probes to visualize interacting proteins (Fig. 6i). In fibrotic kidney of patient, PLA results showed strong fluorescent signals in TECs (Fig. 6j), suggesting a direct interaction between NEU1 and ALK5.

As a further characterization of NEU1-ALK5 interaction, we determined the specific binding domain of ALK5. ALK5 consists of an extracellular domain (amino acids, 1−125), a transmembrane domain

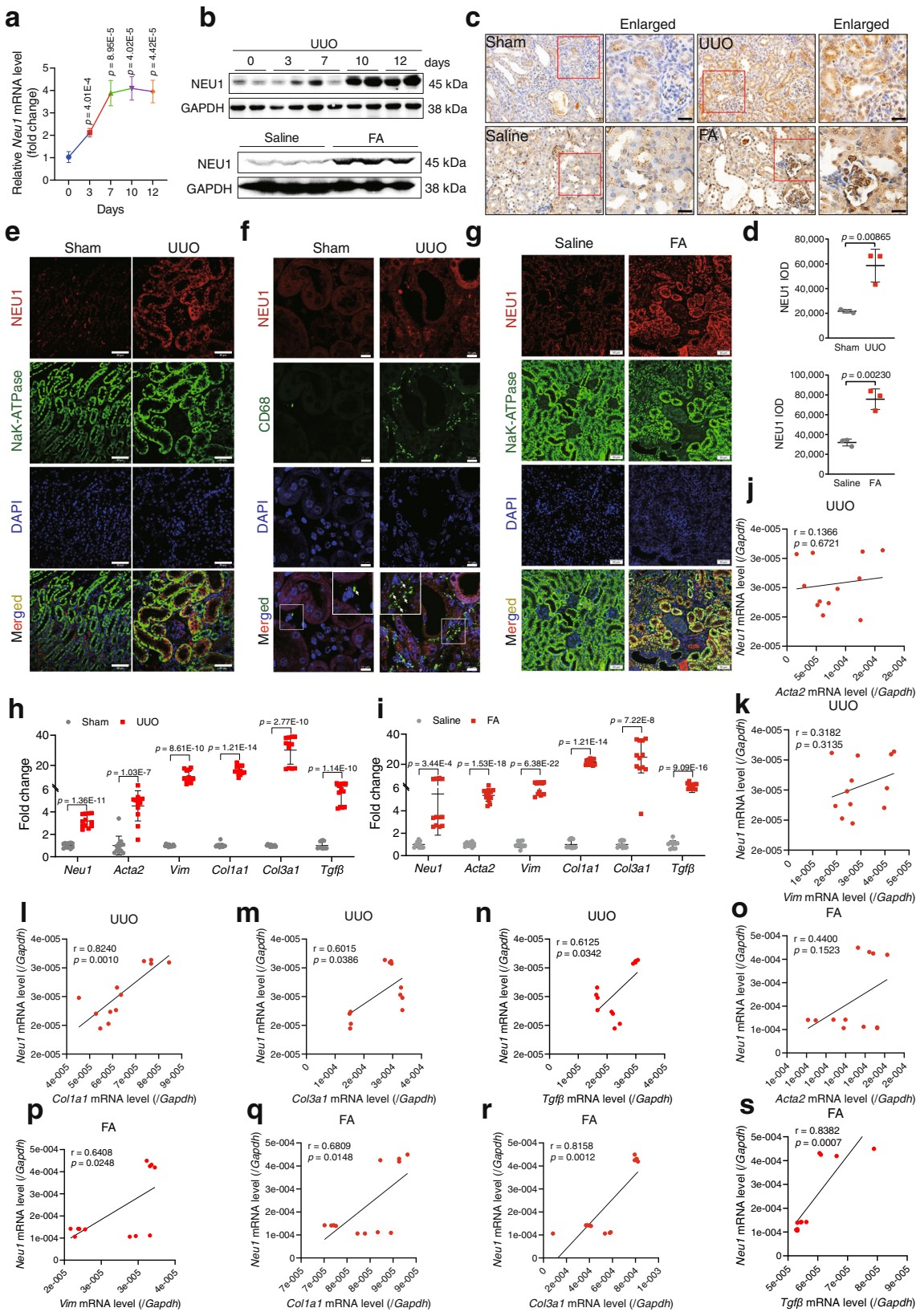

(amino acids, 126–174), a GS domain (amino acids, 175–204), and Ser/Thr kinase domain (amino acids, 205–495) (Supplementary Fig. 7d)[36]. Surface plasmon resonance showed that NEU1 displayed a favorable binding affinity (Fig. 6k) to the amino acid 162–403 region of ALK5 with an estimated equilibrium dissociation constant ($K_D$) of 3.14 nM (Fig. 6k), but could not bind to 1–125 or 200–503 region using

commercially available recombinant proteins (Supplementary Fig. 7e, f). It was thus reasonable to predict that the 160–200 amino acid region of ALK5 is the NEU1-binding domain (Supplementary Fig. 7d). Furthermore, we constructed a Flag-Tag plasmid containing 160–200 amino acid sequences, and observed that NEU1 interacted with this domain of ALK5 by Co-IP (Fig. 6l). PLA (Fig. 6m) and Co-IP (Fig. 6n)

**Fig. 2 | NEU1 is elevated in mouse fibrotic kidneys. a** NEU1 mRNA levels in kidneys of mice subjected to unilateral ureteral ligation (UUO) for 0, 3, 7, 10, and 12 days. $n = 4$ mice per group. Unpaired two-tailed $t$-test. Data were presented as mean ± SD. **b** Western blots of NEU1 levels in kidney from mice subjected to UUO for 0, 3, 7, 10, and 12 days or administered folic acid (FA, 250 mg/kg, *i.p.*) injection for 4 weeks. $n = 3$ mice per group. **c, d** Immunohistochemistry of kidney sections (**c**) and quantitative results (**d**) from mice after UUO surgery for 10 days or folic acid injection for 4 weeks. The NEU1 was determined with IOD by Image-Pro Plus 6.0. $n = 3$ mice per group. Scale bar, 20 μm. Data were presented as mean ± SD. Unpaired two-tailed $t$-test. IOD, integrated optical density. **e**–**g** Immunofluorescence images of NEU1 in kidney from mice subjected to UUO (**e, f**) or folic acid (**g**). Na⁺/K⁺-ATPase was used as tubular epithelial cell marker, CD68 was used as macrophage marker. $n = 3$ mice per group. Scale bar, 50 μm. **h, i** *Neu1*, *Acta2*, *Vim*, *Col1a1*, *Col3a1*, and *Tgfβ* mRNA level in kidney from mice after UUO surgery for 10 days (**h**) or folic acid injection for 4 weeks (**i**). $n = 6$ mice per group. Data were presented as mean ± SD. Unpaired two-tailed $t$-test. **j**–**s** Pearson's correlation of NEU1 with *Acta2*, *Vim*, *Col1a1*, *Col3a1*, and *Tgfβ* mRNA. $n = 12$, two-tailed Pearson $\chi^2$ test.

assays confirmed that 160–200 domain plasmid competed with ALK5 and inhibited NEU1-ALK5 binding.

## NEU1 interacted with and stabilized ALK5 to enhance ALK5-SMAD2/3 signaling pathway

To investigate the effects of NEU1-ALK5 interaction on ALK5, we measured the stability of ALK5 in the presence or absence of NEU1. NEU1 knockdown promoted ALK5 degradation, whereas NEU1 overexpression inhibited ALK5 degradation (Fig. 7a, b), suggesting that NEU1 interacted with and stabilized ALK5. ALK5 is able to phosphorylate its substrates, the SMAD family[37]. NEU1 silence markedly suppressed TGFβ-induced SMAD2/3 activation in a time-dependent manner, while NEU1 overexpression sustained SMAD2/3 continuous activation in the presence of TGFβ. (Supplementary Fig. 8a–d). In HK-2 cells and the mouse kidney, the phosphorylation of ALK5 and the downstream SMAD2/3 were also significantly inhibited upon NEU1 silencing (Fig. 7c, d and Supplementary Fig. 8e, f). On the contrary, NEU1 overexpression augmented TGFβ- or UUO-induced activation of ALK5 and SMAD2/3 (Fig. 7e, f and Supplementary Fig. 8g, h). Inhibiting the binding of NEU1 and ALK5 by transfecting the cells with ALK5₁₆₀₋₂₀₀ plasmid significantly suppressed SMAD2/3 activation (Fig. 7g, h).

We added phosphatase to cell lysis buffer to remove phosphate groups of proteins. The Co-IP showed that phosphatase significantly inhibited the interaction between NEU1 and ALK5 (Fig. 7i), suggesting their interaction depends on the phosphorylation state of ALK5. When ALK5 was knocked down (Fig. 7j), NEU1 overexpression-induced KIM1 (Fig. 7k) and ALK5-SMAD2/3 signaling pathway activation (Fig. 7l, m) were abolished in HK-2 cells, indicating that NEU1 promoted renal fibrosis in an ALK5-dependent manner. Collectively, these results suggested that NEU1 interacted with the GS domain (amino acids 160–200) of ALK5 in cytoplasm, and then enhanced the ALK5-SMAD2/3 signaling pathway, contributing to renal fibrosis (Fig. 7n).

To explore whether NEU1 acts in the cells as an enzyme that removes terminal sialic acid residues of ALK5, we employed biotin-labeled sambucus nigra lectin (SNL) to specifically bind to sialic acid via α-2-6 linkage. Results showed that the SNL level of ALK5 was decreased in response to TGFβ stimulation, and this decrease was aggravated in NEU1-overexpression cells (Supplementary Fig. 9a). Mutation of enzyme active sites (mtNEU1: D103A, Y370A, E394A) reduced the effects of NEU1 overexpression on TGFβ-induced ALK5-SMAD2/3 activation (Supplementary Fig. 9b–e).

## Targeting NEU1 by salvianolic acid B protected kidney

To identify candidate compounds targeting mammalian NEU1, SPR was employed to screen the binding affinities of 74 natural products from medicinal plants with recombinant human NEU1 protein (Fig. 8a, Supplementary Data 2 and Supplementary Table 4). The top two compounds with the strongest NEU-binding affinities are salvianolic acid B from *Salvia miltiorrhiza* with $K_D$ at 21.57 nM (Fig. 8b) and rosmarinic acid from the *Rosmarinus officinalis* genus with $K_D$ at 141 nM (Supplementary Data 2 and Supplementary Table 4). Interestingly, some high-molecule polysaccharides showed the potential to bind to NEU1 such as Glehniae radix polysaccharide IV with $K_D$ at 240 nM and

Panax quinquefolium polysaccharides V with $K_D$ at 6.39 μM (Supplementary Data 2 and Supplementary Table 4). Co-IP (Supplementary Fig. 10a) and PLA experiments (Supplementary Fig. 10b) showed that salvianolic acid B significantly inhibited the interaction between NEU1 and ALK5. TGFβ-induced ALK5-SMAD2/3 signaling pathway activation was also blocked by salvianolic acid B (Supplementary Fig. 10c, d).

Subsequently, we investigated the protective effects of salvianolic acid B from renal injury in mouse models (Fig. 8d), and compared its effects with salvianolic acid A, a compound also from *Salvia miltiorrhiza* but showing relatively weak NEU1-binding affinity with $K_D$ at 52.30 μM (Fig. 8c). HE and Masson staining data demonstrated that salvianolic acid B (40 mg/kg, tail vein injection) significantly attenuated UUO-induced renal injury and renal fibrosis (Fig. 8e, f). Salvianolic acid B treatment markedly inhibited *Kim1* (Fig. 8g), *Snai1* (Fig. 8h), and *Snai2* expression (Fig. 8i). Salvianolic acid B also suppressed proinflammatory cytokine production (*Tnf-α*, *Il-6*, and *Il1β*) and collage deposition (Fig. 8j–o). In line with these, the phosphorylation of ALK5 and the down-stream phosphorylation of SMAD2/3 were inhibited by salvianolic acid B (Fig. 8p). In comparison with salvianolic acid B, salvianolic acid A at the same concentration (40 mg/kg) showed comparable effects on some markers such as *Kim1* (Fig. 8g), *Col1a1* (Fig. 8n), and *Col3a1* (Fig. 8o) expression, but relatively weaker effects on collage deposition (Fig. 8f), *Snai2* (Fig. 8i), and *Tnfα* (Fig. 8j) in UUO-induced mouse model. However, salvianolic acid A failed to suppress UUO-induced *Snai1* (Fig. 8h), *Il6* (Fig. 8k), *Il1β* (Fig. 8l), *Vim* (Fig. 8m), and phosphorylations of ALK5 and SMAD2/3 (Fig. 8p). The protective effects of salvianolic acid B from renal injury and its stronger effects than salvianolic acid A were replicated in ischemia/reperfusion-induced mouse model (Supplementary Fig. 11). The obviously better kidney-protective effects of salvianolic acid B than salvianolic acid A are in close agreement with the stronger NEU1-binding affinity of the former than the later.

To test whether NEU1 mediate the protective effects of salvianolic acid B, we employed a *Neu1* CKO mouse model (Fig. 9a). In *Neu1* CKO mice, treatment of salvianolic acid B (40 mg/kg) failed to further reduce renal injury and renal fibrosis in response to UUO stimulation, as evidenced by HE and Masson staining (Fig. 9b, c). In agreement, salvianolic acid B cannot further inhibit UUO-induced *Kim1* expression (Fig. 9d), EMT (Fig. 9b, e–g), inflammation factor elevation (Fig. 9h), and collage production (Fig. 9h) in *Neu1* CKO mice. In addition, the inhibition effects on phosphorylations of ALK5 were not further enhanced by salvianolic acid B treatment in *Neu1*CKO mice (Fig. 9i, j). These data indicated that NEU1 is required for salvianolic acid B in renal protection.

## Discussion

This study identified a key role for TEC-located NEU1 in renal injury and renal fibrosis based on the results of genetic, in vivo, in vitro, and pharmacological experiments. The major findings of this study include the following: (i) we observed that NEU1 was significantly elevated in TEC of fibrotic kidneys from human and mice; (ii) we characterized NEU1 as a promotor of renal fibrosis using genetically-engineered mice and epithelial cellular models; (iii) mechanistically, NEU1 interacted with ALK5 at the amino acid 160–200 region and enhanced the ALK5-

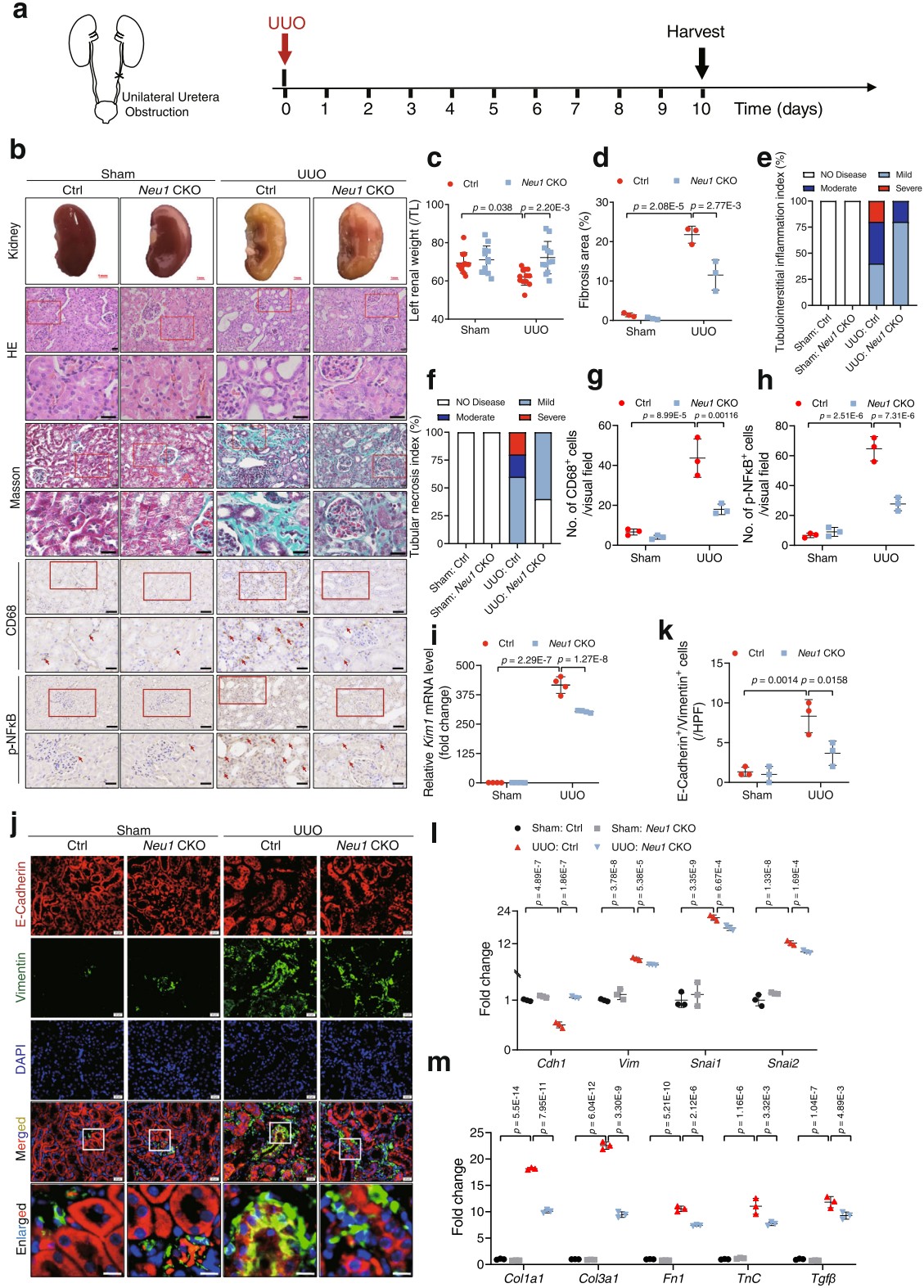

**Fig. 3 | TEC-specific deletion of *Neu1* alleviates UUO-induced mouse renal fibrosis. a** Scheme of the experimental approach. UUO, unilateral ureteral obstruction. **b** The gross appearance of kidneys (Scale bar, 1 mm), Hematoxylin and eosin (HE, Scale bar, 20 μm), Masson's trichrome staining (Scale bar, 20 μm), immunohistochemistry staining of CD68, and p-NFκB (Scale bar, 50 μm) from control (Ctrl) and *Neu1* CKO mice 10 days after UUO. The red arrow indicates positive cells. *n* = 3 mice per group. **c** The ratio of left renal weight to tibia length (TL). *n* = 12 mice per group. (**d**) Statistical results for interstitial collagen analyzed by Image Pro-Plus software. *n* = 3 mice per group. **e**, **f** Morphometric analysis,

assessing percentage of tubular necrosis index (**e**) and tubulointerstitial inflammation index (**f**). *n* = 3 mice per group. **g**, **h** Quantitative results of CD68 (**g**) and p-NFκB (**h**). *n* = 3 mice per group. **i** *Kim1* mRNA levels. *n* = 4 mice per group. **j** Images of immunofluorescence staining. Scale bars, 20 μm. **k** Statistical analysis of staining double-positive cells of E-cadherin and Vimentin. HPF, high power field. *n* = 3 mice per group. **l**, **m** EMT and extracellular matrix-associated gene mRNA levels. *n* = 3 mice per group. Gene expression levels were normalized to *Gapdh*. All statistic data were presented as mean ± SD, two-way ANOVA followed by Tukey's multiple comparisons test.

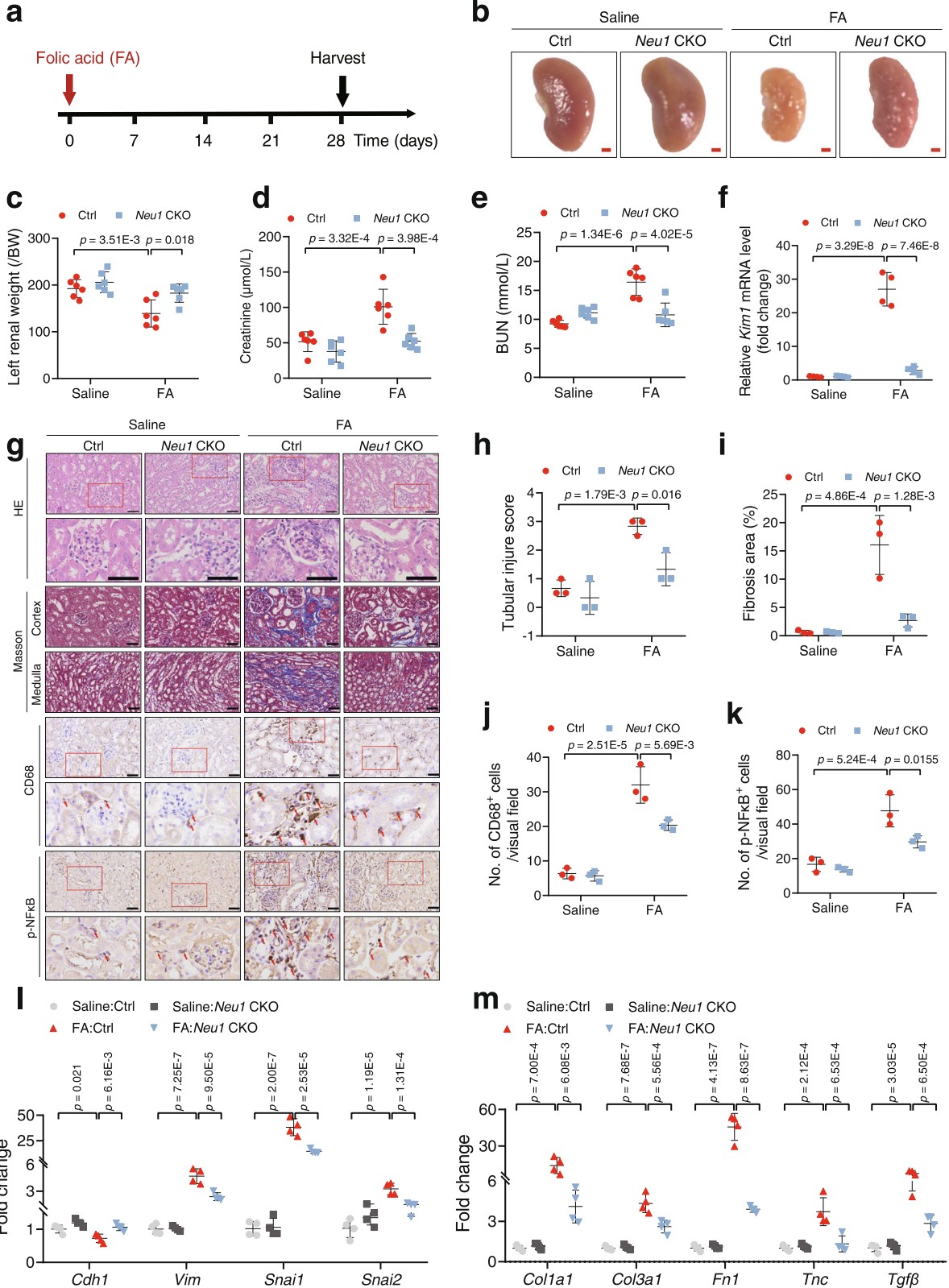

**Fig. 4 | TEC-specific deletion of *Neu1* alleviates folic acid-induced renal fibrosis in mice. a** Scheme of the experimental approach. The mice were intraperitoneally injected with folic acid (250 mg/kg). **b** The gross appearance of whole kidneys from control (Ctrl) and *Neu1* CKO mice 4 weeks after folic acid injection. Scale bar, 1 mm. **c** The ratio of left renal weight to body weight (BW). *n* = 6 mice per group. **d**, **e** Creatinine (**d**) and blood urea nitrogen (**e**) in serum measured by ELISA. *n* = 6 mice per group. **f** *Kim1* mRNA levels. *n* = 4 mice per group. **g** Hematoxylin and eosin (HE), Masson's trichrome staining, and immunohistochemistry staining of CD68

and p-NFκB in kidney sections from Ctrl and *Neu1* CKO mice 28 days after folic acid administration. *n* = 3 mice per group. Scale bar, 50 μm. The red arrow indicates positive cells. **h**–**k** Quantification of tubular injury score (**h**), fibrosis area (**i**), staining of CD68 (**j**), and p-NFκB (**k**). *n* = 3 mice per group. **l**, **m** EMT and extracellular matrix associate gene mRNA level. *n* = 4 mice per group. Gene expression levels were normalized to *Gapdh*. All statistic data were presented as mean ± SD. Two-way ANOVA followed by Tukey's multiple comparisons test. All tests were two-tailed.

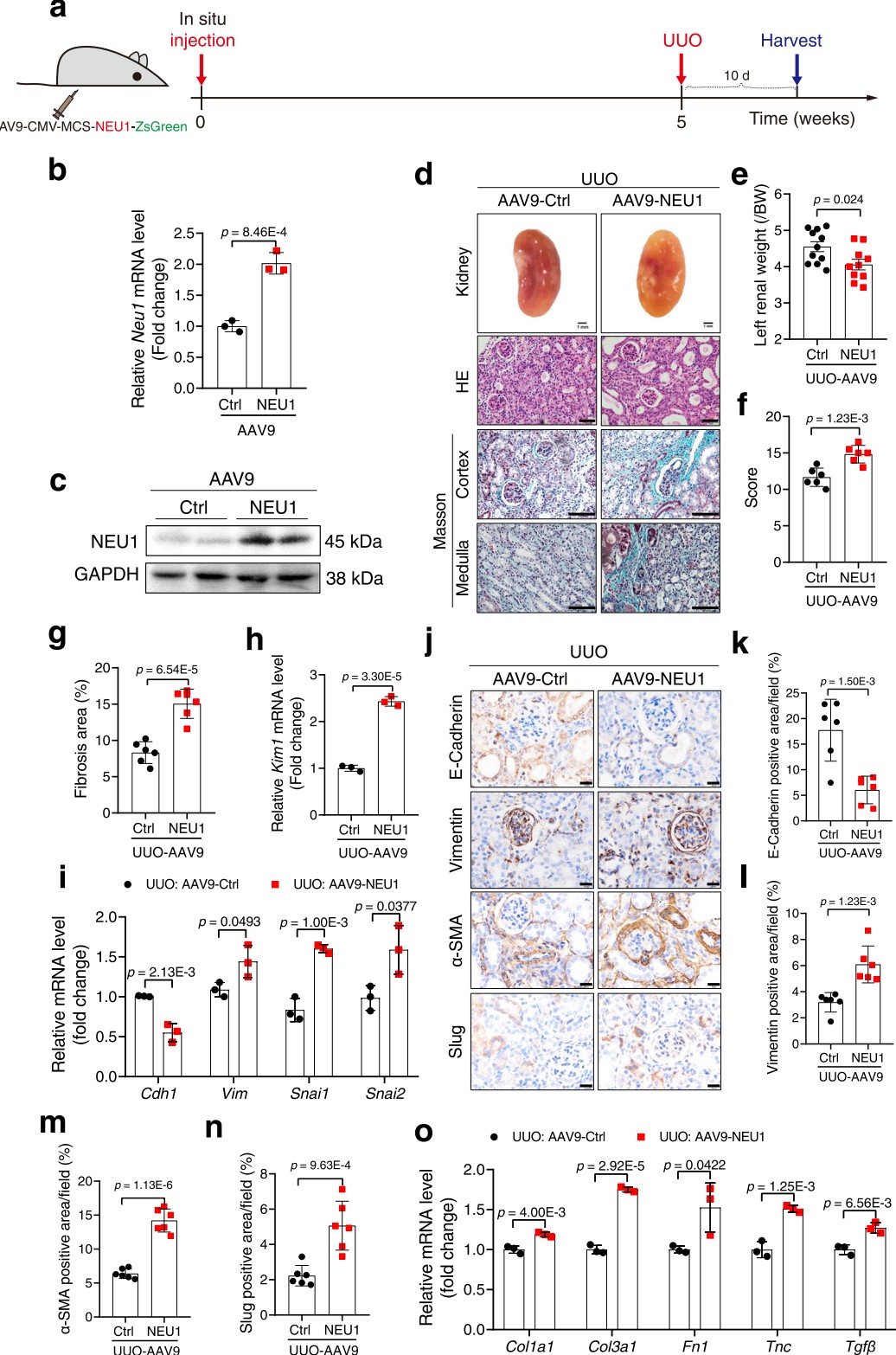

**Fig. 5 | NEU1 overexpression aggravates UUO-induced renal fibrosis.**
**a** Schematic diagram of NEU1 overexpression in mice. Mice were in situ injected with AAV9 encoding NEU1 or scramble. After the injection for 5 weeks, the mice were subjected to UUO surgery. **b** mRNA levels of *Neu1* in the cortices of kidneys. *n* = 3 mice per group. **c** NEU1 protein levels in the kidneys of AAV9-Ctrl and AAV9-NEU1 mice. *n* = 2 mice per group. **d** The gross appearance of kidneys (Scale bar, 1 mm), kidney cross-sections stained with HE (Scale bar, 50 μm), and Masson's trichrome (Scale bar, 100 μm). *n* = 6 mice per group. **e** The ratio of left renal weight to body weight (BW, mg/g). AAV9-Ctrl, *n* = 11; AAV9-NEU1, *n* = 10. **f** Statistical

analysis of tubular injury score. *n* = 6 mice per group. **g** Statistical results for interstitial collagen in d analyzed by Image Pro-Plus software. *n* = 6 mice per group. **h** *Kim1* mRNA level. *n* = 3 mice per group. **i** mRNA levels of the indicated genes in kidneys. *n* = 3 mice per group. Gene expression levels were normalized to *Gapdh*. **j**–**n** Images of immunohistochemical staining using indicated antibodies. Scale bars, 20 μm. *n* = 6 mice per group. **o** mRNA levels of the indicated genes in kidneys. *n* = 3 mice per group. All statistic data were presented as mean ± SD. Unpaired two-tailed *t*-test.

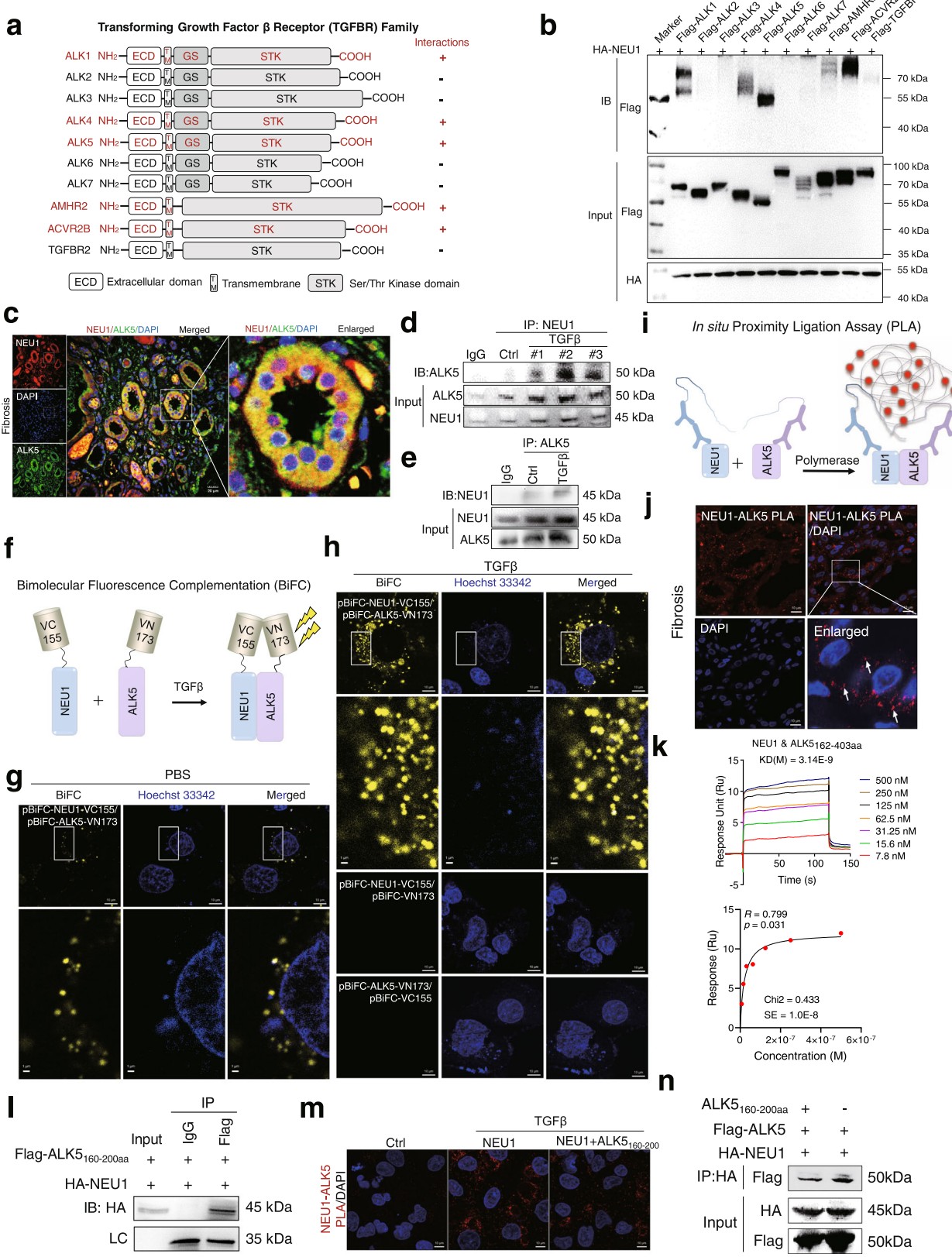

SMAD2/3 signaling pathway; and (iv) salvianolic acid B screened from natural compounds showed high affinity to human NEU1 and effectively prevented renal injury.

The role of neuraminidases in organ fibrosis remains poorly understood, especially in renal fibrosis. In this work, we demonstrated that NEU1 was noticeably elevated in patients with renal fibrosis, in mice subjected to UUO or administered folic acid, and in TGFβ-stimulated human tubular epithelial HK-2 cells. The increased NEU1 was mainly localized in the TECs. Our results were consistent with a previous study that reported an upregulation of NEU1 in lung epithelial cells of patients with pulmonary fibrosis[38]. However, the expression of NEU1 in the fibroblasts of patients with pulmonary fibrosis was

**Fig. 6 | NEU1 interacts with GS domain of ALK5. a** Schematic representation of the full-length forms of transforming growth factor β family proteins (ALK1, ALK2, ALK3, ALK4, ALK5, ALK6, ALK7, AMHR2, ACVR2B, and TGRBR2 receptors). The red "+" indicates the combination with NEU1. ECD, extracellular domain; TM, transmembrane domain; GS, glycine-serine repeats; STK, serine/threonine kinase domain. **b** Co-immunoprecipitation of NEU1 with transforming growth factor β family proteins in HEK293T. Two independent experiments were performed. **c** Confocal images of NEU1 (red) and ALK5 (green) localization in the fibrotic kidney of patients. Scale bar, 20 μm. **d, e** Co-immunoprecipitation of NEU1 and ALK5 in HK-2 cells stimulated with TGFβ (10 ng/ml) for 24 h. Two independent experiments were performed. **f** Scheme of NEU1 and ALK5 fusion proteins used for bimolecular fluorescence complementation (BiFC) analysis. **g, h** BiFC signals were detected in HK-2 cells. Representative fluorescence images of HK-2 cells co-expression of NEU1-VC155 and ALK5-VN173 plasmid without (**g**) or with (**h**) TGFβ stimulation. Scale bar, 10 μm. The magnified image scale was 1 μm. **i** Schematic diagram of in situ proximity ligation assay (NEU1-ALK5 PLA). **j** Interaction between NEU1 and ALK5 (NEU1-ALK5, red arrow) was analyzed by PLA in the fibrotic kidney of patients. Scale bars, 10 μm. **k** The interaction between NEU1 and ALK5$_{162-403aa}$ tested by SPR. The frequency response and fitting curves were displayed, two-tailed Pearson's test. **l** Co-immunoprecipitation of NEU1 and ALK5$_{160-200aa}$ in HEK293T cells. **m** Interaction between NEU1 and ALK5 (NEU1-ALK5, red) was analyzed by PLA in HK-2 cells transfected with ALK5$_{160-200aa}$ plasmid. Scale bars, 10 μm. **n** Co-immunoprecipitation of NEU1 and ALK5 in HEK293T cells transfected with ALK5$_{160-200aa}$ plasmid. **c, g, h, j** and **l–n** were repeated three times independently with similar results.

decreased in another publication[39]. Large multicenter samples are therefore warranted to resolve this inconsistency.

In an effort to investigate the function of NEU1 in CKD, we generated TEC-specific NEU1-knockout mice using a Cre-loxP conditional gene targeting as well as NEU1-overexpressed mice via adeno-associated virus gene transfer. Conditional genetic engineering employed in this work is more powerful than conventional knockout/knock-in technology for studying NEU1 function. Tubule-specific NEU1 loss was sufficient to protect kidney from UUO- or folic acid-induced injury in mice and TGFβ-induced injury in HK-2 cells. These altered phenotypes and molecular markers included EMT, inflammatory cytokines, fibrosis-associated genes, and renal morphology. Conversely, NEU1 overexpression aggravated fibrosis-associated phenotypes in vitro and in vivo. Although the influence of neuraminidases in other renal cell types such as myofibroblasts cannot be totally excluded in renal fibrosis, this work sheds light on a clear promotor role for tubular NEU1 in renal injury.

This work centered on the role of NEU1 in renal fibrosis. Aside from NEU1, there are 3 other isoforms (NEU2, NEU3, and NEU4) in the human neuraminidase family. These mainly differ in their protein sequences, subcellular localization, and enzymatic properties. NEU1, the most abundant form, displays less than 10% similarity with the other 3 isoforms[40]. NEU2, NEU3, and NEU4 share largely common amino acid sequences. Different neuraminidase members may produce variable effects depending on the organ and cell type. We and others have demonstrated the contribution of NEU1 in cardiovascular diseases[20,21,24,25,41,42]. NEU2 and NEU4 were shown to inhibit cancers[43,44]. NEU3 was reported to act as a pro-inflammatory mediator in the intestine and lung, but trigger inflammation in brain[45–47]. Besides NEU1 that has been decoded in this work, the roles of NEU2, NEU3, and NEU4 in CKD progression need to be explored in-depth in the near future.

NEU1 deficiency is associated with sialic acid-rich macromolecular storage, leading to lysosomal disorder, sialidosis[17,48]. There were no significant morphological differences between *Neu1*CKO mice and the wild-type mice. Besides, the inflammation, fibrosis, and EMT indicators in the kidney of *Neu1*CKO mice are almost similar with those in the control mice. In addition, we did not observe any side effects in tubule-specific NEU1-knockout mice compared with the wide-type mice for up to 10 weeks (Figs. 3 and 4). This was in consistent with what was observed in cardiomyocyte-specific NEU1-knockout mice by us and others[20,39]. In line with this, there is no case report on sialidosis syndrome caused by use of neuraminidase inhibitors such as oseltamivir for up to 6 weeks[49]. In general, pathological changes were observed in mice with systemic knockout of NEU1 but less in tissue-conditional KO mice[39,50].

NEU1 was originally described to be localized in lysosomes, where it is involved in breakdown of sialo-glycoconjugates by removing terminal sialic acids[15]. Emerging evidence has demonstrated that NEU1 can be sorted to the cell surface to desialylate membrane receptors such as TLR4 and insulin receptors[19,51–53]. We have previously shown

that NEU1 translocated to the nucleus to bind with transcriptional factors GATA4, promoting cardiac hypertrophy and remodeling[20]. In this work, we have demonstrated that NEU1 was activated in the cytoplasm to bind to ALK5, promoting renal fibrosis. ALK5 is composed of small cysteine-rich extracellular parts, single transmembrane regions, and intracellular parts. The intracellular parts can be further divided into GS domain, serine/threonine kinase domain, and C-terminal tail[37]. In general, various ligands activated the TGFβ type I receptor by binding to the extracellular parts of ALK5[54–56]. In contrast, we found that NEU1 was directly bound to the GS domain of the intracellular parts rather than the extracellular parts. Inhibiting the binding of NEU1 and ALK5 significantly suppressed SMAD2/3 activation. Actually, the GS domain located at the 160–200 region is responsible for the phosphorylation of the TGFβ I receptor kinase and activation of downstream signaling pathway[36]. It is thus presumed that the binding of NEU1 with the GS domain promoted the phosphorylation of the serine/threonine kinase domain.

Interestingly, NEU1 affected ALK5 sialylation in the presence of TGFβ, and NEU1 enzymatic activity is required for enhancing the TGFβ pathway. It appears that the level of NEU1 determines the sialyaltion of ALK5 and the NEU1 enzymatic activity affects ALK5 pathway. However, the exact molecular basis remains unknown and depends on the characterization of the crystal structure of NEU1-ALK5 complex.

Development of candidates inhibiting or interacting with NEU1 is still a challenge because of its less investigation and unavailability of crystal structure. We employed a SPR strategy to identify NEU1-interacting compounds from medicinal plants. Salvianolic acid B, the most abundant ingredient in *Salvia miltiorrhiza*, exhibited the strongest affinity with NEU1 than other compounds. Salvianolic acid B inhibited NEU1 protein expression and enzyme activity (Supplementary Fig. 12), and suppressed the interaction between NEU1 and ALK5. In agreement with the affinity assay, salvianolic acid B showed promising kidney-protective effects and clearly stronger effects than salvianolic acid A in UUO-induced and ischemia/reperfusion-induced mouse models. Several reports have also shown that salvianolic acid B improved kidney dysfunction by inhibiting endoplasmic reticulum stress and PI3K/Akt/Nrf2 pathway, or by activating sirt1-mediated autophagy and Nrf2/NLRP3-mediated pyroptosis[57–60]. Small-molecule drugs have the potential to interact with multiple targets and exhibit diverse pharmacological activities[61–63]. It is likely that salvianolic acid B inhibited renal fibrosis through intricate, distinct, and interrelated signaling pathways. Here, we provide evidence that NEU1 is one of direct interacting target for salvianolic acid B, and NEU1 is required for this compound's effects in renal protection.

This study has some limitations. First, we have shown that NEU1 interacted with the GS domain of ALK5, but the specific binding sites for NEU1 were not characterized because of the paucity of information on this protein. Second, in addition to activating the ALK5-SMAD2/3 signaling pathway, NEU1 may also be involved in other signaling pathways that remain to be explored. Thirdly, whether other signaling

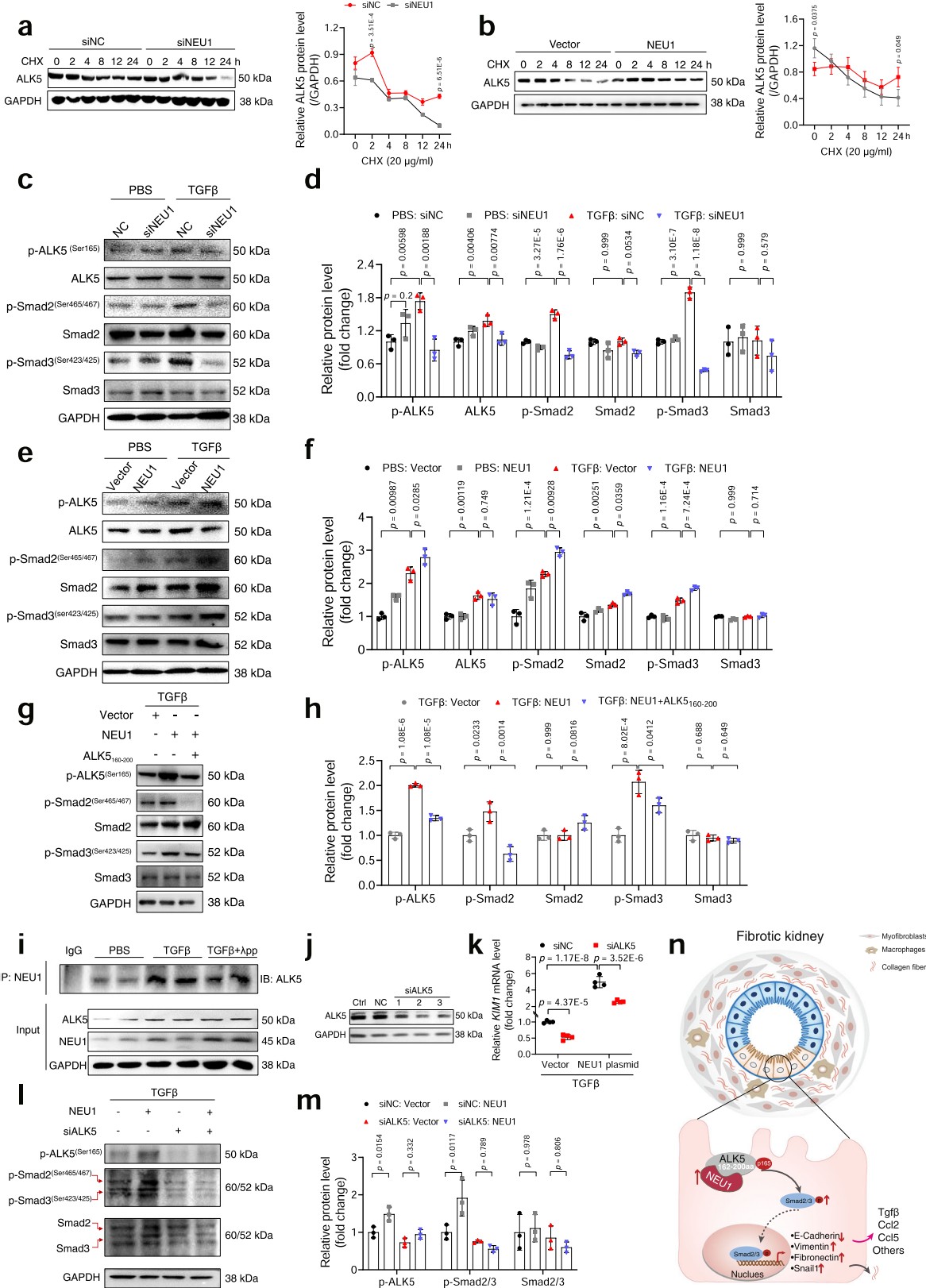

pathways or targets are involved in salvianolic acid B's effects against renal fibrosis as well as their complex interrelated mechanisms remain to be explored.

In conclusion, this study identifies a promotor role for NEU1 in renal fibrosis and suggests a potential therapeutic approach by targeting NEU1 to treat CKD.

## Methods

### Human kidney biopsy studies

This study complied with the ethical guidelines of the 1975 Declaration of Helsinki. Human kidney biopsy samples were obtained from Jiangsu Province Hospital. The protocol concerning the use of human kidney biopsy samples in this study was approved by the Committee on

**Fig. 7 | NEU1 stabilizes ALK5 and enhances ALK5-SMAD2/3 signaling pathway.**
**a, b** HK-2 cells transfected with siNEU1 (**a**) or NEU1 full-length plasmid (**b**) for 24 h and treated with TGFβ (10 ng/ml) for 24 h. Then the cells were incubated with cycloheximide (CHX, 20 μg/ml) for the indicated periods of time (0, 2, 4, 8, 12, 24 h) (left). Lysates were harvested from the cells and analyzed by western blots. Quantitation of ALK5 protein levels were shown in the right pane. $n = 3$ samples per group. Unpaired $t$-test. **c–f** Western blots (**c, e**) and quantitative results (**d, f**) of p-ALK5, ALK5, p-SMAD2/3, and SMAD2/3 in HK-2 cells transfected with siNEU1 or NEU1 plasmid for 24 h and treated with TGFβ (10 ng/ml) for 24 h. $n = 3$ samples per group. Relative protein levels were shown after normalization to GAPDH. One-way ANOVA followed by Tukey's multiple comparisons test. **g, h** Western blots (**g**) and quantitative results (**h**) of p-ALK5, p-SMAD2/3, and SMAD2/3 in HK-2 cells transfected with NEU1 and ALK5$_{160\text{-}200}$ plasmid for 24 h and treated with TGFβ (10 ng/ml) for 24 h. $n = 3$ samples per group. Relative protein levels were shown after normalization to GAPDH. One-way ANOVA followed by Tukey's multiple comparisons test. **i** Co-immunoprecipitation of NEU1 and ALK5 in HK-2 cells stimulated with TGFβ (10 ng/ml) for 24 h. λpp, Lambda Protein Phosphatase. Two independent experiments were performed. **j** Western blots of ALK5 in HK-2 cells. siALK5-1, siALK5-2, and siALK5-3 were 3 different siRNA sequences. HK-2 cells were transduced with siALK5 for 48 h. Two independent experiments were performed. **k–m** mRNA levels of KIM1 (**k**) and protein level of p-ALK5, p-SMAD2/3, and SMAD2/3 (**l, m**) in HK-2 cells transduced with NEU1 full-length plasmid and ALK5 siRNA for 48 h. KIM1 mRNA normalized to GAPDH. Relative protein levels were shown after normalization to GAPDH. **k** $n = 3$ samples per group. **l, m** $n = 3$ samples per group. Two-way ANOVA followed by Tukey's multiple comparisons test. **n** The proposed mechanisms of NEU1-mediated renal fibrosis. All statistic data were presented as mean ± SD. All tests were two-tailed.

Research Ethics of Jiangsu Province Hospital. The ethic number is 2016-SR-029. The written, informed consent to participate was obtained from all study participants. All the human study participants agreed to participation for free. Eight participants who underwent biopsy procedures because of microscopic hematuria but had no specific pathologic alterations were designated as the non-renal fibrosis group. Eight CKD patients were diagnosed renal fibrosis by biopsy procedures. Detailed information on participants is provided in Supplementary Table 1.

## Mice
Mice were maintained in the center for Experimental Animals at China Pharmaceutical University, Nanjing, China. All procedures involving experimental animals were performed following protocols approved by the Committee for Animal Research of China Pharmaceutical University and conformed to the Guide for the Care and Use of Laboratory Animals. The ethic number is 2021-11-003. Mice of the same genotype were randomly assigned to different treatment groups. C57BL/6J male mice aged 8–10 weeks (body weights of 20–23 g) were used. All animals were maintained under constant humidity and temperature at standard facilities under specific pathogen-free conditions with free access to water and chow (10% kcal from fat; Xietong Organism, China). Mice were euthanized by cervical dislocation. The investigators were not blinded to treatments, but no subjective assessments were made.

## Renal tubular epithelial cell-specific *Neu1* knockout mice
To specifically knockout *Neu1* in renal tubular epithelial cells, we generated a mouse line by crossing females of the *Neu1*-floxed line (*Neu1*$^{fl/fl}$) with the Ksp-Cre (B6.Cg-Tg (Ksp1.3-cre) 91Igr/J, male, aged 8–9 weeks) transgenic strain obtained from Shanghai Model Organisms Center, Inc. The filial 1 progeny, mice (male or female), litters with heterozygous deletion of *Neu1* gene (*Neu1*$^{fl/+}$) that harbored the Ksp1.3/Cre transgene (*Neu1*$^{fl/+}$; Ksp1.3-Cre) were obtained and they were further crossed with the opposite sex of *Neu1*$^{fl/fl}$ mice to obtain mice expressing complete deletion of *Neu1* in the filial 2 progeny (*Neu1*$^{fl/fl}$; Ksp1.3-Cre), referred to as *Neu1* CKO mice.

## Adeno-associated virus (AAV) infected mice
To overexpression NEU1 in the kidney, AAV9-mediated delivery was employed. C57BL/6J mice were used (male, 8–10 weeks, B&K Universal Group Ltd. Shanghai, China). AAV9 encoding mouse NEU1 was provided by Hanheng Biotechnology (Shanghai, China). Briefly, male mice were anesthetized with pentobarbital sodium (30 mg/kg, Sigma-Aldrich, catalog no. P3761) by intraperitoneal injection, and were injected with $1.5 \times 10^{12}$ vector genome (vg)/ml HBAAV9-CMV-Neu1-3xflag-ZsGreen and $1.5 \times 10^{12}$ vg/ml HBAAV9- ZsGreen control into 5 different sites (10 μl at each site) of the renal cortex with a glass micropipette.

## Unilateral ureteral obstruction (UUO)-induced fibrosis model
Male mice were anesthetized with pentobarbital sodium (30 mg/kg) by intraperitoneal injection. The abdomen was opened, and the left ureter was ligated with 5-0 silk. The abdomen was then closed with running sutures and the skin was closed with interrupted sutures. After surgery, the mice were maintained in a temperature-controlled room with a 12-h light-dark cycle and were reared on standard chow and water ad libitum.

## Folic acid (FA)-induced fibrosis model
Male mice (8 weeks old) were intraperitoneally injected with a single dose of folic acid (250 mg/kg, Sigma-Aldrich, 7876) dissolved in 0.3 M sodium bicarbonate. Mice injected with sodium bicarbonate served as vehicle control (Saline).

## IR-induced renal injure (IRI) model
Male mice were anesthetized with pentobarbital sodium (30 mg/kg) by intraperitoneal injection. The abdomen was opened, and bilateral renal pedicles were clipped for 30 min using microaneurysm clamps. During the ischemic period, body temperature was maintained between 37 °C using a heat pad. After removal of the clamps, reperfusion of the kidneys was visually confirmed. Mice were euthanized 24 h after IRI, and kidney tissues were collected for analyses. Sham control animals were subjected to the identical operation without renal pedicle clamping.

## Salvianolic acid A and salvianolic acid B treatment in vivo
Based on the dosage conversion among different species and previous reports, the mice were injected salvianolic acid A and salvianolic acid B at 40 mg/kg/d by a tail vein. C57BL/6J mice (male, 8–10 weeks, B&K Universal Group Ltd. Shanghai, China) underwent UUO or IRI surgery. In UUO-induced model, daily injected of the salvianolic acid A and salvianolic acid B or saline was started on the day of UUO surgery and lasted for 10 days. In IR-induced mouse model, mice were injected salvianolic acid A and salvianolic acid B or a comparable volume of saline continuous 3 days before IRI surgery.

## Histological analysis
Mouse kidneys were fixed in 4% paraformaldehyde (PFA, Sigma-Aldrich, catalog no. P6148) in PBS and embedded in paraffin. Paraffin was cut into sections using a paraffin microtome with stainless steel knives. The sections were mounted on glass slides, deparaffinized with xylene, dehydrated through graded series of ethanol, and stained with hematoxylin-eosin. The tubules were evaluated according to the following scoring system: 0 = no tubular injury; 1 = 10% or fewer tubules injured; 2 = 11–25% tubules injured; 3 = 26–50% tubules injured; 4 = 51–74% tubules injured; and 5 = 75% or more tubules injured. The tubulointerstitial inflammation index were evaluated according to degree of inflammatory cell infiltration: 0 = no inflammatory cell infiltration; 1 = 10% or fewer inflammatory cell infiltration; 2 = 11–25% inflammatory cell infiltration; 3 = 26–50% inflammatory cell infiltration;

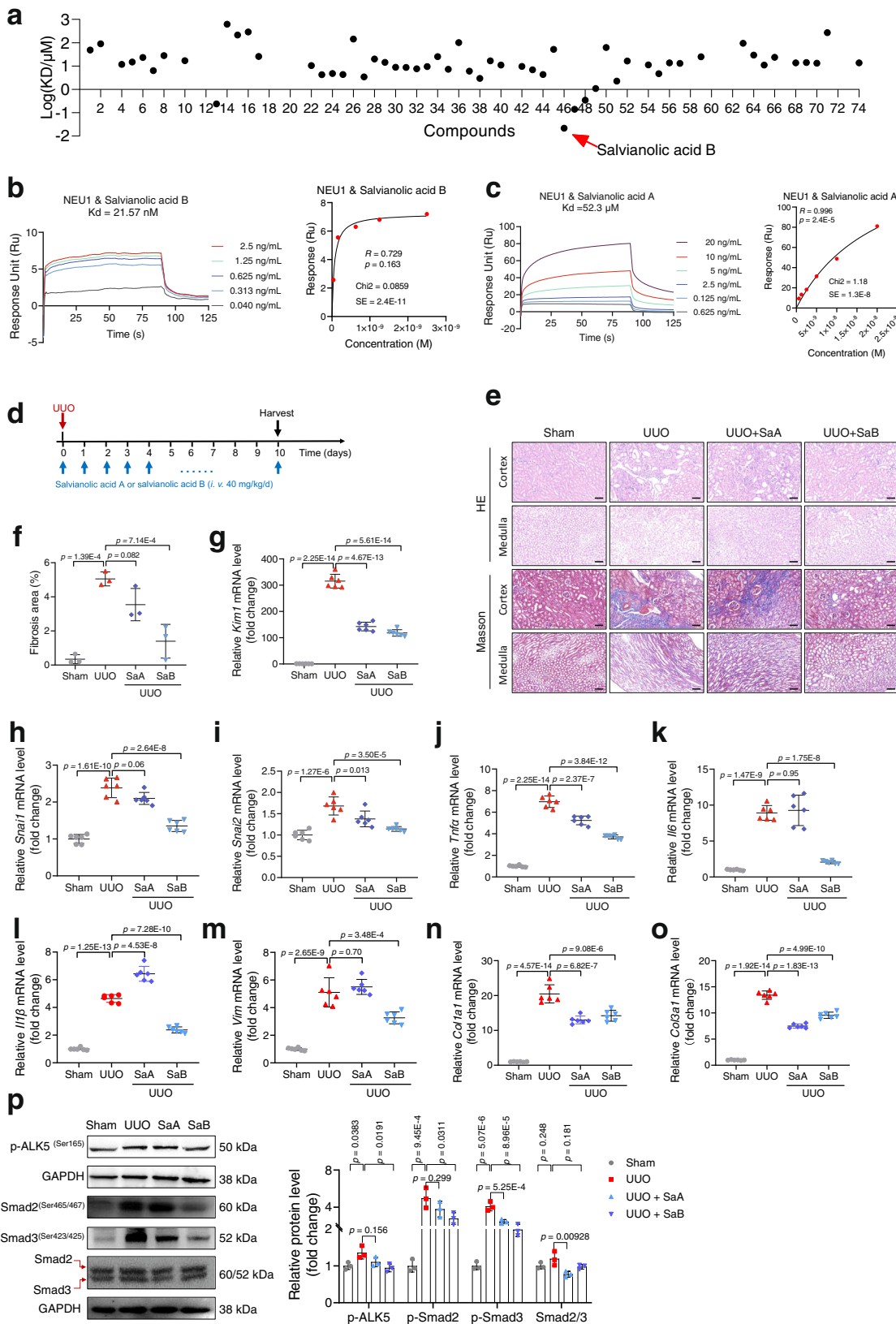

and 4 = 75% or more inflammatory cell infiltration. To evaluate collagen deposition, sections were stained with Masson's Trichrome. The fibrotic areas were calculated by Image Pro-Plus software (Media Cybernetics, Rockville, MD, USA) as the summation of blue-stained areas divided by total ventricular area. Image acquisition was performed on the Hamamatsu Nano Zoomer 2.0 RS scanner.

**ELISA assay**

Creatinine and urea nitrogen concentration in serum were measured using the Creatinine (Cr) Assay kit and Urea Assay (BUN) kit following the manufacturer's instructions. These two kits were purchased from Nanjing Jiancheng Bioengineering Institute. IL6 and TNFα concentrations in serum were measured using the Mouse IL6 ELISA Kit and

**Fig. 8 | Targeting NEU1 by salvianolic acid B alleviates UUO-induced renal fibrosis. a** The interactions between 74 compounds with recombinant human NEU1 determined by surface plasmon resonance (SPR). $K_D$, dissociation constant. The compound 3, 9, 11, 12, 18, 19, 20, 21, 41, 53, 58, 60, 61, 62, 67, 72, 73 have no $K_D$ value because they do not bind to NEU1. **b**, **c** The interaction between NEU1 and salvianolic acid B (SaB) (**b**) and salvianolic acid A (SaA) (**c**) was tested by SPR. The frequency response and fitting curves were displayed. Pearson's test. **d** Scheme of the experimental approach. UUO, unilateral ureteral obstruction. **e** Hematoxylin and eosin (HE) and Masson's trichrome staining from control (Ctrl) and SaA or SaB-treated mice 10 days after UUO. Scale bar, 50 mm. $n = 3$ mice per group. **f** Statistical results for interstitial collagen analyzed by Image Pro-Plus software. $n = 3$ mice per group. **g**–**o** the mRNA levels of kidney injury molecule 1 (*Kim1*, **g**), EMT associate genes (*Snai1* and *Snai2*, **h**, **i**), inflammatory cytokines associate genes (*Tnfα*, *Il1*, and *Il6*, **j**–**l**) and extracellular matrix associate genes (*Vim*, *Col1a1*, and *Col3a1*, **m**–**o**) in kidney samples. All normalized to *Gapdh*. $n = 6$ samples per group. **p** Western blots (**p**, left panel) and quantitative results (**p**, right panel) of p-ALK5 (ser165), p-SMAD2/3, and SMAD2/3 in kidney from control (Ctrl) and SaA or SaB-treated mice 10 days after UUO. $n = 3$ mice per group. All statistic data were presented as mean ± SD, one-way ANOVA followed by Dunnett's multiple comparisons test. All tests were two-sided.

Mouse TNFα ELISA Kit following the manufacturer's instructions. These two kits were purchased from Proteintech Group, Inc.

### Neuraminidase enzyme activity assay
Neuraminidase enzyme activity was measured by a fluorometric assay with substrate 2′-(4-methylumbelliferyl)-α-d-N-acetylneuraminic acid (4-MU-NANA) (Abcam, ab138888). Samples were lysed with 0.1% Triton X-100. After centrifugation at $12,000 \times g$ at 4 °C for 15 min, the supernatants were subjected to a neuraminidase assay at 37 °C for 1 h. Reaction products were measured at 320 nm for excitation and 450 nm for emission with fluorescence microplate reader (SpectraMaxiD5, Molecular, USA).

### Immunohistochemistry analysis
Mouse kidneys were fixed in 4% paraformaldehyde for at least 24 hours and embedded in paraffin. Paraffin sections were deparaffinized and hydrated in graded ethanol series before staining with the peroxidase-antiperoxidase method. Antigens were retrieved by boiling the sections for 20 min in 10 mM citric acid solution (pH = 6) or 10 mM Tris and 1 mM EDTA (pH = 9). Endogenous peroxidase was blocked by incubation in 3% hydrogen peroxide. The sections were incubated overnight at 4 °C with primary antibodies. The sections were incubated with the corresponding secondary antibodies. 3,3′-diaminobenzidine (DAB) (Sigma-Aldrich, St Louis, MO) was used as chromogen. Sections were lightly counterstained with hematoxylin and were dehydrated and coverslipped. Image acquisition was performed on the Hamamatsu NanoZoomer 2.0 RS scanner.

### Immunofluorescence staining
For the kidney tissues immunofluorescence staining, sections were permeabilized with 0.1% Triton X-100 and blocked with 5% BSA in PBS for 20 min at room temperature. Then, sections were incubated at 4 °C overnight with the primary antibodies. Fluorescently labeled secondary antibodies were used. Slides were counterstained with DAPI. Samples were analyzed, and pictures were taken using FV3000 confocal scanning microscope.

### Transmission electron microscopy (TEM)
Renal tubules and glomeruli structure were examined by standard transmission electron microscopy. Fresh kidney was fixed with a mixture of 2.5% glutaraldehyde 2 h, washed, dehydrated, and embedded in resin according to standard procedures. Embedded samples were analyzed by a JEOL 1010 electron microscope (Tokyo, Japan).

### PCR array
Gene expression profiling was analyzed by a mouse fibrosis PCR Array (Wcgene Biotech, Shanghai, China) according to the manufacturer's protocol. The *β-actin* and *Gapdh* were used as endogenous controls. The relative gene expression levels of target genes were calculated using the $2^{-\Delta\Delta Ct}$ method. Data were normalized to the reference gene based on the cycle threshold (Ct) values. The log2 (fold-change) was calculated based on the $2^{-\Delta\Delta Ct}$ method.

### Cell culture, transfection, and treatment
Human proximal tubular epithelial cells (HK-2) and human embryonic kidney cells (HEK293T) were obtained from the Cell Bank of the Chinese Academy of Sciences (Shanghai, China) and cultured in DMEM/F12 or DMEM medium (KeyGEN BioTECH) containing 4.5 g/L of D-glucose supplemented with 10% fetal bovine serum (FBS, Gibco).

HK-2 cells were transfected with siNEU1/siALK5 at the indicated sequences or transfected with NEU1 expression plasmid (HA-NEU1). HEK293T cells were transfected with HA-NEU1, Flag-ALK5$_{160-200aa}$, Flag-ALK1, Flag-ALK2, Flag-ALK3, Flag-ALK4, Flag-ALK5, Flag-ALK6, Flag-ALK7, Flag-AMHR2, Flag-ACVR2B, or Flag-TGFBR2 plasmids. siRNA transfected with Lipofectamine 2000 and plasmids transfected with Lipofectamine 3000 according to the manufacturer's protocol (Thermo Fisher Scientific).

For NEU1 maintain SMAD2/3 activation test, the HK-2 cells were transfected with siNEU1 or HA-NEU1 plasmid and then stimulated with or without TGFβ (10 ng/ml, PeproTech) for 0, 6, 12, 24, 36, or 48 h. For ALK5 stability test, HK-2 cells were treated with cycloheximide (CHX; MCE, HY-12320) with or without siNEU1/HA-NEU1 at different time points (0, 2, 4, 8, 12, 24 h), then the cells were stimulated with TGFβ (10 ng/ml) for 24 h. HK-2 cells were treated with salvianolic acid A (10 μM) and salvianolic acid B (5 μM or 10 μM) for 24 h and stimulated with or without TGFβ (10 ng/ml, PeproTech) for 24 h.

For protein dephosphorylation test, HK-2 cells were stimulated with TGFβ (10 ng/ml) for 24 h. Then the cells were lysed with 25 mM Tris-HCl (pH 7.4) lysis buffer containing 150 mM NaCl, 1% NP-40, 1 mM EDTA, 5% glycerol. The cell lysis solution was incubated with lambda protein phosphatase (λpp, 100 U/ml, Beyotime) and MnCl$_2$ (1 mM) for 30 min at 30 °C.

### Plasmids and siRNA
HA-NEU1, Flag-ALK5$_{160-200aa}$, Flag-ALK1, Flag-ALK2, Flag-ALK3, Flag-ALK4, Flag-ALK5, Flag-ALK6, Flag-ALK7, Flag-AMHR2, Flag-ACVR2B, Flag-TGFBR2, pBiFC-VC155, pBiFC-VN173, pBiFC-ALK5-VN173, pBiFC-NEU1-VC155, and NEU1 (D103A/Y370A/E394A)-HA were generated by cloning the indicated gene into pcDNA3.1. Those plasmids were purchased from Public Protein/Plasmid Library (Nanjing, China). Human NEU1 siRNA (siNEU1-1/2/3), human ALK5 siRNA (siALK5-1/2/3), and their vector controls were purchased from GenePharma (Beijing, China), siRNA sequences were provided in Supplementary Table 2.

### Real-time fluorescence quantitative PCR (RT-qPCR)
Total RNA was isolated from tissues or HK-2 cells by TRIzol reagent (Invitrogen) according to the manufacturer's instructions. RNA was reverse transcribed using the Hifair® II 1st Strand cDNA Synthesis SuperMix for qPCR (Yeasen Biotech) according to the manufacturer's protocol. The cDNA was stored at −20 °C until use. The real-time polymerase chain reaction was performed using Hieff® qPCR SYBR Green Master Mix (Yeasen Biotech) according to the manufacturer's instructions. The data were normalized to *Gapdh* and analyzed using the ΔΔCt method. The primers used are listed in Supplementary Table 3.

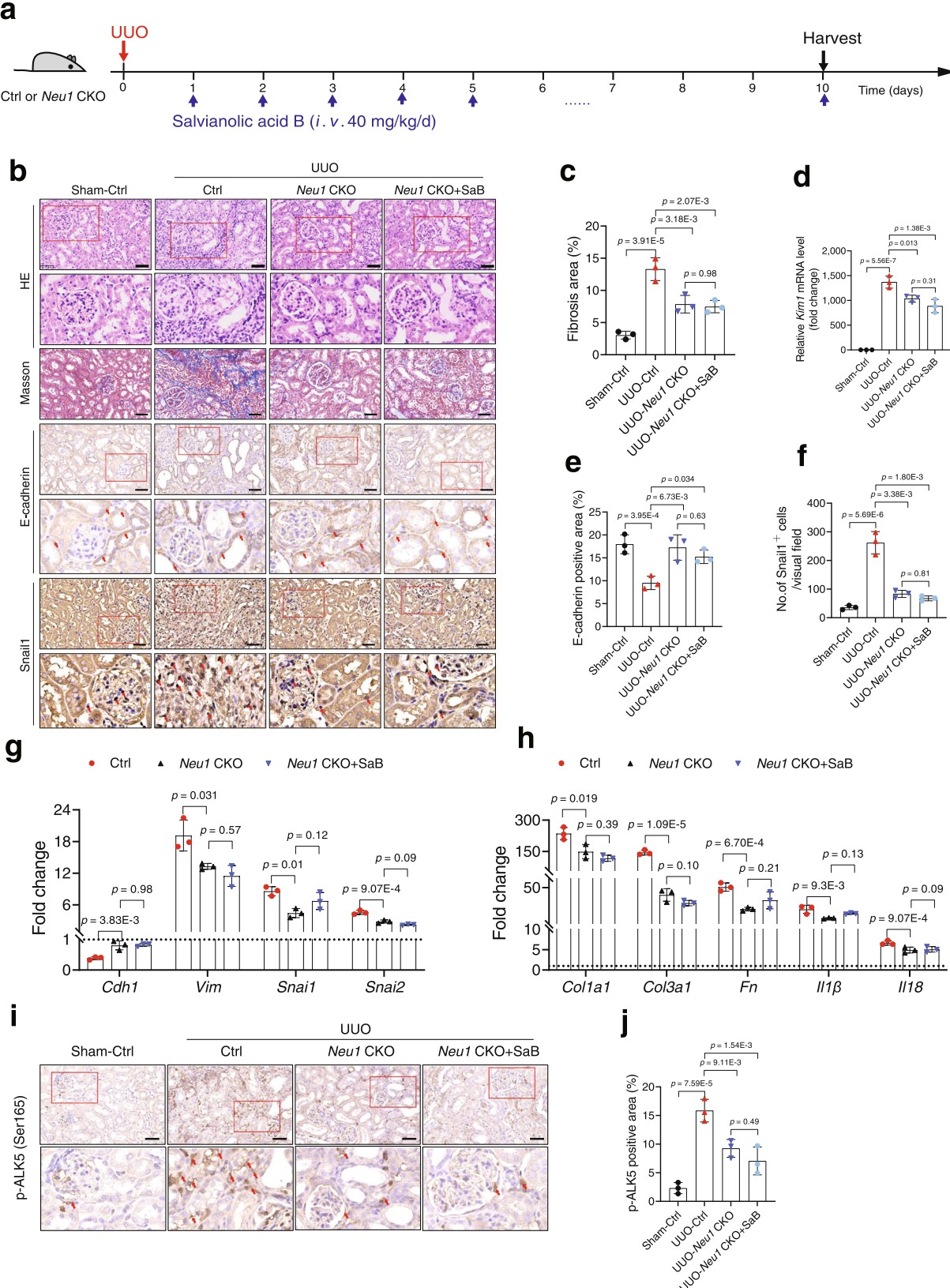

## Co-immunoprecipitation

HK-2 or HEK293T cells were lysed with 25 mM Tris-HCl (pH 7.4) lysis buffer containing 150 mM NaCl, 1% NP-40, 1 mM EDTA, 5% glycerol, and protease inhibitor (Roche). The cell lysates were mixed with NEU1, ALK5, Flag-tag, or HA-tag antibody that was preincubated with Protein A/G Magnetic Beads (Thermo Scientific) for 12 h and washed three times with cold lysis buffer excluding 0.1% SDS. Coprecipitates with NEU1, ALK5, SNL, Flag, or HA proteins were separated by SDS-PAGE and subjected to western blots with ALK5, NEU1, SNL, HA, or Flag antibodies. The non-heavy chain IgG secondary antibody (Santa Cruz, m-IgGκ BP-HRP, sc-516102) was used.

**Fig. 9 | NEU1 mediates the renal protective effects of salvianolic acid B.**
**a** Scheme of the experimental approach. The *Neu1* CKO mice were treated with salvianolic acid B (SaB) at the indicated doses for 10 continuous days after being subjected to UUO surgery. **b** Representative gross appearance of kidneys (Scale bar, 2 mm), kidney cross-sections stained with HE (Scale bar, 50 μm), Masson's trichrome (Scale bar, 50 μm), immunohistochemical staining with E-Cadherin and Snail (Scale bar, 50 μm). *n* = 3 mice per group. The red arrow indicates positive area. **c** Statistical results for interstitial collagen in b analyzed by Image Pro-Plus software. *n* = 3 mice per group. **d** *Kim1* mRNA level. *n* = 3 mice per group. **e, f** Quantification of staining of E-Cadherin (**e**) and Snail1 (**f**). *n* = 3 mice per group. **g, h** mRNA levels of the indicated genes in kidneys determined by qRT-PCR. *n* = 3 mice per group. Dotted line represents the expression in sham control tissue. Gene expression levels were normalized to *Gapdh*. **i** Representative image of immunohistochemical staining with p-ALK5 (ser165) (Scale bar, 50 μm). *n* = 3 mice per group. **j** Quantification of staining of p-ALK5 (ser165). *n* = 3 mice per group. All data were presented as mean ± SD, one-way ANOVA followed by Tukey's multiple comparisons test. All tests were two-tailed.

## Proximity ligation assay (PLA)

Interactions between NEU1 and ALK5 was detected by in situ proximity ligation assay (PLA) in HK-2 cells and human kidney biopsy sample. For HK-2 cells, cells were grown on glass bottom dish and fixed with 4% paraformaldehyde in PBS for 15 min. For paraffin-embedded human kidney biopsy sample, following dewaxing and rehydration of tissue sections, antigen retrieval was performed by heating the slides for 30 min at 95 °C in Tris-EDTA buffer. From this point, the tissue sections and the fixed HK-2 cells were treated identically, and the PLA protocol was followed according to the manufacturers' instructions. The In-situ PLA (Duolink® in situ Detection Reagents Red Kit, Sigma-Aldrich, DUO92008; Duolink® in situ PLA Probe Anti-Rabbit PLUS, Sigma-Aldrich, DUO92002; Duolink® in situ PLA Probe Anti-Mouse MINUS, Sigma-Aldrich, DUO92004) was performed according to manufacturer's instruction. Fluorescence signal amplification was observed using the Zeiss LSM800 confocal laser scanning microscopy.

## Bimolecular fluorescence complementation (BiFC)

HK-2 cells were plated on glass bottom dish. Transfections were carried out using the lip3000 reagent, with pBiFC-NEU1-VC155 and pBiFC-ALK5-VN173 plasmid. Cells were stimulated with or without TGFβ (10 ng/ml, PeproTech) for 24 h after transfection 24 h, and were stained with Hoechst 33342 (C1028, Beyotime) to label nuclear DNA or with Lyso-Tracker (C1046, Beyotime), Mito-Tracker (40740ES50, Yeasen), and CM-Tracker (C1036, Beyotime) to label lysosome, mitochondrion, and cell plasma membrane, respectively. For the control group, cells were transfected with pBiFC-HA-NEU1-VC155 and pBiFC-VN173 plasmids or pBiFC-ALK5-VN173 and pBiFC-VC155 plasmids. Fluorescence signal amplification was observed using the Zeiss LSM800 confocal laser scanning microscopy.

## LC-MS/MS

LC-MS/MS analyses were performed on the Q Exactive HF mass spectrometer coupled with UltiMate 3000 RSLCnano system. Coprecipitates with NEU1 peptides were loaded through auto-sampler and seperated in a C18 analytical column (75 μm × 25 cm, C18, 1.9 μm, 100 Å). Mobile phase A (0.1% formic acid) and mobile phase B (80% ACN, 0.1% formic acid) were used to establish the seperation gradient. The flow rate for separation was 300 nL/min. For DDA mode analysis, each scan cycle is consisted of one full-scan mass spectrum (R = 60 K, AGC = 3e6, max IT = 20 ms, scan range = 350–1800 $m/z$) followed by 20 MS/MS events (R = 15 K, AGC = 2e5, max IT = 50 ms). HCD collision energy was set to 28. Isolation window for precursor selection was set to 1.6 Da. Former target ion exclusion was set for 30 s. MaxQuant (V1.6.6) software was used for data analysis.

## Western blots

Western blots were performed to detect protein expression in kidneys or cells. For denaturing gels, whole-cell lysates were heated (99 °C) for 10 min with SDS-PAGE loading buffer (Beyotime). Proteins were then separated by polyacrylamide gel electrophoresis in acrylamide gels (10%) and transferred using a Bio-Rad western system to polyvinylidene difluoride (EMD Millipore) membranes, which were immediately placed in 5% non-fat milk in Tris-buffered saline (TBS, 50 mM Tris, pH 7.6, 150 mM NaCl)-Tween (0.1% Tween20) buffer or in 1×Carbo-Free Blocking Solution (for SNL detection) for blocking (2 h at 25 °C). Membranes were then washed in TBS-Tween buffer for 5 min, followed by incubation with specific primary antibodies at 4 °C overnight. Membranes were then washed 4 times for 40 min in TBS-Tween buffer, and incubated with a horseradish peroxidase-conjugated anti-mouse antibody, or anti-rabbit antibody at room temperature for 1 h. The resulting immunoblots were visualized using ECL Western Blotting Substrate (KeyGEN), according to the manufacturers' instructions.

For non-denaturing gels, cells were extracted in native extraction buffer (25 mM Tris-HCl pH 7.4, 150 mM NaCl, 1 mM EDTA, 1% NP-40, and 5% glycerol), and protease inhibitor cocktail (Roche)), prior to centrifugation at 12,000 × *g* for 10 min at 4 °C. 25 μg of protein was loaded onto 10% Tris-glycine gels (Beyotime) following the manufacturer's protocol for native gel electrophoresis in Tris–glycine running buffer (25 mM Tris pH ~8.3 and 192 mM glycine), at 120 V for ~2 h. Samples were subjected to electrophoresis at 4 °C to prevent protein denaturation. Proteins were transferred to PVDF membranes then blocked and probed with antibodies (see below).

## Antibodies

The primary antibodies used were as follows: anti-NEU1 (Proteintech, 67032-1-Ig, 1:1000 for IB, 1:100 for PLA), anti-NEU1(Santa Cruz, sc-166824, 1:100 for IP, 1:100 for IF, and 1:100 for IHC), anti-E-cadherin (CST, 3195s, 1:1000 for IB, 1:400 for IF, and 1:400 for ICH), anti-Vimentin (CST, 5741s, 1:1000 for IB, 1:400 for IF, 1:400 for ICH), anti-Snail (CST, 3879s, 1:1000 for IB and 1:400 for ICH), anti-Slug (CST, 9585s, 1:1000 for IB and 1:400 for ICH), anti-GAPDH (CST, 5174s, 1:1000 for IB), anti-α-smooth muscle actin (CST, 19245s, 1:1000 for IB and 1:400 for ICH), anti-HA-tag (Santa, sc-7392, 1:1000 for IB and 1:100 for IP), anti-Flag-tag (Affinity, T0053, 1:1000 for IB and 1:100 for IP), anti-ALK5 (Affinity, AF5347, 1:1000 for IB, 1:300 for IF, and 1:100 for IP, 1:100 for PLA), anti-p-ALK5 (Affinity, AF8080, 1:1000 for IB and 1:100 for IHC), anti-CD68 (CST, 97778, 1:200 for IHC), anti-p-NFκB (CST, 3037S, 1:200 for IHC), anti-Na/K ATPase (Abcam, ab254025, 1:200 for IF), anti-KIM1 (Cloud-Clone Corp, LAA785Mu81, 1:100 for IF), anti-Smad2 (CST, #5339, 1:1000 for IB), anti-Smad3 (CST, #9523, 1:1000 for IB), anti-p-Smad2 (CST, #3108, 1:1000 for IB), anti-p-Smad3 (CST, #9520, 1:1000 for IB), anti-LAMP1 (Abcam, ab24170, 1:200 for IHC), anti-PPCA (Abcam, ab184553, 1:1000 for IB), anti-biotinylated SNL lectin (Vector Labs, B-1305-2, 1:500 for IB).

Anti-rabbit IgG, HRP-linked antibody (CST, 7074, 1:2000 for IB), anti-mouse IgG, HRP-linked antibody (ZSGB-BIO, ZB2305, 1:1000 for IB), Alexa Fluor 488 goat anti-rabbit IgG (Yeasen Biotech, catalog no. 33106ES60, 1:100 for IF), Alexa Fluor 594 goat anti-mouse IgG (Yeasen Biotech, catalog no. 33212ES60, 1:100 for IF), HRP-Streptavidin (Beyotime, A0305, 1:500 for IB). the non-heavy chain IgG secondary antibody (Santa Cruz, m-IgGκ BP-HRP, sc-516102, 1:1000 for IB) and mouse anti-Rabbit IgG HRP (Abmart, #M21006, 1:1000 for IB).

## Surface plasmon resonance (SPR)

A Biacore T200 instrument was used to measure the binding kinetics of human ALK5$_{1-125aa}$, ALK5$_{162-403aa}$, ALK5$_{200-503aa}$ (Sino Biological Inc) and compounds to the NEU1. Measurements were performed at 25 °C. Samples were dissolved in HBS-EP+, pH 7.4 (GE Healthcare). Human NEU1 recombinant protein (gifted by Dr. Xiao Yibei lab) were

immobilized on the sensor chip (CM5, GE Healthcare) using the amine-coupling method according to standard protocols. The recombinant NEU1 proteins coupling solution was injected over the activated chip surface to achieve an immobilization level of 8000–10,000 resonance units (RU) for proteins-NEU1 interaction measurement, and 100–200 Ru for small molecule compound-NEU1 interaction measurement. Human ALK5 recombinant protein were prepared at final concentrations of 500, 250, 125, 62.50, 31.25, 15.60, 7.80, and 0 (blank) nM by dilution into running buffer. The compounds were prepared at indicated concentrations by dilution into running buffer. Data were processed using standard double-referencing and fit to a 1:1 binding model using Biacore T200 Evaluation software to determine the association rate ($K_{on}$, $M^{-1} s^{-1}$), dissociation rate ($K_{off}$, $s^{-1}$), estimate of error (SE), Chi-Square ($Chi^2$), and statistic and maximum response ($R_{max}$; response units (RU)). The equilibrium dissociation constant ($K_d$) was calculated from the relationship $K_D = K_{off}/K_{on}$ (M).

### Statistics

Statistical analysis was performed using the GraphPad Prism software package. Differences among groups were tested by one-way ANOVA or two-way ANOVA, followed by Tukey's test as appropriate. Differences between the two groups were tested using the unpaired $t$-test. Results were expressed as mean ± standard deviation (SD) and as the number (percent) for categorical variables. All tests were two-sided, and $p < 0.05$ was considered statistically significant.

### Reporting summary

Further information on research design is available in the Nature Portfolio Reporting Summary linked to this article.

## Data availability

The authors declare that all data supporting the findings of this study are available within the article and its Supplementary information files. The source data are provided as a Source data file. The publicly available data of human renal transcriptomics used in this study are available in the Gene Expression Omnibus (GEO) database under accession codes GSE66494. The following publicly available Nephroseq database (http://v5.nephroseq.org/) are analyzed in the present study: 'Ju CKD TubInt' dataset and 'Ju CKD Glom' dataset. Source data are provided with this paper.

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

## Acknowledgements

This study is supported in part by the National Natural Science Foundation of China (No. 81930107 and No. 81825023 to L.-W.Q., No. 82174013 to L.Z., and No. 82200292 to Q.-Q.C.), Natural Science Foundation of Jiangsu province (BK20211579 to L.Z. and BK20220466 to Q.-Q.C.), and grants from Young Scholars Fostering Fund of the First Affiliated Hospital of Nanjing Medical University (No. PY2022019 to K.L.). Special thanks go to Dr. Raphael N. Alolga for editing the manuscript. The authors express their appreciation to Wei Jiang in China Pharmaceutical University for assistance in SPR experiments. They thank Yibei Xiao in China Pharmaceutical University for providing us human NEU1 recombinant protein.

## Author contributions

L.Z., L.W.Q., and P.L. designed the experiments. Q.Q.C., J.Y.Z., X.Y.Y., and T.T.W. performed the HK-2 cell experiments. K.L. and G.M. collected and analyzed the human kidney samples. Q.Q.C., N.S., H.M.X., S.J.J., and Y.W. performed the animal studies. Q.Q.C. and P.P.W. performed the protein–protein interaction and protein–compounds interaction experiments. G.M. performed the data statistics analysis. Q.Q.C., L.W.Q., and L.Z. wrote the paper.

## Competing interests

The authors declare no competing interests.
