## [Peer Review File · Nature Communications]

Neuraminidase 1 plays a promotor role in renal fibrosis of male miceREVIEWER COMMENTS

Reviewer #1 (Remarks to the Author):

“Neuraminidase 1 plays a promotor role in renal fibrosis” by Lei Zhang and colleagues. In their work, the authors found that that neuraminidase 1 (NEU1) is significantly upregulated in the fibrotic kidneys of patients and mice model. In contrary, NEU1 overexpression exacerbated progressive renal fibrosis. Mechanistically, NEU1 interacted with TGF β type I receptor ALK5 at the 160-200aa region to activate the ALK5-SMAD2 signaling pathway. Meanwhile, they found that Salvianolic acid B effectively protected mice from renal fibrosis in a NEU1-dependent manner. Overall, the experiment was well designed and partial of the data was graceful. However, several concerns have to be satisfactorily addressed.

Major

1. The authors speculate that NEU1 interacted with TGF β type I receptor ALK5 at the 160-200aa region to activate the ALK5-SMAD2 signaling pathway. As we known, the TGF β /Smad signaling pathway promotes renal fibrosis mainly by phosphorylating R-Smad proteins in the cytoplasm for signaling. However, in recent years, it has been found that Smad2 and Smad3 have different effects when activated by phosphorylation, with Smad3 activation promoting fibrotic lesions and Smad2 activation inhibiting fibrotic lesions. In this paper, only phosphorylated Smad2 was detected, what is the status of Smad3 activation? What is the effect of intervention NEU1 and ALK5 binding on Smad3 activation?

2. Upon TGF β binding, the type II receptor kinase transphosphorylates the GS domain of type I receptor, resulting in the activation of type I receptor kinases. Phosphorylation of the GS domain is a critical event in signal transduction by the serine-threonine kinase receptors. Upon phosphorylation of the GS domain by T β R-II, the T β R-I kinase is converted to an active conformation. The activated T β R-I then phosphorylate the intracellular R-Smads. Therefore, what is the state of the NEU1-bound ALK5 protein, phosphorylated? Non-phosphorylated? After NEU1 binds to ALK, does it promote phosphorylation activation of ALK? Or promote phosphorylation of R-Smad protein by activated ALK? As author mentioned, the interaction between NEU1 and ALK5 was enhanced in response to TGF β stimulation in HK-2 cells (Fig. 6f, g). Is NEU1 promotion of ALK activation dependent on the presence of TGF β ?

Minor

1. Kim-1 is a marker of renal tubular cell injury, but it is more commonly used to assess tubular injury in acute kidney injury, and more clinical and basic studies have found that it is not a sensitive indicator of the severity of disease in chronic kidney disease. Why choose this index for the evaluation of chronic tubular injury?

2. Many of the immunohistochemical stained images in the text are too small and not high resolution and do not clearly show positive staining, e.g. fig.2C, fig.3b, fig.4g, fig.8b, fig.8i. It is recommended to enlarge the images and mark the positive staining sites with arrows. Similarly, pathological histological images are too small to see specific tissue structural changes, so please replace them.

3. Replace Supplementary Fig.5a GAPDH.

4. In Fig.2, it is suggested to detect fibrosis indexes in UUO and FA-induced kidney injury models simultaneously, and evaluate and analyze them comprehensively with NEU1 expression changes. Also combining the results of fig.2 and Supplementary fig.2 experiments would better illustrate the relationship between NEU1 expression changes and renal disease in both in vivo and in vitro models.

5. Fig.3f, how to analyze and count tubulointerstitial inflammation index? Not described in the methods.

6. Fig. 3b and Supplementary Fig. 6e, no positive staining for CD68 and p-NK κ B was observed in kidney tissue sections.

7. Comparing Fig. 6h, i and Fig. 7p, why the p-smad2 activation was significantly increased in the sham group in Fig. 7p for the same sham group.

8. In Fig. 8b, f, there is no difference in the distribution and expression of Snail in each group of kidney tissues

Reviewer #2 (Remarks to the Author):

The manuscript by Qian-Qian Chen et al. "Neuraminidase 1 plays a promotor role in renal fibrosis" describes a novel role for mammalian neuraminidase 1 (lysosomal neuraminidase, NEU1) in renal fibrosis. The authors demonstrate increased NEU1 expression in cohorts of human patients with renal fibrosis. They further show that NEU1 levels are significantly elevated in tubular epithelial cells (TEC) of fibrotic kidneys in human patients and mouse models. The authors also show that depletion of NEU1 in conditional TEC-specific NEU1 KO mice and epithelial cellular models reduces severity of fibrotic changes, while overexpression of NEU1 has an opposite effect. Finally they demonstrate that NEU1 interacts with the major TGF β receptor of TEC, ALK5, and activates ALK5-SMAD2 signaling pathway, potentially leading to development of fibrosis. Finally, they show that salvianolic acid B binds with high affinity to recombinant human NEU1 protein and effectively prevents renal injury in mouse unilateral ureteral obstruction (UUO) model. These are important and novel findings potentially leading to development of novel therapeutic approaches to treat fibrosis and chronic kidney disease.

At the same time, the study falls short of clarifying the mechanism by which NEU1 induces fibrosis. The authors propose that NEU1 physically interacts with Transforming Growth Factor β Receptor ALK5 and induces its activation and downstream TGF-beta signalling. They do not demonstrate, however, how exactly NEU1 does this. NEU1 acts in the cells as an enzyme that removes terminal sialic acid residues from glycan chains of glycoproteins. Still, the authors did not attempt to explore whether ALK5 is sialylated and whether it is a substrate of NEU1. Besides, the authors did not consider alternative hypotheses explaining a positive action of NEU1 on fibrosis, such as induction of pro-inflammatory cytokines and recruitment/activation of leucocytes and other immune cells. This action of NEU1 has been convincingly demonstrated in the current literature, but are not explored by the authors in the current study.

Other specific problems are listed below.

Figure 1d. This image could be improved if the authors use adjacent kidney sections for the three types of staining (NEU1, Masson, HE). As presented the selected sections are very different, which prevents the readers from appreciating a correlation between NEU1 increase and renal fibrosis.

Supplementary Fig. 2g, h. The authors have used the plasmid encoding for HA-tagged NEU1 to overexpress the enzyme in HK-2 cells. Previous studies, however, demonstrated that in order to produce active NEU1, the protein must be co-expressed with its protective protein, PPCA (PMID: 9501080, PMID: 9480870). If overexpressed without PPCA, NEU1 is not targeted to lysosomes, but is retained in the ER where it forms inactive inclusions (PMID: 9501080). The authors should demonstrate the increase of NEU1 enzymatic activity in transfected cells. They also need to demonstrate reduced NEU1 activity in the kidney tissue of NEU1 CKO mice (Sup. Fig. 3) and increased NEU1 activity in the kidney of mice injected with AAV9-NEU1 (Figure 5 and Sup. Fig. 6).

Importantly, NEU1 KO mice and sialidosis patients develop a kidney disease caused by accumulation of lysosomal storage materials in podocytes (PMID: 12023988). Did the authors observe any pathological changes in the kidney of their Neu1 CKO? Pathological examination of kidney tissues using electron microscopy and immunohistochemical assays specific for lysosomal storage

phenotype need to be included in the paper.

Figure 1a, Sup. Fig. 6. CD68 and PNF-kB staining is barely visible. The images need to be improved, or immunofluorescence should be used instead.

Sup. Fig. 4. It seems that the expression of pro-inflammatory and fibrosis-associated genes (panel c) was analysed in Sham Ctr, Ctr-UJO and Neu1 CKO-UJO mice but not in Sham Neu1 CKO mice? Why?

Figure 6. Experiments conducted by the authors demonstrate that NEU1 makes a stable complex with ALK5 in solution, but they do not show if these two proteins interact in the live cell. This could be demonstrated with experimental approaches similar to BioID or BRET. The immunofluorescent images provided by the authors are taken with small magnification and show just a general co-expression of two proteins in tissues but not their co-localization in specific subcellular structures. Even more surprising are the results showing that NEU1 presumably binds to the cytoplasmic domain (amino acid 162-403) of ALK5. NEU1 in cells is located either on the inner side of the lysosomal membrane or on the extracellular side of the plasma membrane, so its presence and its interaction with ALK5 in the cytoplasm are unlikely. Besides, if produced without PPCA, NEU1 tends to self-aggregate (PMID: 35014832; PMID: 19666471). This could potentially affect the results.

Page 11, line 215 and Figure 6j. SPR binding measurement of NEU1 and ALK5 indicates " $K_d = 37.4$ nM". This measurement should include an estimate of error and the method by which it was determined, statistical analysis should also be reported. The SPR curves, as labelled, indicate that the 250 nM concentration of the analyte gives higher response than the 500 nM concentration. Is this accurate? Can the fit curves be provided in the supporting information? The protocol (pages 27-28) lists analyte concentration as ng/mL, while the figure shows it in nM.

Page 21, line 396 What is the source and purity of the salvianolic acid A and B and rosmarinic acid samples?

Page 12, Lines 227 – 233 and Figure 7a. SPR binding measurements of small molecules to NEU1. The values given in Fig 7A are in " $\log_{10}(K_d)$ " and range from -2 to +3; What are the units of K_d here (mM? nM?) Again, the measured K_d values should include an estimate of the error and be reported with appropriate estimations of significance. Comparison of the results in Fig 6j and Fig 7b & 7c shows that the former involves the binding of a large peptide (241 residues, ~26 kDa), which gives a maximum RU of ~100 RU (at 500 ng/mL); while small molecules, Sal A and Sal B (MW < 500 Da) give maximum RU of up to 80 RU (at 20 ng/mL). This probably implies that many more than one equivalent of the compound is binding to the immobilized protein. Is this a specific interaction, and were any negative control proteins used to confirm this interaction? Does denatured NEU1 give the same response?

Figure 7. The activity of the small molecule compounds used here, salvianolic acids A and B (Sal A, Sal B) and rosmarinic acid, seems likely to be due to non-specific interactions with multiple proteins. The authors will need to provide more data before they can conclude that these compounds specifically target NEU1 and are not acting through other mechanisms/targets. Sal A and Sal B are known to inhibit a large number of enzymes, suggesting that they are pan-assay interference compounds (PAINS; see Baell and Holloway, J Med Chem, 2010 and more recent reviews). Their structures all include a catechol (catechol A) group which could account for these effects. While it is possible these compounds interact with the intended target, there should be an evidence presented that this is a specific effect, not attributable to the wide range of previously reported effects/interactions of these compounds with targets besides NEU1. Sal A and Sal B are known as ROS scavengers, and specific activities of these compounds have been reported to include: amyloid beta aggregation, alpha synuclein aggregation, MMP-9 enzyme inhibition, heme oxygenase-1 enzyme inhibition, cyclooxygenase-2 enzyme inhibition, inhibition of platelet adhesion, inhibition of PI3K, and many others. Rosmarinic acid is reported to inhibit carbonic anhydrase, complement C3,

amylase, MARK4, and many others (see for example: 10.1155/2019/3281260; 10.1186/s40035-019-0159-7; 10.1124/jpet.111.190736; 10.1186/1471-2210-10-10; 10.1002/ijc.24160; 10.1016/j.thromres.2008.05.020 10.1111/j.1538-7836.2010.03859.x; 10.3906/kim-1403-5; 10.1016/0192-0561(88)90026-4; 10.1038/s41598-020-65648-z).

Do authors propose that Sal B inhibits catalytic activity of NEU1? Do they propose that it reduces the protein levels or localization of NEU1 in the cell or prevents its interaction with ALK5? Experiments answering these questions need to be conducted. Also, strangely, Sal B reduces the fibrosis area to 2% (Fig. 7b) and also reduces levels of fibrotic and pro-inflammatory genes (Fig. 7 h-o). This effect exceeds the improvement observed in the NEU1 conditional KO (Neu1 CKO, Figure 3). At the same time Sal B-treated mice with the genetic depletion of NEU1 (Fig. 8) in the same model show higher levels of fibrosis and inflammation. From my point of view, this is not fully compatible with the hypothesis of the authors that the improvement of fibrosis by Sal B treatment is mediated by its action on NEU1.

Minor comments

Abstract.

“the functions of mammalian neuraminidases remain largely unexplored.” This is an overstatement. Many biological functions of mammalian sialidase have been described. Remove or rephrase.

Page 4, lines 63-66. “First, mammalian neuraminidases, typically located in the lysosome, are responsible for the initial step of degradation of glycoconjugates by removing sialic acids¹⁶. Neuraminidases deficiency leads to sialidosis, a disease characterized by tissue accumulation of sialo-glycopeptides and sialo-oligosaccharides¹⁷.” As it is written, this statement is not correct. Of the 4 mammalian neuraminidases (NEU1-NEU4) only two, NEU1 and NEU4, are found in lysosomes. Besides, sialidosis is caused specifically by genetic deficiency of NEU1.

Page 4, lines 73 and below “On account of their various functions, mammalian neuraminidases have been shown to play an emerging role in several human diseases, including cardiovascular diseases, cancers, neurodegenerative disorders, and lung diseases²².”

PUBMED search reveals a number of recent publications on the roles of NEU1 in autoimmune, cardiovascular and lung diseases and the use of specific inhibitors of this enzyme to block these pathways. I encourage the authors to include these references instead of the review they cite.

Discussion

Line 297. “We and others have demonstrated the contribution of NEU1 in cardiovascular diseases^{20, 21, 30}.” The list of references is incomplete.

Line 311. “Emerging evidence has demonstrated that NEU1 can be sorted to the cell surface to desialylate membrane receptors such as TLR4 and insulin receptors^{19, 37}.” Earlier publications showing that NEU1 activates TLR4 receptors exist.

Line 306. “In line with this, there is no case report on sialidosis syndrome caused by use of neuraminidase inhibitors such as oseltamivir for up to 6 weeks³⁶.” Oseltamivir does not show any inhibitory activity for NEU1. It is specific for the influenza neuraminidase.

Reviewer #3 (Remarks to the Author):

In the present study, the authors found that NEU1, a mammalian neuraminidase, promotes renal fibrosis, using loss-of-function as well as gain-of function experiments in two mouse models of renal fibrosis (UUO and folic acid-induced models). As a possible mechanism to explain this phenomenon,

the authors found that NEU1 enhances the TGF-beta signaling pathway through interacting with TGF-beta type I receptor. This is also a novel finding. Furthermore, based on binding to NEU1, the authors identified two compounds that are effective on renal injury in mouse models. However, mechanistic studies how NEU1 enhances TGF-beta signaling are not of satisfactory quality, though the authors extensively performed animal experiments and chemical biological study. Specific points to improve this manuscript are listed below.

Major points

- 1) Figure 6 and thereafter: The authors used an anti-pSmad2 antibody (AF3450, Ser250, not Ser205 as described in this manuscript) to evaluate Smad2 activation. This is not appropriate, because Smad2 protein is activated by ALK5-mediated C-terminal phosphorylation (Ser 465/467). Because the authors stated that NEU1 associates with ALK5 to enhance the downstream signaling, the C-terminal phosphorylation of Smad2 should be examined.
- 2) Figure 6c: This immunofluorescence image simply indicates that NEU1 and ALK5 are expressed in the same cells in a fibrotic kidney and does not necessarily indicate that they are colocalized in kidney cells. NEU1 is known to be usually localized in the lysosome. Therefore, possible interaction between NEU1 and ALK5 should be robustly demonstrated. The authors should consider detection of their colocalization using in situ PLA assay.
- 3) Figure 6h and i: These data importantly reveal that the ALK5 expression level was decreased by NEU1 KO and increased by NEU1 overexpression. It appears likely that NEU1 interacts with and stabilize ALK5, to enhance TGFb signaling. I recommend the authors to examine this possibility using HK-2 cells, examining the effects of NEU1 on ALK5 stability after TGFb stimulation (ALK5 is downregulated after TGFb stimulation by the action of a negative feedback loop involving Smad7 and Smurf2, PMID22378783). In addition, Smad phosphorylation usually declines within 24 h after stimulation. It would be nice to monitor time course of Smad C-terminal phosphorylation after TGFb stimulation in the presence and absence of NEU1. NEU1 may sustain Smad phosphorylation.
- 4) Using HK-2 cells, effects of SA-A and SA-B on ALK5 expression, pALK5, pSmad2 (C-terminal phosphorylation) after TGFb stimulation should be examined. In addition, examine if these compounds affect NEU1-ALK5 interaction. These data would significantly strengthen this paper.
- 5) In the abstract, the authors stated "Neu1 interacted with TGFb type I receptor ALK5 at the 160-200aa region to activate ALK5-SMAD2 signaling pathway". However, knockdown or overexpression of NEU1 did not affect TGFb target gene expression in the absence of TGFb ligand (Supplementary Figure 2e and i). Therefore, "to activate ALK5-SMAD2 signaling pathway" is misleading. Instead, "to enhance ALK5-SMAD2 signaling pathway" would be appropriate. Same to lines 202, 269, and 339.
- 6) Is NEU1 enzymatic activity required for enhancing the TGF-beta pathway?

Minor points

- 7) Figure 1d: The authors should have used serial sections. Otherwise, the data are not convincing enough.
- 8) Line 119, "NEU1 mediated TGFb-induced HK-2 cell injury": I am afraid that the authors did not examine "HK-2 cell injury". In addition, induction of KIM1 by NEU1 overexpression was attenuated by ALK5 knockdown (Supplementary Fig. 7d), indicating that ALK5 mediates NEU1-induced cell response(s). Reconsider this description.
- 9) Line 202, "As expected, the interaction between NEU1 and ALK5 was enhanced in response to TGFb stimulation in HK-2 cells": Why did the authors "expect" that the interaction would be enhanced by TGFb stimulation? Is this an important event for enhancement of TGFb signaling by NEU1?
- 10) In Supplementary Fig. 2, the authors observed that siNEU1 attenuates TGFb-induced gene expression (24 h stimulation). Is this a general effect on TGFb signaling or a limited effect on EMT-related genes? This point should be made clear.
- 11) Figure 1j: Data on NEU1 and serum creatine levels, although described in the legend, are missing. Current Fig. 1j should be Fig. 1k (mis-labeled).
- 12) Anti-KIM1 antibody is not described in "Methods".
- 13) Line 147; knockdown >> knockout //

Responses to the Reviewers' Comments

We deeply appreciate the comments from Reviewer 1# on our manuscript NCOMMS-22-29974. The following are our responses.

Reviewer #1 (Remarks to the Author):

General Comments: “Neuraminidase 1 plays a promotor role in renal fibrosis” by Lei Zhang and colleagues. In their work, the authors found that that neuraminidase 1 (NEU1) is significantly upregulated in the fibrotic kidneys of patients and mice model. In contrary, NEU1 overexpression exacerbated progressive renal fibrosis. Mechanistically, NEU1 interacted with TGF β type I receptor ALK5 at the 160-200aa region to activate the ALK5-SMAD2 signaling pathway. Meanwhile, they found that Salvianolic acid B effectively protected mice from renal fibrosis in a NEU1-dependent manner. Overall, the experiment was well designed and partial of the data was graceful. However, several concerns have to be satisfactorily addressed.

Response: We thank the reviewer for the positive comments.

Major Comment 1: The authors speculate that NEU1 interacted with TGF β type I receptor ALK5 at the 160-200aa region to activate the ALK5-SMAD2 signaling pathway. As we known, the TGF β /Smad signaling pathway promotes renal fibrosis mainly by phosphorylating R- Smad proteins in the cytoplasm for signaling. However, in recent years, it has been found that Smad2 and Smad3 have different effects when activated by phosphorylation, with Smad3 activation promoting fibrotic lesions and Smad2 activation inhibiting fibrotic lesions. In this paper, only phosphorylated Smad2 was detected, what is the status of Smad3 activation? What is the effect of intervention NEU1 and ALK5 binding on Smad3 activation?

Response: Based on the reviewer's suggestion, we measured the phosphorylation of Smad3 in TGF β -stimulated HK-2 cells transfected with NEU1 siRNA or NEU1 full-length plasmid. We observed that NEU1 knockdown suppressed both Smad2 (Ser465/467) phosphorylation and Smad3 (Ser423/425) phosphorylation in response to TGF β stimulation (Fig. 7c, d). Conversely, NEU1 overexpression promoted TGF β -induced phosphorylations of Smad2 (Ser465/467) and Smad3 (Ser423/425) (Fig. 7e, f).

As a further characterization, we detected phosphorylations of Smad2 and Smad3 in *Neu1*CKO mice, NEU1-overexpressed mice, as well as salvianolic acid B-treated mice subjected to UUO surgery. In the mouse kidney, the UUO-stimulated phosphorylations of Smad2 (Ser465/467) and Smad3 (Ser423/425) were significantly inhibited upon NEU1 knockout (Supplementary Fig. 8e, f). On the contrary, NEU1 overexpression augmented UUO-induced activation of Smad2 (Ser465/467) and Smad3 (Ser423/425) (Supplementary Fig. 8g, h). Salvianolic acid B treatment markedly inhibited the UUO-induced activation of Smad2 (Ser465/467) and Smad3 (Ser423/425) (Supplementary Fig. 10c, d).

Truncation (Supplementary Fig. 7c-e) and co-immunoprecipitation (Co-IP) (Fig. 6l) experiments have shown that the 160-200 amino acid region of ALK5 is the NEU1-binding domain. PLA (Fig. 6m) and Co-IP (Fig. 6n) assays confirmed that 160-200 domain plasmid competed with ALK5 and inhibited NEU1-ALK5 binding. To investigate the effect of NEU1-ALK5 binding on Smad3 activation, we transfected the cells with ALK5₁₆₀₋₂₀₀ plasmid. As expected, ALK5₁₆₀₋₂₀₀ domain reduced phosphorylations of Smad2 (Ser465/467) and Smad3 (Ser423/425) (Fig. 7g, h). As a result, NEU1 bound to ALK5 and promoted its phosphorylation activation.

Taken together, these results indicated that both Smad2 and Smad3 are activated in NEU1-involved renal fibrosis in TGF β -stimulated HK-2 cells and UUO-induced mouse model. Intervening the binding of NEU1 and ALK5 significantly inhibited Smad2 and Smad3 activation. We have added the relevant information in the Revised Manuscript (Lines 247-254, Page 13 and Lines 274-275, Page 14).

Fig.7

Fig. 7c-f Western blots (**c, e**) and quantitative results (**d, f**) of p-ALK5, ALK5, p-Smad2, Smad2, p-Smad3, and Smad3 in HK-2 cells transfected with siNEU1 or NEU1 plasmid for 24 h and then treated with TGFβ (10 ng/ml) for 24 h. $n = 3$ samples per group. Relative protein levels were shown after normalization to GAPDH. Data were presented as the mean \pm SD. One-way ANOVA.

Supplementary Fig. 8

Supplementary Fig. 8 e-h, Western blots and quantitation of p-Smad2 (Ser465/467), and p-Smad3 (Ser423/425) protein in kidney from *Neu1* CKO (**e, f**) or NEU1 overexpressed (**g, h**) mice subjected to UUO for 10 days. Data were presented as the mean \pm SD. $n = 3$ samples per group. One-way ANOVA.

Supplementary Fig. 10

Supplementary Fig. 10 c, d Representative western blots (c) and quantitative results (d) of p-ALK5(Ser165), p-Smad2(Ser465/467), Smad2, p-Smad3 (Ser423/425), and Smad3 in HK-2 cells treated with Salvianolic acid A (SaA, 10 μ M) and Salvianolic acid B (SaB, 10 μ M). Relative protein levels were shown after normalization to GAPDH. $n = 3$ mice per group. Data were presented as mean \pm SD. One-way ANOVA.

Fig.7

Fig. 7 g, h Western blots (g) and quantitative results (h) of p-ALK5(Ser165), p-Smad2(Ser465/467), Smad2, p-Smad3 (Ser423/425), and Smad3 in HK-2 cells transfected with NEU1 and ALK5₁₆₀₋₂₀₀ plasmid for 24 h and treated with TGF β (10 ng/ml) for 24 h. $n = 3$ samples per group. Relative protein levels were shown after normalization to GAPDH. One-way ANOVA.

Major Comment 2: Upon TGF β binding, the type II receptor kinase transphosphorylates the GS domain of type I receptor, resulting in the activation of type I receptor kinases. Phosphorylation of the GS domain is a critical event in signal transduction by the serine-threonine kinase receptors. Upon phosphorylation of the GS domain by T β R-II, the T β R-I kinase is converted to an active conformation. The activated T β R-I then phosphorylate the intracellular R-Smads. Therefore, what is the state of the NEU1-bound ALK5 protein, phosphorylated? Non-phosphorylated? After NEU1 binds to ALK, does it promote phosphorylation activation of ALK? Or promote phosphorylation of R-Smad protein by activated ALK? As author mentioned, the interaction between NEU1 and ALK5 was enhanced in response to TGF β stimulation in HK-2 cells (Fig. 6f, g). Is NEU1 promotion of ALK activation dependent on the presence of TGF β ?

Response: Based on the reviewer's comments, we added phosphatase to cell lysis buffer to remove phosphate groups of proteins. The co-immunoprecipitation results showed that phosphatase significantly inhibited the interaction between NEU1 and ALK5 (Fig. 7i), suggesting that their interaction partly depended on the

phosphorylation state of ALK5. We have added the relevant information in the Revised Manuscript (Lines 255-257, Page 13).

Truncation (Supplementary Fig. 7c-e, Fig.6k) and Co-IP (Fig. 6l) experiments have shown that 160-200 amino acid region of ALK5 is the NEU1-binding domain. PLA (Fig. 6m) and Co-IP (Fig. 6n) assays confirmed that 160-200 domain plasmid competed with ALK5 and inhibited NEU1-ALK5 binding. To investigate the effect of NEU1-ALK5 binding on Smad3 activation, we transfected the cells with ALK5₁₆₀₋₂₀₀ plasmid. As expected, ALK5₁₆₀₋₂₀₀ domain reduced phosphorylations of Smad2 (Ser465/467) and Smad3 (Ser423/425) (Fig. 7g, h). As a result, NEU1 bound to ALK5 and promoted its phosphorylation activation.

We then tested the effects of NEU1 overexpression on ALK5-Smad2/3 pathways in the absence or presence of TGF β stimulation. The results showed that single NEU1 overexpression was insufficient to promote the phosphorylations of ALK5 (Ser165) and Smad2 (Ser465/467). Under the condition of TGF β stimulation, NEU1 overexpression significantly aggravated ALK5 (Ser165), Smad2 (Ser465/467) and Smad3 (Ser423/425) activation (Fig.7e, f). Consequently, NEU1 promoted the activation of ALK5 and the downstream signaling pathway in a TGF β -dependent manner. We have added the relevant information in the Revised Manuscript (Lines 247-254, Page 13).

We employed loss of function to study the effects of ALK5 on NEU1-involved Smad2/3 activation. ALK5 knockdown (Fig.7j) effectively blocked the NEU1-mediated phosphorylation of Smad2 (Ser465/467) and Smad3 (Ser423/425) in response to TGF β stimulation (Fig.7l, m). We have added the relevant information in the Revised Manuscript (Lines 257-260, Page 13).

Fig. 7 i Co-immunoprecipitation of NEU1 and ALK5 in HK-2 cells stimulated with TGFβ (10 ng/ml) for 24 h. λpp, Lambda Protein Phosphatase. Two independent experiments were performed.

Supplementary Fig. 7

Supplementary Fig. 7 c Schematic diagram of NEU1 binding with ALK5. **d** The interaction between NEU1 and ALK5_{1-125aa} was tested by SPR. **e** The interaction between NEU1 and ALK5_{200-503aa} was tested by SPR.

Fig. 6k

Fig. 6 k The interaction between NEU1 and ALK5_{162-403aa} tested by SPR. The frequency response and fitting curves were displayed.

Fig. 6

Fig. 6 l Co-immunoprecipitation of NEU1 and ALK5_{160-200aa} in HEK293T cells. **m** Interaction between NEU1 and ALK5 (NEU1-ALK5, red) was analyzed by PLA in HK-2 cells transfected with ALK5_{160-200aa} plasmid. Scale bars, 10 μm. **n** Co-immunoprecipitation of NEU1 and ALK5 in HEK293T cells transfected with ALK5_{160-200aa} plasmid.

Fig.7

Fig. 7 g, h Western blots (**g**) and quantitative results (**h**) of p-ALK5(Ser165), p-Smad2(Ser465/467), Smad2, p-Smad3 (Ser423/425), and Smad3 in HK-2 cells transfected with NEU1 and ALK5₁₆₀₋₂₀₀ plasmid for 24 h and treated with TGFβ (10 ng/ml) for 24 h. $n = 3$ samples per group. Relative protein levels were shown after normalization to GAPDH. One-way ANOVA.

Fig.7

Fig. 7 e, f Western blots (**e**) and quantitative results (**f**) of p-ALK5(Ser165), p-Smad2(Ser465/467), Smad2, p-Smad3 (Ser423/425), and Smad3 in HK-2 cells transfected with NEU1 plasmid for 24 h and treated with TGFβ (10 ng/ml) for 24 h. $n = 3$ samples per group. Relative protein levels were shown after normalization to GAPDH. Data were presented as the mean \pm SD. Two-way ANOVA.

Fig.7

Fig. 7 j Western blots of ALK5 in HK-2 cells. siALK5-1, siALK5-2, and siALK5-3 are 3 different siRNA sequences. HK-2 cells were transduced with siALK5 for 48 h. Two independent experiments were performed. **l, m** The protein level of p-ALK5(Ser165), p-Smad2(Ser465/467), Smad2, p-Smad3 (Ser423/425), and Smad3 in HK-2 cells transduced with NEU1 full-length plasmid and ALK5 siRNA for 48 h. *KIMI* mRNA normalized to *GAPDH*. Relative protein levels were shown after normalization to GAPDH. $n = 3$ samples per group. Data were presented as the mean \pm SD. Two-way ANOVA.

Minor Comment 1: Kim-1 is a marker of renal tubular cell injury, but it is more commonly used to assess tubular injury in acute kidney injury, and more clinical and basic studies have found that it is not a sensitive indicator of the severity of disease in chronic kidney disease. Why choose this index for the evaluation of chronic tubular injury?

Response: We agree with the reviewer that KIM1 is commonly used to assess tubular injury in acute kidney injury, since it appears more sensitive than other markers. KIM1 is localized on the apical surface of surviving proximal tubule epithelial cells and is almost undetectable in healthy kidneys (*J Biol Chem.* 1998; 273(7): 4135-4142). When the kidneys are stimulated by ureteral ligation or other insult, KIM1 is strongly upregulated within 2 days and falls thereafter, but remains at a relatively higher level than the healthy kidneys during 14-day observation (*J Clin Invest.* 2013; 123(9): 4023-4035). Kuehn EW et al also found that KIM1 is expressed in CKD and its level correlates directly with interstitial fibrosis (*Am J Physiol Renal Physiol*, 2002, 283(6): 1326-1336). Benjamin D et al established conditional expression of KIM1 in renal epithelial cells (*Kim1^{RECG}*) and found that sustained KIM1 expression in the absence of an injury stimulus promoted kidney fibrosis, providing a link between acute and recurrent injury with progressive chronic kidney disease (*J Clin Invest.* 2013; 123(9): 4023-4035). These results suggest that KIM1 can be used as an indicator of tubular epithelial injury in CKD.

More importantly, we also employed other indices for evaluation of chronic kidney injury, including renal tubule morphology by HE staining, EMT-associated genes transcription and protein expression, inflammatory cytokines, chemokines, and fibrogenic factors. We have added the relevant information in the Revised Manuscript (Line 111, Page 6).

Minor Comment 2: Many of the immunohistochemical stained images in the text are too small and not high resolution and do not clearly show positive staining, e.g. fig.2C, fig.3b, fig.4g, fig.8b, fig.8i. It is recommended to enlarge the images and mark the positive staining sites with arrows. Similarly, pathological histological images are too small to see specific tissue structural changes, so please replace them.

Response: We thank the reviewer for this suggestion. We have provided the high-resolution images of all the immunohistochemical staining and pathological histological images. At the same time, we enlarged the images and marked the positive staining sites with arrows. Please see Fig. 2c, Fig. 3b, Fig. 4g, Fig. 9b, and Fig. 9i of the Revised Manuscript.

Fig. 2c

Fig. 2 c Immunohistochemistry of kidney sections from mice after UUO surgery for 10 days or folic acid injection for 4 weeks. $n = 3$ mice. Scale bar, 20 μm . Data were presented as mean \pm SD.

Fig.3b

Fig. 3 b The gross appearance of kidneys (Scale bar, 1 mm), Hematoxylin and eosin (HE, Scale bar, 20 μ m), Masson's trichrome staining (Scale bar, 20 μ m), immunohistochemistry staining of CD68, and p-NF κ B (Scale bar, 50 μ m) from control (Ctrl) and *Neu1* CKO mice 10 days after UUO. The red arrow indicates positive cells. $n = 3$ mice per group.

Fig.4g

Fig. 4 g Hematoxylin and eosin (HE) (scale bar, 1 mm), Masson's trichrome staining (scale bar, 50 μ m), and immunohistochemistry staining of CD68 and p-NF κ B (scale bar, 50 μ m) in kidney sections from Ctrl and *Neu1* CKO mice 28 days after folic acid administration. $n = 3$ mice per group. The red arrow indicates positive cells.

Fig.9b

Fig. 9 b Representative gross appearance of kidneys (Scale bar, 2 mm), kidney cross-sections stained with HE (Scale bar, 50 μm), Masson's trichrome (Scale bar, 50 μm), immunohistochemical staining with E-Cadherin and Snail (Scale bar, 50 μm). $n = 3$ mice per group. The red arrow indicates positive area.

Fig.9i

Fig. 9 i Representative image of immunohistochemical staining with p-ALK5 (Ser165) (Scale bar, 50 μ m). $n = 3$ mice per group.

Minor Comment 3: Replace Supplementary Fig.5a GAPDH.

Response: We repeated this experiment and replaced Supplementary Fig. 5a in the Revised Manuscript.

Supplementary Fig. 5a

Supplementary Fig. 5a Western blots of E-Cadherin, Vimentin, α -SMA, Snail, and Slug in kidneys of control (Ctrl) and *Neu1* CKO mice 28 days after folic acid (250 mg/kg) injection. Relative protein levels were shown after normalization to GAPDH.

Minor Comment 4: In Fig.2, it is suggested to detect fibrosis indexes in UUO and FA-induced kidney injury models simultaneously, and evaluate and analyze them comprehensively with NEU1 expression changes. Also combining the results of fig.2 and Supplementary fig.2 experiments would better illustrate the relationship between NEU1 expression changes and renal disease in both in vivo and in vitro models.

Response: Based on the reviewer’s suggestion, we measured the fibrosis indices in the UUO and FA-induced kidney injury models, and then correlated them with NEU1 expression. The results showed that the mRNA expressions of *Acta2*, *Vim*, *Coll1a1*, *Col3a1*, and *Tgfb* greatly increased in fibrotic kidneys compared with control group (Fig. 2h, i). Of note, the level of NEU1 showed a positive correlation with these fibrosis indices (Fig. 2j-s). We have added the relevant information in the Revised Manuscript (Lines 122-125, Page 7).

Fig.2

Fig. 2 h, i *Neu1*, *Acta2*, *Vim*, *Coll1a1*, *Col3a1*, and *Tgfb* mRNA level in kidney from mice after UUO surgery for 10 days (g) or folic acid injection for 4 weeks (h). $n = 12$ mice per group. Unpaired two-tailed t -test. **j-s** Pearson’s correlation of NEU1 with *Acta2*, *Vim*, *Coll1a1*, *Col3a1*, and *Tgfb* mRNA. $n = 12$. Pearson χ^2 test.

Minor Comment 5: Fig.3f, how to analyze and count tubulointerstitial inflammation index? Not described in the methods.

Response: Tubulointerstitial inflammation on HE stained section was analyzed and counted by renal pathological score. We have added the relevant description in the Methods part of the Revised Manuscript (Lines 466-473, Page 25).

Minor Comment 6: Fig.3b and Supplementary Fig.6e, no positive staining for CD68 and p-NKκB was observed in kidney tissue sections.

Response: We have provided the high-resolution images and marked the positive staining sites with arrows. Please see Figure 3b and Supplementary Figure 6f, g of the Revised Manuscript.

Fig.3b

Fig. 3 b The gross appearance of kidneys (Scale bar, 1 mm), Hematoxylin and eosin (HE, Scale bar, 20 μm), Masson's trichrome staining (Scale bar, 20 μm), immunohistochemistry staining of CD68, and p-NFκB (Scale bar, 50 μm) from control (Ctrl) and *Neu1* CKO mice 10 days after UUO. The red arrow indicates positive cells. $n = 3$ mice per group.

Supplementary Fig. 6 f, g Immunohistochemistry staining of CD68 or p-NFκB in kidney sections (f, left; g, left) and quantitative results (f, right; g, right) from AAV9-Ctrl and AAV9-NEU1 mice 10 days after UUO. $n = 3$ mice per group. Data are presented as mean \pm SD. Unpaired two-tailed t -test.

Minor Comment 7: Comparing Fig.6h, i and Fig.7p, why the p-smad2 activation was significantly increased in the sham group in Fig.7p for the same sham group.

Response: As rightly observed by the reviewer, the protein expression of p-Smad2 differed in the same sham group among different assays. Experimental inconsistency is a common issue in biological assays, especially using animal tissues. Individual heterogeneity of mice, sample treatment, instability of target proteins, and other unknown factors probably contributed to this experimental inconsistency. We repeated determination of p-Smad2 in all groups, and achieved a good consistency between experimental groups (Supplementary Fig. 8e-h and Fig. 8p). We have added the relevant information in the Revised Manuscript (Lines 249-253, Page 13 and Lines 290, Page 15).

Supplementary Fig. 8

Supplementary Fig. 8 e-h, Western blots and quantitation of p-Smad2(Ser465/467), and p-Smad3 (Ser423/425) protein in kidney from *Neu1* CKO (e, f) or NEU1 overexpressed (g, h) mice subjected to UUO for 10 days. Data were presented as the mean ± SD. n = 3 samples per group. One-way ANOVA.

Fig. 8p

Fig. 8 p, Western blots (p, left) and quantitative results (p, right) of p-Smad2 (Ser465/467), Smad2, p-Smad3 (Ser423/425), and Smad3 in kidney from control (Ctrl) and SaA- or SaB-treated mice 10 days after UUO. n = 3 mice per group. All statistic data were presented as mean ± SD, one-way ANOVA.

Minor Comment 8: In Fig.8b, f, there is no difference in the distribution and expression of Snail in each group of kidney tissues.

Response: We have provided the high-resolution images and marked the positive staining sites with arrows. Please see Fig. 9b and Fig. 9f of the Revised Manuscript.

Fig.9 b Representative immunohistochemical staining with Snail (Scale bar, 50 μ m). $n = 3$ mice per group. The red arrow indicates positive area.

Fig.9 f Quantification of staining of Snail1. $n = 3$ mice per group. Data were presented as mean \pm SD, one-way ANOVA.

We deeply appreciate the comments from Reviewer 2# on our manuscript NCOMMS-22-29974. The following are our responses.

Reviewer #2 (Remarks to the Author):

General Comments: The manuscript by Qian-Qian Chen et al. “Neuraminidase 1 plays a promotor role in renal fibrosis” describes a novel role for mammalian neuraminidase 1 (lysosomal neuraminidase, NEU1) in renal fibrosis. The authors demonstrate increased NEU1 expression in cohorts of human patients with renal fibrosis. They further show that NEU1 levels are significantly elevated in tubular epithelial cells (TEC) of fibrotic kidneys in human patients and mouse models. The authors also show that depletion of NEU1 in conditional TEC-specific NEU1 KO mice and epithelial cellular models reduces severity of fibrotic changes, while overexpression of NEU1 has an opposite effect. Finally, they demonstrate that NEU1 interacts with the major TGF β receptor of TEC, ALK5, and activates ALK5-SMAD2 signaling pathway, potentially leading to development of fibrosis. Finally, they show that salvianolic acid B binds with high affinity to recombinant human NEU1 protein and effectively prevents renal injury in mouse unilateral ureteral obstruction (UUO) model. These are important and novel findings potentially leading to development of novel therapeutic approaches to treat fibrosis and chronic kidney disease.

Response: We thank the reviewer for the positive comments.

Major Comment 1: At the same time, the study falls short of clarifying the mechanism by which NEU1 induces fibrosis. The authors propose that NEU1 physically interacts with Transforming Growth Factor β Receptor ALK5 and induces its activation and downstream TGF-beta signalling. They do not demonstrate, however, how exactly NEU1 does this. NEU1 acts in the cells as an enzyme that removes terminal sialic acid residues from glycan chains of glycoproteins. Still, the authors did not attempt to explore whether ALK5 is sialylated and whether it is a substrate of NEU1. Besides, the authors did not consider alternative hypotheses explaining a positive action of NEU1 on fibrosis, such as induction of pro-inflammatory cytokines and recruitment/activation of leucocytes and other immune cells. This action of NEU1 has been convincingly

demonstrated in the current literature, but are not explored by the authors in the current study.

Response: (1) We have employed confocal imaging (Fig. 6c), Co-IP (Fig. 6d, e) and SPR assay (Fig. 6k) to test the binding between NEU1 and ALK5. In line with the reviewer's comments, we performed additional experiments to confirm the binding of NEU1 with ALK5 including bimolecular fluorescence complementation (BiFC) and *in situ* proximity ligation assay (PLA).

BiFC enables the visualization of direct protein-protein interactions in living cells (*Curr Protoc Protein Sci*, 2005, Chapter19, Unit 19.10). In this assay, NEU1 and ALK5 were fused to the C- (VC155) and N-terminal (VN173) non-fluorescent fragments of the Venus fluorescent protein. If NEU1 and ALK5 interact, the VC155 and VN173 fragments are brought into proximity and produce a fluorescent signal (Fig. 6f). Of note, fluorescent signals were much higher in TGF β group than in control group, suggesting a strong binding of NEU1 with ALK5 upon TGF β stimulation (Fig. 6g, h). We have added the relevant information in the Revised Manuscript (Lines 220-225, Page 12).

Per Reviewer #3's suggestion, we also designed a PLA experiment to validate the interaction of NEU1 and ALK5. DNA hybridization and amplification steps with fluorescent probes were used to visualize interacting proteins by fluorescence microscopy (Fig. 6i) (*Curr Protoc Immunol*, 2018, 123: e58). PLA results showed that fluorescent signals were much higher in TGF β group than in control group, indicating a stronger binding of NEU1 with ALK5 upon TGF β stimulation (Supplementary Fig. 10b). SaB treatment inhibited the binding of NEU1 with ALK5 (Supplementary Fig. 10b). We have added the relevant information in the Revised Manuscript (Lines 225-228, Page 12 and Lines 272-274, Page 14).

(2) Next, we measured the stability of ALK5 and Smad2/3 phosphorylation in the presence or absence of NEU1. HK-2 cells were transfected with NEU1 siRNA or NEU1 full-length plasmid, followed by protein synthesis inhibitor cycloheximide (CHX). The results showed that NEU1 knockdown promoted ALK5 degradation, while NEU1 overexpression inhibited ALK5 degradation (Fig. 7a, b), suggesting that NEU1 stabilized ALK5. NEU1 silence markedly suppressed TGF β -induced Smad2/3 activation in a time-dependent manner, while NEU1 overexpression sustained continuous activation of Smad2/3 in the presence of TGF β stimulation (Supplementary Fig. 8a-d). We have added the relevant information in the Revised Manuscript (Lines

243-249, Page 13).

(3) To investigate the effect of NEU1-ALK5 binding on Smad2/3 activation, we transfected the cells with ALK5₁₆₀₋₂₀₀ plasmid. ALK5₁₆₀₋₂₀₀ is the NEU1-binding domain that can compete with ALK5. As expected, ALK5₁₆₀₋₂₀₀ domain reduced phosphorylations of Smad2 (Ser465/467) and Smad3 (Ser423/425) (Fig. 7g, h). As a result, NEU1 bound to ALK5, promoting downstream phosphorylations of Smad2 and Smad3. These results suggest that the interaction between NEU1 and ALK5 enhanced ALK5 stability and promoted Smad2/3 activation. We have added the relevant information in the Revised Manuscript (Lines 253-254, Page 13).

(4) To explore whether NEU1 acts in the cells as an enzyme that removes terminal sialic acid residues of ALK5, we measured the sambucus nigra lectin (SNL) expression of ALK5. SNL binds specifically to sialic acid attached to terminal galactose via α -2-6 linkage (*J Clin Invest*, 2018, 128: 309-322). SNL lectin levels indicated the relative sialylation of the protein (*J Biol Chem*, 2022 298(3): 101594). Our results showed that the SNL level of ALK5 was decreased in response to TGF β stimulation, and this decrease was aggravated in NEU1-overexpressed cells (Supplementary Fig. 12a). This result indicates that NEU1 was involved in ALK5 desialylation. We have added the relevant information in the Revised Manuscript (Lines 370-374, Page 19).

(5) We agree with the reviewer that NEU1-mediated pro-inflammatory cytokines and recruitment/activation of leucocytes and other immune cells might play roles in renal fibrosis. We measured the inflammatory cytokines and chemokines level in *Neu1*CKO mice and NEU1-overexpressed mice. The results showed that TECs-specific NEU1 deficiency inhibited UUO-induced inflammatory cytokines and chemokines expression (Supplementary Fig. 4f, g), while NEU1 overexpression aggravated UUO-induced inflammatory response (Fig. 5o and Supplementary Fig. 6g-i). Previous studies have showed that NEU1 is also elevated in macrophages upon stimulus, leading to disease progression (*Basic Res Cardiol*, 2020, 115(6):62; *FASEB J*, 2022, 36(5): e22285). We investigated the colocalization of NEU1 with macrophages. The results showed that there was non-significant colocalization of NEU1 in macrophages, but a noticeable colocalization of NEU1 in TECs (Fig. 2e, f). These results suggest that NEU1 was mainly elevated in tubule epithelial cells (TECs), and TECs-localized NEU1

is involved in inflammation and renal fibrosis. We have added the relevant information in the Revised Manuscript (Line 120, Page 7).

Fig. 6

Fig. 6 **f** Scheme of NEU1 and ALK5 fusion proteins used for BiFC analysis. **g, h** BiFC signals were detected in HK-2 cells. Representative fluorescence images of HK-2 cells coexpression NEU1-VC155 and ALK5-VN173 plasmid without (**g**) or with (**h**) stimulated with TGFβ. Scale bar, 10 μm. The magnified image scale was 1 μm.

Supplementary Fig. 10b

Supplementary Fig. 10b Interaction between NEU1 and ALK5 (NEU1-ALK5, red) was analyzed by PLA in HK-2 cells treated with SaB (10 μ M) stimulated with TGF β for 24h. Scale bars, 10 μ m. $n = 3$ samples per group.

Fig. 7

Fig.7 a, b HK-2 cells transfected with siNEU1 (**a**) or NEU1 plasmid (**b**) for 24 h and treated with TGF β (10 ng/ml) for 24 h. Then the cells were incubated with cycloheximide (CHX, 20 μ g/ml) for the indicated periods of time (0, 2, 4, 8, 12, 24 h) (left). Lysates were harvested from the cells and analyzed by western blotting (**a**, left panel; **b**, left panel). Quantitation of ALK5 protein levels were shown in the right pane.

Supplementary Fig. 8

Supplementary Fig. 8 a-d HK-2 cells transfected with siNEU1 (**a, b**) or NEU1 plasmid (**c, d**) for 24 h. Then the cells were stimulated with TGF β (10 ng/ml) for the indicated periods of time (0, 6, 12, 24, 36, 48 h). Lysates were harvested from the cells and analyzed by western blotting (**a, c**) Quantitation of ALK5 protein levels were shown in **b, d**. Data were presented as the mean \pm SD. $n = 3$ samples per group. Unpaired two-tailed t -test.

Fig. 7

Fig. 7 g, h Western blots (**g**) and quantitative results (**h**) of p-ALK5(Ser165), p-Smad2(Ser465/467), Smad2, p-Smad3 (Ser423/425), and Smad3 in HK-2 cells transfected with NEU1 and ALK5₁₆₀₋₂₀₀ plasmid for 24 h and treated with TGFβ (10 ng/ml) for 24 h. *n* = 3 samples per group. Relative protein levels were shown after normalization to GAPDH. One-way ANOVA.

Supplementary Fig. 12a

Supplementary Fig. 12 a The level of ALK5 protein sialylation was detected by measuring the alpha 2-6-linked sialic acid bond (biotinylated sambucus nigra lectin, SNL) expression.

Supplementary Fig.4

Supplementary Fig. 4 f, g the mRNA levels of chemokines (*Ccl1*, *Ccl2*, *Ccl3*, *Ccl4*, and *Ccl5*) (**f**) and inflammatory cytokines-associate genes (**g**) in kidney samples, all normalized to *Gapdh*. Data were presented as the mean ± SD. *n* = 4 samples per group. Two-way ANOVA.

Fig. 5 o

Fig. 5 o mRNA levels of the indicated genes in kidneys. $n = 3$ mice per group. Data were presented as mean \pm SD. Unpaired two-tailed t -test.

Supplementary Fig. 6

Supplementary Fig. 6 g, h Immunohistochemistry staining of CD68 or p-NF κ B in kidney sections (g, left; h, left) and quantitative results (g, right; h, right) from AAV9-Ctrl and AAV9-NEU1 mice 10 days after UOO. $n = 3$ mice per group. Data are presented as mean \pm SD. Unpaired two-tailed t -test. **i** the mRNA levels of chemokines (*Ccl1*, *Ccl2*, *Ccl3*, *Ccl4*, *Ccl5*, *Cxcr10*, and *Cxcr12*), inflammatory cytokines (*Il1 β* , *Il6*, and *Il18*), interferons (*Ifna*, *Ifnb*, and *Ifng*), adhesion molecules (*Icam* and *Vcam*), and tumor necrosis factor (*Tnfa*) in kidney samples, all normalized to *Gapdh*. Data were presented as the mean \pm SD. $n = 4$ samples per group. Unpaired two-tailed t -test.

Fig.2

Fig. 2 e, f Immunofluorescence images of NEU1 (e) and CD68 (f) in kidney from mice subjected to UUO. Na⁺/K⁺-ATPase was used as a tubular epithelial cell marker, CD68 was used as a macrophage marker. *n* = 3 mice per group. Scale bar, 50 μ m.

Major Comment 2: Figure 1d. This image could be improved if the authors use adjacent kidney sections for the three types of staining (NEU1, Masson, HE). As presented the selected sections are very different, which prevents the readers from appreciating a correlation between NEU1 increase and renal fibrosis.

Response: To address the issue of different tissue sections for the three types of staining (NEU1, Masson, HE), we repeated this experiment using patients with renal fibrosis (*n* = 8) and without renal fibrosis (no specific pathologic alterations by biopsy procedures, *n* = 8). We have provided the adjacent kidney sections for the three types of staining, and reanalyzed the correlation between NEU1 expression and renal injury (Fig. 1d-k). The quality of the images for the adjacent kidney sections was greatly improved. We have added the relevant information in the Revised Manuscript (Lines 103-109, Page 6).

Fig. 1 d Tissue sections of kidney from patients with non-renal fibrosis or renal fibrosis by immunohistochemistry, Masson staining, and HE staining. Scale bar = 20 μm . NRF: non-renal fibrosis; RF: Renal fibrosis. **e-g** Quantification of NEU1 expression (**e**), fibrotic area (**f**), and score of kidney damage (**g**) based on immunohistochemistry, Masson, or HE staining in d. Data were presented as mean \pm SD. $n = 8$ samples per group. Unpaired two-tailed t -test. IOD: integrated optical density. **h** The correlation of NEU1 expression and degree of tubular degeneration ($n = 16$, Pearson χ^2 test). **i-k** Pearson's correlation of NEU1 with serum creatinine level (**i**), blood urea nitrogen (BUN) (**j**) and glomerular filtration rate (GFR, **k**). $n = 16$ samples. Pearson χ^2 test.

Fig. 1

Major Comment 3: Supplementary Fig. 2g, h. The authors have used the plasmid encoding for HA-tagged NEU1 to overexpress the enzyme in HK-2 cells. Previous studies, however, demonstrated that in order to produce active NEU1, the protein must be co-expressed with its protective protein, PPCA (PMID: 9501080, PMID: 9480870). If overexpressed without PPCA, NEU1 is not targeted to lysosomes, but is retained in the ER where it forms inactive inclusions (PMID: 9501080). The authors should demonstrate the increase of NEU1 enzymatic activity in transfected cells. They also need to demonstrate reduced NEU1 activity in the kidney tissue of NEU1 CKO mice (Sup. Fig. 3) and increased NEU1 activity in the kidney of mice injected with AAV9-

NEU1 (Figure 5 and Sup. Fig. 6).

Response: Based on the reviewer's advices, we measured the NEU1 enzymatic activity after the cells were transfected with the NEU1 full-length plasmid. The results showed that NEU1 enzymatic activity was significantly elevated when NEU1 was overexpressed (Supplementary Fig. 12d). In the kidney tissue, we found that NEU1 activity was decreased in *Neu1*CKO mice (Supplementary Fig. 3e), while the enzyme activity was increased in the mice injected with AAV9-NEU1 (Supplementary Fig. 6b). As the reviewer mentioned, previous studies have demonstrated that PPCA is required for the active NEU1 (*EMBO J*, 1998, 17(6): 1588-1597; *Biochem J*, 1998, 330 (Pt 2): 641-650). Interestingly, we observed that PPCA expression was markedly increased in kidney tissues of mice with NEU1-overexpression (Supplementary Fig. 6c). We have added the relevant information in the Revised Manuscript (Line 143, Page 8, Line 186, Page 10, Lines 188-189, Page 10, and Lines 374-375, Page 19).

Supplementary Fig. 12 d, Neuraminidase activity in HK-2 cells by ELISA. $n = 6$ samples per group. Data were presented as mean \pm SD. One-way ANOVA.

Supplementary Fig. 3 e and Supplementary Fig. 6 b Neuraminidase activity in kidneys by ELISA. $n = 6$ samples per group. Data were presented as mean \pm SD. Unpaired two-tailed t-test.

Supplementary Fig. 6 c Representative western blots of PPCA protein in the kidney of AAV9-Ctrl and AAV9-NEU1 mice.

Major Comment 4: Importantly, NEU1 KO mice and sialidosis patients develop a kidney disease caused by accumulation of lysosomal storage materials in podocytes (PMID: 12023988). Did the authors observe any pathological changes in the kidney of their *Neu1* CKO? Pathological examination of kidney tissues using electron microscopy and immunohistochemical assays specific for lysosomal storage phenotype need to be included in the paper.

Response: Per the suggestion of the reviewer, we employed electron microscopy and immunohistochemical assays to examine possible pathological changes of kidney tissues. Electron microscopy showed that there were no significant morphological differences between *Neu1*CKO mice and the wild type mice in the podocytes, glomerular basement membrane, the blood vessel, and tubular morphology (Supplementary Fig. 3f). Besides, the inflammation, fibrosis, and EMT indicators in the kidneys of *Neu1*CKO mice were almost similar to that of the wild type mice (Fig. 4g-m and Supplementary Fig. 5). A minor change was that a few vacuoles (examples indicated by arrows) was detected in the cytosol of tubular cells in *Neu1*CKO mice but not present in control mice (Supplementary Fig. 3f). We therefore concluded that there was no obvious lysosomal storage phenotype in *Neu1*CKO mice. We have added the relevant information in the Revised Manuscript (Lines 145-149, Page 8).

Next, we performed immunohistochemical assay for LAMP1, an indicator of exacerbated lysosomal exocytosis (*Expert Opin Orphan Drugs*, 2015, 3(5): 491-504; *Sci Adv*, 2019, 5(7): eaav3270). Here, we observed that LAMP1 expression was slightly increased but not significantly altered ($p = 0.39$) in the kidney tissues of *Neu1*CKO mice compared with control groups (Supplementary Fig. 3g). We have added the relevant information in the Revised Manuscript (Lines 150-153, Page 8).

In general, pathological changes were observed in mice with systemic knockout of NEU1 (*Nat Commun*, 2013, 4: 2734; *Sci Adv*, 2019, 5(7): eaav3270) but less in tissue-specific KO mice. In a previous work, we did not observe any side effects in cardiomyocyte-specific NEU1-knockout mice (*Eur Heart J*, 2021, 42(36): 3770-3782). Here, we showed no obvious side effects in TECs-specific NEU1-knockout mice. In close agreement, there is no case report on sialidosis syndrome caused by use of influenza virus neuraminidase inhibitors such as oseltamivir for up to 6 weeks (*N Engl J Med*, 2005, 353: 1363-1373). We have added the relevant information in the Revised Manuscript (Lines 344-353, Page 18).

Supplementary Fig.3

Supplementary Fig. 3 f Representative transmission electron microscopy (TEM) images of kidney tubules and glomerulus from *Neu1*CKO and Ctrl mice. The quantitative results (right) of glomerular basement membrane thickness. The asterisk indicates vacuoles. The arrows indicate glomerular basement membrane. $n = 2$ mice per group. Scale bar, 2 μm . **g** Immunohistochemistry staining of LAMP1 in kidney sections from Ctrl and *Neu1* CKO mice. $n = 3$ mice per group. All data were presented as mean \pm SD. Unpaired two-tailed t -test.

Major Comment 5: Figure 1a, Sup. Fig. 6. CD68 and p-NF-kB staining is barely visible. The images need to be improved, or immunofluorescence should be used instead.

Response: We have provided the high-resolution images and marked the positive staining sites with arrows. Please see Supplementary Fig.6 g and h of the Revised Manuscript.

Supplementary Fig. 6 g, h Immunohistochemistry staining of CD68 or p-NFκB in kidney sections (g, left; h, left) and quantitative results (g, right; h, right) from AAV9-Ctrl and AAV9-NEU1 mice 10 days after UUO. $n = 3$ mice per group. Data are presented as mean \pm SD. Unpaired two-tailed t -test.

Major Comment 6: Sup. Fig. 4. It seems that the expression of pro-inflammatory and fibrosis-associated genes (panel c) was analysed in Sham Ctr, Ctr-UUO and Neu1 CKO-UUO mice but not in Sham Neu1 CKO mice? Why?

Response: We did not observe any obvious differences in morphology, EMT, inflammation, and collagen deposition between sham-control group and sham-*Neu1*CKO group. Considering the experimental cost, we compared Sham-control, UUO-control and UUO-*Neu1*CKO mice but not Sham-*Neu1*CKO mice in PCR array

involving 84 fibrosis-related mRNAs. Actually, we compared all of the four groups in terms of some pro-inflammatory cytokines, chemokines and fibrotic factors in Supplementary Fig. 4f, g.

Supplementary Fig.4

Supplementary Fig. 4 f, g the mRNA levels of chemokines (*Ccl1*, *Ccl2*, *Ccl3*, *Ccl4*, and *Ccl5*) (f) and inflammatory cytokines-associate genes (g) in kidney samples, all normalized to *Gapdh*. Data were presented as the mean \pm SD. $n = 4$ samples per group. Two-way ANOVA.

Major Comment 7: Figure 6. Experiments conducted by the authors demonstrate that NEU1 makes a stable complex with ALK5 in solution, but they do not show if these two proteins interact in the live cell. This could be demonstrated with experimental approaches similar to BioID or BRET. The immunofluorescent images provided by the authors are taken with small magnification and show just a general co-expression of two proteins in tissues but not their co-localization in specific subcellular structures. Even more surprising are the results showing that NEU1 presumably binds to the cytoplasmic domain (amino acid 162-403) of ALK5. NEU1 in cells is located either on the inner side of the lysosomal membrane or on the extracellular side of the plasma membrane, so its presence and its interaction with ALK5 in the cytoplasm are unlikely.

Response: Based on the reviewer's suggestion, we used bimolecular fluorescence complementation (BiFC) to detect the direct interaction between NEU1 and ALK5 in cells. BiFC is based on recombination of a functional fluorescent protein when 2 nonfluorescent fragments are brought together by 2 interacting proteins (*Curr Protoc Protein Sci*, 2005, Chapter19, Unit 19.10), which enables the visualization of protein-protein interactions in living cells. In this assay, NEU1 and ALK5 are fused to the C- (VC155) and N-terminal (VN173) non-fluorescent fragments of the Venus fluorescent protein, respectively. If NEU1 and ALK5 interact, the VC155 and VN173 fragments are brought into proximity and produce a fluorescent signal (Fig. 6f). The

results showed that the fluorescence complementation was apparent after co-transfection of constructs of NEU1-VC155 and ALK5-VN173 while no signals were noted in the cells transfected with individual constructs. Of note, the signals were markedly enhanced in the cytoplasm in response to TGF β stimulation (Fig. 6g, h), suggesting a strong interaction between NEU1 and ALK5. We have added the relevant information in the Revised Manuscript (Lines 220-225, Page 12).

Another technology, *in situ* proximity ligation assay (PLA), was also performed with proximity probes against NEU1 and ALK5 to validate the interaction of NEU1 and ALK5 in fibrotic kidneys of patients and HK-2 cells. PLA is a powerful method to quantify endogenous or exogenous overexpressed protein-protein interactions in cells and identify localization of these interactions. PLA uses specific DNA sequences covalently linked to specific protein antibodies. A DNA hybridization step followed by DNA amplification with fluorescent probes was used to visualize interacting proteins by fluorescence microscopy (Fig. 6i), thereby detecting proteins with high sensitivity and specificity (*Curr Protoc Immunol*, 2018, 123(1): e58). In fibrotic kidneys of patients, PLA results showed strong fluorescent signals in TECs, suggesting a direct interaction between NEU1 and ALK5 (Fig. 6j). In HK-2 cells, fluorescent signals in cytoplasm were much higher in TGF β group than in control group (supplementary Fig. 10b). We have added the relevant information in the Revised Manuscript (Lines 225-229, Page 12 and Lines 272-274, Page 14).

As the reviewer pointed out, many studies have reported that NEU1 in cells is located on the inner side of the lysosomal membrane or on the extracellular side of the plasma membrane. Recently, we found that NEU1 can translocate into the nucleus in response to pressure overload in cardiomyocytes *in vitro* and *in vivo* (*Eur Heart J*, 2021, 42, 3770-3782), indicating that NEU1 can localize in other subcellular organelles. Initially we detected the binding affinity of NEU1 with different domains of ALK5, including the extracellular and intracellular regions. SPR results showed that NEU1 strongly binds to the intracellular domain of ALK5 (Fig. 6k), but does not interact with the extracellular domain of ALK5 (supplementary Fig. 7d, e). BiFC and PLA experiments further confirmed that the NEU1-ALK5 complexes were formed in the cytoplasm. Thus, we assumed that the interaction between NEU1 and ALK5 in cytoplasm, rather than in the cell surface, played the dominant role in renal fibrosis. We have added relevant information in the Discussion of the Revised Manuscript (Lines

Fig. 6

Fig. 6 f, Scheme of NEU1 and ALK5 fusion proteins used for bimolecular fluorescence complementation (BiFC) analysis. **g, h** BiFC signals were detected in HK-2 cells. Representative fluorescence images of HK-2 cells co-expression of NEU1-VC155 and ALK5-VN173 plasmid without (**g**) or with (**h**) TGFβ stimulation. Scale bar, 10 μm. The magnified image scale was 1 μm. **i** Simple schematic diagram of *in situ* proximity ligation assay (NEU1-ALK5 PLA). **j** Interaction between NEU1 and ALK5 (NEU1-ALK5, red arrow) was analyzed by PLA in the fibrotic kidney of patients. Scale bars, 10 μm.

Supplementary Fig. 10b

Supplementary Fig. 10 b Interaction between NEU1 and ALK5 (NEU1-ALK5, red) was analyzed

by PLA in HK-2 cells treated with SaB (10 μ M) stimulated with TGF β for 24h. Scale bars, 10 μ m. $n = 3$ samples per group.

Supplementary Fig. 7

Supplementary Fig. 7 c Schematic diagram of NEU1 binding with ALK5. **d** The interaction between NEU1 and ALK5_{1-125aa} was tested by SPR. **e** The interaction between NEU1 and ALK5_{200-503aa} was tested by SPR.

Fig. 6k

Fig. 6 k The interaction between NEU1 and ALK5_{162-403aa} tested by SPR. The frequency response and fitting curves were displayed.

Major Comment 8: Besides, if produced without PPCA, NEU1 tends to self-aggregate (PMID: 35014832; PMID: 19666471). This could potentially affect the results.

Response: We measured the oligomerization of NEU1 by Native-PAGE, and found that NEU1 did not form aggregation when the cells were transfected with NEU1 full-length plasmid (Supplementary Fig. 2k). Next, we detected PPCA expression in kidney of mice transfected with AAV9-NEU1, and the results showed that NEU1 overexpression promoted the increase of PPCA (Supplementary Fig. 6c). We have added relevant information in the Revised Manuscript (Lines 135-137, Page 7 and Lines 188-189, Page 10).

Supplementary Fig. 2k NEU1 protein in HK-2 cells transfected with NEU1 plasmid detected with native polyacrylamide gel electrophoresis (Native-PAGE) and SDS polyacrylamide gel electrophoresis (SDS-PAGE).

Supplementary Fig. 6c Representative western blots of PPCA protein in the kidney of AAV9-Ctrl and AAV9-NEU1 mice.

Major Comment 9: Page 11, line 215 and Figure 6j. SPR binding measurement of NEU1 and ALK5 indicates “ $K_d = 37.4 \text{ nM}$ ”. This measurement should include an estimate of error and the method by which it was determined, statistical analysis should also be reported. The SPR curves, as labelled, indicate that the 250 nM concentration of the analyte gives higher response than the 500 nM concentration. Is this accurate? Can the fit curves be provided in the supporting information? The protocol (pages 27-28) lists analyte concentration as ng/mL, while the figure shows it in nM.

Response: Per the reviewer’s suggestion, we have provided the affinity curve of NEU1 and ALK5 along with Chi2, correlation coefficient, and an estimate of error (SE). The method by which it was determined and statistical analysis have also been provided (Line 646, Page 33).

As the reviewer mentioned, it is less common that the 250 nM concentration of the analyte gives a higher response than the 500 nM concentration. This abnormal response might be attributed to the amounts of NEU1 protein immobilized on the chip surface, the concentration range of analyte (31.25 nM to 500 nM), and other unknown factors. We repeated this experiment with ligand concentrations ranging from 7.8 nM to 500 nM. The affinity curve was fitted, and ALK5_{162-403aa} exhibited a strong binding to NEU1 with $K_D = 3.14 \text{ nM}$ (Fig. 6k). We have added a detailed description of the detection method in the Revised Manuscript (Lines 641-648, Page 33).

Fig. 6k

Fig. 6 k The interaction between NEU1 and ALK5_{162-403aa} tested by SPR. The frequency response and fitting curves were displayed.

Major Comment 10: Page 21, line 396 What is the source and purity of the salvianolic acid A and B and rosmarinic acid samples?

Response: Salvianolic acid A, salvianolic acid B, and rosmarinic acid were purchased from Chengdu MUST Bio-technology Co. LTD. The purity of these 3 compounds is 98%. We have added the source and purity of all the compounds in the Revised Manuscript (Supplementary Table 4).

Major Comment 11: Page 12, Lines 227 – 233 and Figure 7a. SPR binding measurements of small molecules to NEU1. The values given in Fig 7A are in “log₁₀(Kd)” and range from -2 to +3; What are the units of Kd here (mM? nM?) Again, the measured Kd values should include an estimate of the error and be reported with appropriate estimations of significance. Comparison of the results in Fig 6j and Fig 7b & 7c shows that the former involves the binding of a large peptide (241 residues, ~26 kDa), which gives a maximum RU of ~100 RU (at 500 ng/mL); while small molecules, Sal A and Sal B (MW < 500 Da) give maximum RU of up to 80 RU (at 20 ng/mL). This probably implies that many more than one equivalent of the compound is binding to the immobilized protein. Is this a specific interaction, and were any negative control proteins used to confirm this interaction? Does denatured NEU1 give the same response?

Response: We missed some information in Figure 7a. Here, the units of Kd are μM . We have added the units in the Revised Manuscript. The standard error (SE) and the estimations of significance (Chi²) were also provided in the Revised Manuscript.

Regarding the SPR response unite (Ru) value, we believe that the Ru between

ligand-protein and small molecule-protein cannot be compatible. There are several factors affecting the Ru value. The amount of protein (NEU1) immobilized on the chip surface and the molecular weight of the analyte are two key determinants. Generally, the amounts of proteins immobilized on chips for protein-small molecule interactions are approximately 40-fold higher than that for protein-protein interactions. Here, we used different amounts of NEU1 immobilized on chips for analyzing NEU1-ALK5 affinity and NEU1-SaA affinity. We have added the relevant information in the Methods part of the Revised Manuscript (Lines 637-641, Page 32).

Usually, a negative control is not required in SPR experiments (*Sci Adv*, 2018, 4(11): eaau8408, *Langmuir*, 2022 Nov 23, doi: 10.1021/acs.langmuir.2c02022, *Comput Biol Med*, 2022, 151(Pt A): 106288). Often, no Ru responses are obtained when there is no interaction between the analyte and the target protein. Meanwhile, if the kinetic curves cannot be effectively fitted, the exact K_D values cannot be obtained. These analytes without K_D will also be identified as having no interaction with the target protein. Here, we found that NEU1 could not bind to the amino acid 1-125 or 200-503 region of ALK5. Compared with these negative data, the interactions between NEU1 and ALK5₁₆₂₋₄₀₃ domain with well fitted kinetic curves and the exact K_D values are regarded as specific binding.

In this work, NEU1 protein used was inactive recombinant NEU1 protein. It is still a challenge to purify the active mammal NEU1 protein. To produce active NEU1, the protein must be co-expressed with its protective protein PPCA (*EMBO J*, 1998, 17(6): 1588-1597; *Biochem J*, 1998, 330(Pt 2): 641-650). Besides SPR, we performed PLA, BiFC, and co-immunoprecipitation to further verify the interaction between NEU1 and ALK5.

Major Comment 12: Figure 7. The activity of the small molecule compounds used here, salvianolic acids A and B (Sal A, Sal B) and rosmarinic acid, seems likely to be due to non-specific interactions with multiple proteins. The authors will need to provide more data before they can conclude that these compounds specifically target NEU1 and are not acting through other mechanisms/targets. Sal A and Sal B are known to inhibit a large number of enzymes, suggesting that they are pan-assay interference compounds (PAINs; see Baell and Holloway, *J Med Chem*, 2010 and more recent reviews). Their structures all include a catechol (catechol A) group which could account for these

effects. While it is possible these compounds interact with the intended target, there should be an evidence presented that this is a specific effect, not attributable to the wide range of previously reported effects/interactions of these compounds with targets besides NEU1. Sal A and Sal B are known as ROS scavengers, and specific activities of these compounds have been reported to include: amyloid beta aggregation, alpha synuclein aggregation, MMP-9 enzyme inhibition, heme oxygenase-1 enzyme inhibition, cyclooxygenase-2 enzyme inhibition, inhibition of platelet adhesion, inhibition of PI3K, and many others. Rosmarinic acid is reported to inhibit carbonic anhydrase, complement C3, amylase, MARK4, and many others (see for example: 10.1155/2019/3281260; 10.1186/s40035-019-0159-7; 10.1124/jpet.111.190736; 10.1186/1471-2210-10-10; 10.1002/ijc.24160; 10.1016/j.thromres.2008.05.02010.1111/j.1538-7836.2010.03859.x;10.3906/kim-1403-5; 10.1016/0192-0561(88)90026-4; 10.1038/s41598-020-65648-z).

Response: We agree with the reviewer that we cannot exclude the fact that these compounds (SaA, SaB, and Rosmarinic acid) are acting through other mechanisms/targets besides NEU1. In general, small-molecule drugs have the potential to interact with multiple targets and exhibit diverse pharmacological activities (*Oxid Med Cell Longev*, 2019, 2019: 3281260; *Transl Neurodegener*, 2019, 8: 18. *Sci Rep*, 2020, 10(1): 10300). It is likely that SaA, SaB, and Rosmarinic acid inhibited renal fibrosis through intricate, distinct, and interrelated signaling pathways. We have added relevant information in the Discussion part of the Revised Manuscript (Lines 389-393, Page 20).

Here, we showed that: (1) SaB from *Salvia miltiorrhiza* was screened to exhibit the strongest NEU-binding affinities with a K_D at 21.57 nM by SPR assay (Fig. 7a, Supplementary Fig. 8 and Supplementary Table 4). (2) In UUO mouse model, SaB (40 mg/kg, tail vein injection) significantly attenuated renal injury and renal fibrosis (Fig. 7e, f). In line with this, SaB effectively inhibited Kim1 (Fig. 7g), Snai1 (Fig. 7h), and Snai2 expression (Fig. 7i), the phosphorylation of ALK5 and the down-stream phosphorylation of SMAD2/3. SaB also suppressed proinflammatory cytokine production (Tnf- α , Il-6, and Il1 β) and collagen deposition (Fig. 7j-o). (3) In *Neu1* CKO mice, SaB failed to further reduce renal injury and renal fibrosis in response to UUO stimulation, as evidenced by HE and Masson staining (Fig. 8b, c), Kim1 expression, inflammation factor, collagen production. These data in part indicate that NEU1 is

required for the protective effects of SaB against renal injury and fibrosis.

In this work, we centered on the role for NEU1 in renal fibrosis and a novel therapeutic approach by targeting NEU1 to treat CKD. Whether other signaling pathways are involved in SaB's effects against renal fibrosis as well as their complex interrelated mechanisms remain to be explored. Based on the reviewer's comment, we have added a paragraph in the discussion to address this issue and stated as a limitation (Lines 398-400, Page 20).

Major Comment 13: Do authors propose that SaB inhibits catalytic activity of NEU1?

Response: We measured the NEU1 enzyme activity after SaB treatment. The ELISA results showed that SaB significantly inhibited the catalytic activity of NEU1 in kidneys of mice subjected to UUO surgery (Supplementary Fig. 13b). We have added the relevant information in the Revised Manuscript (Lines 383-384, Page 20).

Supplementary Fig. 13b

Supplementary Fig. 13 b Neuraminidase activity in kidneys by ELISA. $n = 6$ samples per group. Data were presented as mean \pm SD. One-way ANOVA.

Major Comment 14: Do they propose that it reduces the protein levels or localization of NEU1 in the cell or prevents its interaction with ALK5?

Response: NEU1 protein levels were measured after SaB treatment in mice. We observed that SaB inhibited NEU1 protein expression (Supplementary Fig. 13a). Co-immunoprecipitation, and PLA experiments showed that SaB inhibited the interaction between NEU1 and ALK5 (Supplementary Fig. 10a, b). We have added the relevant information in the Revised Manuscript (Line 383, Page 20 and Lines 272-274, Page 14).

Supplementary Fig. 13a

Supplementary Fig. 13 a Representative western blot of NEU1 protein in kidney from salvianolic acid A or salvianolic acid B treated mice subjected to UUU for 10 days.

Supplementary Fig. 10

Supplementary Fig. 10 a Co-immunoprecipitation of NEU1 and ALK5 in HK-2 cells treated with Salvianolic acid B (SaB, 5 μM, 10 μM) stimulated with TGFβ for 24h. **b** Interaction between NEU1 and ALK5 (NEU1-ALK5, red) was analyzed by PLA in HK-2 cells treated with SaB (10 μM) stimulated with TGFβ for 24h. Scale bars, 10 μm. *n* = 3 samples per group.

Major Comment 15: Experiments answering these questions need to be conducted. Also, strangely, Sal B reduces the fibrosis area to 2% (Fig. 7b) and also reduces levels of fibrotic and pro-inflammatory genes (Fig. 7 h-o). This effect exceeds the improvement observed in the NEU1 conditional KO (Neu1 CKO, Figure 3). At the same time Sal B-treated mice with the genetic depletion of NEU1 (Fig. 8) in the same model show higher levels of fibrosis and inflammation. From my point of view, this is not fully compatible with the hypothesis of the authors that the improvement of fibrosis by Sal B treatment is mediated by its action on NEU1.

Response: The reviewer mentioned the comparison of effects between SaB and *Neu1* CKO against renal fibrosis in mice. In fact, the two experiments were performed at different times, using different batches of mice, and by different students. In Fig. 3d,

the fibrotic area was about 20% in UUO model group and reduced to 10% by *Neu1*CKO. In Fig. 7, the fibrotic area was about 5% in UUO model group and reduced to 2% by SaB treatment. From this point, NEU1 knockout and SaB intervention showed comparable effects in inhibiting renal fibrosis by about 50%.

Per the reviewer’s suggestion, we performed parallel comparison of the effects of *Neu1*CKO and SaB intervention. The results showed that both NEU1 knockdown and SaB treatment significantly inhibited UUO-induced renal fibrosis with similar effects.

We agree with the reviewer that we can only conclude that the amelioration of fibrosis by SaB treatment is partly mediated by its action on NEU1, since SaB has the potential to interact with many other targets that remain to be explored. At least, we observed that in *Neu1* CKO mice, SaB failed to further reduce renal injury and renal fibrosis in response to UUO stimulation, as evidenced by HE and Masson staining (Fig. 8b, c), Kim1 expression, inflammation factor, collagen production. These data in part indicate that NEU1 is required for the protective effects of SaB against renal injury.

Figure 1

Figure 1 a Masson’s trichrome staining from control (Sham), *Neu1* CKO or SaB-treated mice 10 days after UUO. Scale bar, 50 μ m. $n = 3$ mice per group. **f** Statistical results for interstitial collagen analyzed by Image Pro-Plus software. $n = 3$ mice per group. Data were presented as mean \pm SD, one-way ANOVA.

Abstract.

Minor Comment 1: “the functions of mammalian neuraminidases remain largely unexplored.” This is an overstatement. Many biological functions of mammalian sialidase have been described. Remove or rephrase.

Response: We agree with the reviewer's comment, this sentence has been corrected as follows: "the functions of mammalian neuraminidases remain less explored." (Lines 26-27, Page 2 in the Revised Manuscript)

Minor Comment 2: Page 4, lines 63-66. "First, mammalian neuraminidases, typically located in the lysosome, are responsible for the initial step of degradation of glycoconjugates by removing sialic acids¹⁶. Neuraminidases deficiency leads to sialidosis, a disease characterized by tissue accumulation of sialo-glycopeptides and sialo-oligosaccharides¹⁷." As it is written, this statement is not correct. Of the 4 mammalian neuraminidases (NEU1-NEU4) only two, NEU1 and NEU4, are found in lysosomes. Besides, sialidosis is caused specifically by genetic deficiency of NEU1.

Response: We thank the reviewer for these suggestions, and these descriptions have been corrected as follows: "First, the mammalian neuraminidase family consists of four members (NEU1-NEU4), responsible for the initial step of degradation of glycoconjugates by removing sialic acids¹⁶. Among them, NEU1 and NEU4 are typically located in lysosomes. NEU1 deficiency leads to sialidosis, a disease characterized by tissue accumulation of sialo-glycopeptides and sialo-oligosaccharides¹⁷." (Lines 65-67, Page 4 in the Revised Manuscript)

Minor Comment 3: Page 4, lines 73 and below "On account of their various functions, mammalian neuraminidases have been shown to play an emerging role in several human diseases, including cardiovascular diseases, cancers, neurodegenerative disorders, and lung diseases²²." PUBMED search reveals a number of recent publications on the roles of NEU1 in autoimmune, cardiovascular and lung diseases and the use of specific inhibitors of this enzyme to block these pathways. I encourage the authors to include these references instead of the review they cite.

Response: The recent publications on the roles of NEU1 in autoimmune (*Front Immunol*, 2022, 13: 883079; *Glycobiology*, 2021, 31(7): 873-883), cardiovascular (*Eur Heart J*, 2021, 42(36): 3770-3782; *J Am Heart Assoc*, 2021, 10(4): e018756; *Basic Res Cardiol*, 2020, 115(6): 62), lung diseases (*J Pharmacol Exp Ther*, 2021, 376(1):136-146; *Am J Physiol Lung Cell Mol Physiol*, 2016, 310(10): L940-954), and the use of specific inhibitors (*J Med Chem*, 2018, 61(24): 11261-11279; *Glycobiology*, 2016, 26(8): 834-849) of this enzyme to block these pathways have been cited in the Revised

Manuscript.

Discussion

Minor Comment 4: Line 297. “We and others have demonstrated the contribution of NEU1 in cardiovascular diseases^{20, 21, 30}.” The list of references is incomplete.

Response: The recent publications (*Int J Biol Sci*, 2022, 18(2): 826-840; *J Am Heart Assoc*, 2021, 10(4): e018756; *Vascul Pharmacol*, 2018, 103-105: 16-28) on the roles of NEU1 in cardiovascular diseases have been cited in the Revised Manuscript.

Minor Comment 5: Line 311. “Emerging evidence has demonstrated that NEU1 can be sorted to the cell surface to desialylate membrane receptors such as TLR4 and insulin receptors^{19, 37}.” Earlier publications showing that NEU1 activates TLR4 receptors exist.

Response: We have cited related references (*Cell Signal*, 2010, 22(2): 314-324; *J Biol Chem*, 2011, 286(42): 36532-36549) in the Revised Manuscript. (Lines 356-357, Page 18 in the Revised Manuscript)

Minor Comment 6: Line 306. “In line with this, there is no case report on sialidosis syndrome caused by use of neuraminidase inhibitors such as oseltamivir for up to 6 weeks³⁶.” Oseltamivir does not show any inhibitory activity for NEU1. It is specific for the influenza neuraminidase.

Response: Our previous studies have found that the anti-influenza drugs zanamivir and oseltamivir effectively inhibited mammalian NEU1 and reduced TAC- or ligation-induced mouse NEU1 activation to its physiological levels (*Circulation*, 2018, 137(13): 1374-1390; *Eur Heart J*, 2021, 42(36): 3770-3782), suggesting that oseltamivir suppressed both viral neuraminidase and mammalian NEU1.

We deeply appreciate the comments from Reviewer 3# on our manuscript NCOMMS-22-29974. The following are our responses.

Reviewer #3 (Remarks to the Author):

General Comments: In the present study, the authors found that NEU1, a mammalian neuraminidase, promotes renal fibrosis, using loss-of-function as well as gain-of-function experiments in two mouse models of renal fibrosis (UUO and folic acid-induced models). As a possible mechanism to explain this phenomenon, the authors found that NEU1 enhances the TGF-beta signaling pathway through interacting with TGF-beta type I receptor. This is also a novel finding. Furthermore, based on binding to NEU1, the authors identified two compounds that are effective on renal injury in mouse models. However, mechanistic studies how NEU1 enhances TGF-beta signaling are not of satisfactory quality, though the authors extensively performed animal experiments and chemical biological study. Specific points to improve this manuscript are listed below.

Response: We thank the reviewer for the positive comments.

Major Comment 1: Figure 6 and thereafter: The authors used an anti-pSmad2 antibody (AF3450, Ser250, not Ser205 as described in this manuscript) to evaluate Smad2 activation. This is not appropriate, because Smad2 protein is activated by ALK5-mediated C-terminal phosphorylation (Ser 465/467). Because the authors stated that NEU1 associates with ALK5 to enhance the downstream signaling, the C-terminal phosphorylation of Smad2 should be examined.

Response: Based on the reviewer's suggestion, we have replaced all the p-Smad2 (Ser205) with p-Smad2 (Ser465/467). Besides p-Smad2 (Ser465/467), we also detected p-Smad3 (Ser423/425) in the related experiments. We observed that NEU1 knockdown suppressed both Smad2 (Ser465/467) phosphorylation and Smad3 (Ser423/425) phosphorylation in response to TGFβ stimulation (Fig. 7c, d). Conversely, NEU1 overexpression promoted TGFβ-induced phosphorylations of Smad2 (Ser465/467) and Smad3 (Ser423/425) (Fig. 7e, f). As a further characterization, we detected phosphorylations of Smad2 and Smad3 in *Neu1*CKO mice, NEU1-overexpressed mice, as well as salvianolic acid B-treated mice subjected to UUO model. In the mouse

kidneys, the UUO-stimulated phosphorylations of Smad2 (Ser465/467) and Smad3 (Ser423/425) were significantly inhibited upon NEU1 knockout (Supplementary Fig. 8e, f). On the contrary, NEU1 overexpression augmented UUO-induced activation of Smad2 (Ser465/467) and Smad3 (Ser423/425) (Supplementary Fig. 8g, h). Salvianolic acid B treatment markedly inhibited the UUO-induced activation of Smad2 (Ser465/467) and Smad3 (Ser423/425) (Supplementary Fig. 10c, d). We have added the relevant information in the Revised Manuscript (Lines 247-254, Page 13 and Lines 274-275, Page 14)

Fig.7

Fig. 7 c-f Western blots (**c, e**) and quantitative results (**d, f**) of p-ALK5 (Ser165), ALK5, p-Smad2 (Ser465/467), Smad2, p-Smad3 (Ser423/425), and Smad3 in HK-2 cells transfected with siNEU1 or NEU1 plasmid for 24 h and then treated with TGFβ (10 ng/ml) for 24 h. $n = 3$ samples per group. Relative protein levels were shown after normalization to GAPDH. Data were presented as the mean \pm SD. One-way ANOVA.

Supplementary Fig. 8

Supplementary Fig. 8 e-h Western blots (**e, g**) and quantitative results (**f, h**) of p-Smad2 (Ser465/467), and p-Smad3 (Ser423/425) protein in kidney from *Neu1* CKO (**e, f**) or NEU1-

overexpress (g, h) mice subjected to UUO for 10 days. Data were presented as the mean \pm SD. $n = 3$ samples per group. One-way ANOVA.

Supplementary Fig. 10

Supplementary Fig. 10 c, d Representative western blots (c) and quantitative results (d) of p-ALK5(Ser165), p-Smad2(Ser465/467), Smad2, p-Smad3 (Ser423/425), and Smad3 in HK-2 cells treated with Salvianolic acid A (SaA, 10 μ M) and Salvianolic acid B (SaB, 10 μ M). Relative protein levels were shown after normalization to GAPDH. $n = 3$ mice per group. Data were presented as mean \pm SD. One-way ANOVA.

Major Comment 2: Figure 6c: This immunofluorescence image simply indicates that NEU1 and ALK5 are expressed in the same cells in a fibrotic kidney and does not necessarily indicate that they are colocalized in kidney cells. NEU1 is known to be usually localized in the lysosome. Therefore, possible interaction between NEU1 and ALK5 should be robustly demonstrated. The authors should consider detection of their colocalization using in situ PLA assay.

Response: Based on the reviewer's suggestion, *in situ* proximity ligation assay (PLA) was performed with proximity probes against NEU1 and ALK5 to validate the interaction of NEU1 and ALK5 in fibrotic kidney of patient and HK-2 cells. In the fibrotic kidney of the patient, PLA results showed strong fluorescent signals in TECs (Fig. 6i, j), suggesting a direct interaction between NEU1 and ALK5. In HK-2 cells, fluorescent signals in the cytoplasm were much higher in the TGFβ group than the control group (Supplementary Fig. 10b), indicating that they are colocalized and binding in cells. We have added the relevant information in the Revised Manuscript (Line 225-229, Page 12).

Another technology, bimolecular fluorescence complementation (BiFC) was also used to detect the direct interaction between NEU1 and ALK5 in cells. BiFC is based

on recombination of a functional fluorescent protein when 2 nonfluorescent fragments are brought together by 2 interacting proteins (*Curr Protoc Protein Sci*, 2005, Chapter19, Unit 19.10), which enables the visualization of protein-protein interactions in living cells. In this assay, NEU1 and ALK5 are fused to the C- (VC155) and N-terminal (VN173) non-fluorescent fragments of the Venus fluorescent protein, respectively. If NEU1 and ALK5 interact, the VC155 and VN173 fragments are brought into proximity and produce a fluorescent signal (Fig. 6f). The results showed that the fluorescence complementation was apparent after co-transfection of constructs of NEU1-VC155 and ALK5-VN173 while no signals were noted in the cells transfected with individual constructs. Of note, the signals were markedly enhanced in the cytoplasm in response to TGF β stimulation (Fig. 6g, h), suggesting a strong interaction between NEU1 and ALK5. We have added the relevant information in the Revised Manuscript (Lines 220-225, Page 12).

Fig. 6 f, Scheme of NEU1 and ALK5 fusion proteins used for bimolecular fluorescence complementation (BiFC) analysis. **g, h** BiFC signals were detected in HK-2 cells. Representative fluorescence images of HK-2 cells co-expression of NEU1-VC155 and ALK5-VN173 plasmid without (**g**) or with (**h**) TGF β stimulation. Scale bar, 10 μ m. The magnified image scale was 1 μ m. **i** Simple schematic diagram of *in situ* proximity ligation assay (NEU1-ALK5 PLA). **j** Interaction between NEU1 and ALK5 (NEU1-ALK5, red arrow) was analyzed by PLA in the fibrotic kidney of patients. Scale bars, 10 μ m.

Supplementary Fig. 10b

Supplementary Fig. 10 b Interaction between NEU1 and ALK5 (NEU1-ALK5, red) was analyzed by PLA in HK-2 cells treated with SaB (10 μ M) stimulated with TGF β for 24h. Scale bars, 10 μ m. $n = 3$ samples per group.

Major Comment 3: Figure 6h and i: These data importantly reveal that the ALK5 expression level was decreased by NEU1 KO and increased by NEU1 overexpression. It appears likely that NEU1 interacts with and stabilize ALK5, to enhance TGF β signaling. I recommend the authors to examine this possibility using HK-2 cells, examining the effects of NEU1 on ALK5 stability after TGF β stimulation (ALK5 is downregulated after TGF β stimulation by the action of a negative feedback loop involving Smad7 and Smurf2, PMID22378783). In addition, Smad phosphorylation usually declines within 24 h after stimulation. It would be nice to monitor time course of Smad C-terminal phosphorylation after TGF β stimulation in the presence and absence of NEU1. NEU1 may sustain Smad phosphorylation.

Response: Based on the reviewer's suggestion, we measured the stability of ALK5 and Smad2 phosphorylation in the presence or absence of NEU1. HK-2 cells were transfected with NEU1 siRNA or NEU1 full-length plasmid, followed by protein synthesis inhibitor cycloheximide (CHX). The results showed that NEU1 knockdown promoted ALK5 degradation, while NEU1 overexpression inhibited ALK5 degradation (Fig. 7a, b), suggesting that NEU1 stabilized ALK5. NEU1 silence markedly suppressed TGF β -induced Smad2/3 activation in a time-dependent manner, while

NEU1 overexpression sustained continuous activation of Smad2/3 in the presence of TGF β stimulation (Supplementary Fig. 8a-d). We have added the relevant information in the Revised Manuscript (Lines 243-249, Page 13).

Fig. 7

Fig. 7 a, b HK-2 cells transfected with siNEU1 (**a**) or NEU1 plasmid (**b**) for 24 h and treated with TGF β (10 ng/ml) for 24 h. Then the cells were incubated with cycloheximide (CHX, 20 μ g/ml) for the indicated periods of time (0, 2, 4, 8, 12, 24 h) (left). Lysates were harvested from the cells and analyzed by western blotting (**a**, left panel; **b**, left panel). Quantitation of ALK5 protein levels were shown in the right panel.

Supplementary Fig. 8

Supplementary Fig. 8 a-d HK-2 cells transfected with siNEU1 (**a, b**) or NEU1 plasmid (**c, d**) for 24 h. Then the cells were stimulated with TGF β (10 ng/ml) for the indicated periods of time (0, 6, 12, 24, 36, 48 h). Lysates were harvested from the cells and analyzed by western blotting (**a, c**). Quantitation of p-Smad2/3 protein levels were shown in **b, d**. Data were presented as the mean \pm SD. $n = 3$ samples per group. Unpaired two-tailed t -test.

Major Comment 4: Using HK-2 cells, effects of SA-A and SA-B on ALK5 expression, pALK5, pSmad2 (C-terminal phosphorylation) after TGF β stimulation should be examined. In addition, examine if these compounds affect NEU1-ALK5 interaction. These data would significantly strengthen this paper.

Response: Based on the reviewer's suggestion, phosphorylation of ALK5/Smad2 signaling was measured after SaA and SaB treatment in HK-2 cells. SaA and SaB markedly suppressed TGF β -induced the activation of ALK5 (Ser165) and Smad2 (Ser465/467) (Supplementary Fig. 10c, d). Co-IP and PLA experiments showed that SaB inhibited the interaction between NEU1 and ALK5 (Supplementary Fig. 10a, b). We have added the relevant information in the Revised Manuscript (Lines 274-275, Page 14).

Supplementary Fig. 10

Supplementary Fig. 10 a Co-immunoprecipitation of NEU1 and ALK5 in HK-2 cells treated with Salvianolic acid B (SaB, 5 μ M, 10 μ M) stimulated with TGF β for 24h. **b** Interaction between NEU1 and ALK5 (NEU1-ALK5, red) was analyzed by PLA in HK-2 cells treated with SaB (10 μ M) stimulated with TGF β for 24h. Scale bars, 10 μ m. $n = 3$ samples per group. **c, d** Representative western blots (**c**) and quantitative results (**d**) of p-ALK5(ser165), p-Smad2(ser465/467), Smad2, p-Smad3 (ser423/425), and Smad3 in HK-2 cells treated with Salvianolic acid A (SaA, 10 μ M) and

Salvianolic acid B (10 μ M). Relative protein levels were shown after normalization to GAPDH. $n = 3$ mice per group. Data were presented as mean \pm SD. One-way ANOVA.

Major Comment 5: In the abstract, the authors stated “Neu1 interacted with TGF β type I receptor ALK5 at the 160-200aa region to activate ALK5-SMAD2 signaling pathway”. However, knockdown or overexpression of NEU1 did not affect TGF β target gene expression in the absence of TGF β ligand (Supplementary Figure 2e and i). Therefore, “to activate ALK5-SMAD2 signaling pathway” is misleading. Instead, “to enhance ALK5-SMAD2 signaling pathway” would be appropriate. Same to lines 202, 269, and 339.

Response: We thank the Reviewer for this suggestion. We have corrected “to activate ALK5-SMAD2 signaling pathway” into “to enhance ALK5-SMAD2 signaling pathway” in the Revised Manuscript.

Major Comment 6: Is NEU1 enzymatic activity required for enhancing the TGF-beta pathway?

Response: Based on the reviewer’s advice, we measured the TGF β pathway in HK-2 cells transfected with NEU1 full-length plasmid and NEU1 mutant enzyme plasmid. The results showed that NEU1 enzyme inactivation greatly inhibited TGF β pathway compared with NEU1 wild type plasmid group (Supplementary Fig. 12b-e in the Revised Manuscript), indicating that NEU1 enzymatic activity is required for enhancing the TGF β pathway. We have added the relevant information in the Revised Manuscript (Lines 374-375, Page 19)

Supplementary Fig. 12

Supplementary Fig. 12 b Human NEU1 amino acid mutation site (mtNEU1: D103A, Y370A, E394A). **c** Relative mRNA levels of *Neu1* in HK-2 cells transfected with NEU1 and mtNEU1 plasmid. $n = 4$ samples per group. Data were presented as mean \pm SD. One-way ANOVA. **d** Neuraminidase activity in HK-2 cells by ELISA. $n = 6$ samples per group. Data were presented as mean \pm SD. One-way ANOVA. **e** Representative western blots of p-Smad2 (Ser465/467), p-Smad3 (Ser423/425), Smad2, and Smad3 in HK-2 cells transfected with NEU1 and mtNEU1 plasmid.

Minor Comment 1: Figure 1d: The authors should have used serial sections. Otherwise, the data are not convincing enough.

Response: We repeated this experiment using patients with renal fibrosis ($n = 8$) and without renal fibrosis (no specific pathologic alterations by biopsy procedures, $n = 8$). We have provided the adjacent kidney sections for the three types of staining, and reanalyzed the correlation between NEU1 expression and renal injury (Fig. 1d-k). The quality of the images for the adjacent kidney sections was greatly improved. We have added the relevant information in the Revised Manuscript (Lines 100-109, Page 6).

Fig. 1
Fig. 1 d Tissue sections of kidney from patients with non-renal fibrosis or renal fibrosis by immunohistochemistry, Masson staining, and HE staining. Scale bar = 20 μm . NRF: non-renal fibrosis; RF: Renal fibrosis. **e-g** Quantification of NEU1 expression (**e**), fibrotic area (**f**), and score of kidney damage (**g**) based on immunohistochemistry, Masson, or HE staining in **d**. Data were presented as mean \pm SD. $n = 8$ samples per group. Unpaired two-tailed t -test. IOD: integrated optical density. **h** The correlation of NEU1 expression and degree of tubular degeneration ($n = 16$, Pearson χ^2 test). **i-k** Pearson's correlation of NEU1 with serum creatinine level (**i**), blood urea nitrogen (BUN) (**j**), and glomerular filtration rate (GFR) (**k**) ($n = 16$, Pearson χ^2 test).

Minor Comment 2: Line 119, “NEU1 mediated TGF β -induced HK-2 cell injury”: I am afraid that the authors did not examine “HK-2 cell injury”. In addition, induction of KIM1 by NEU1 overexpression was attenuated by ALK5 knockdown (Supplementary Fig. 7d), indicating that ALK5 mediates NEU1-induced cell response(s). Reconsider this description.

Response: We have corrected this description as “NEU1 mediated TGF β -induced changes in HK-2 cells.”

Minor Comment 3: Line 202, “As expected, the interaction between NEU1 and ALK5

was enhanced in response to TGFb stimulation in HK-2 cells”: Why did the authors “expect” that the interaction would be enhanced by TGFb stimulation? Is this an important event for enhancement of TGFb signaling by NEU1?

Response: We have deleted the “As expected” in the Revised Manuscript.

Minor Comment 4: In Supplementary Fig. 2, the authors observed that siNEU1 attenuates TGFb-induced gene expression (24 h stimulation). Is this a general effect on TGFb signaling or a limited effect on EMT-related genes? This point should be made clear.

Response: As the reviewer indicated, we observed that siNEU1 attenuates TGFb-induced gene expression. E-Cadherin (*CDHI*), Vimentin (*VIM*), Snail (*SNAIL*), and Slug (*SNAIL2*) are EMT-related markers. Others such as Fibronectin 1 and KIM1 are associated with TECs injury and renal fibrosis. More importantly, we employed genetic engineering to identify the role of NEU1 not only limited in EMT-related genes (Fig. 4l and Supplementary Fig. 5a, b) but also in pro-inflammatory cytokines (Fig. 4g, k and Supplementary Fig. 5c), chemokines (Supplementary Fig. 5d), and fibrogenic factors (Fig. 4m). We have made it clear that siNEU1 has a general effect on TGFb signaling (please see Lines 127-137, Page 7).

Minor Comment 5: Figure 1j: Data on NEU1 and serum creatine levels, although described in the legend, are missing. Current Fig. 1j should be Fig. 1k (mislabeled).

Response: We have added the data on NEU1 and serum creatine levels in the Revised Manuscript.

Minor Comment 6: Anti-KIM1 antibody is not described in “Methods”.

Response: We have added the description of Anti-KIM1 antibody in the “Methods” of the Revised Manuscript.

Minor Comment 7: Line 147; knockdown >> knockout.

Response: We have corrected “knockdown” into “knockout” in the Revised Manuscript (Line 168, Page 9).

REVIEWER COMMENTS

Reviewer #1 (Remarks to the Author):

The revision has addressed the main issues. But, there are still some issues that need to be addressed for the sake of the rigor of the article.

Some Western Blot bands are of unsatisfactory quality, i. e. Fig 8p p-Smad3, Fig 5c GAPDH.

In Fig 9b and 9i, some positive staining sites marked by arrows were distributed in the glomeruli. These results were also marked as positive and statistically analyzed. However, in discussion, the first sentence: "This study is the first, to our knowledge, to identify a key role for NEU1 in TEC injury and renal fibrosis based on the results of genetic, in vivo, in vitro, and pharmacological experiments." These data did not support "a key role for NEU1 in TEC injury".

Graphic abstract presents the distribution of ALK on the cell membrane, but in Figure 9i, some of the ALK-positive stained cells shown by arrows are distributed in the nucleus.

In Fig 7c, "p-Samd2" to "p-Smad2", and in Figure 7e, "p-Samd3" to "p-Smad3".

In Fig 7c,d why knockdown NEU1 increased the level of p-ALK, was there a statistical difference between PBS:siNC and PBS:siNEU1?

Reviewer #2 (Remarks to the Author):

The authors were very responsive in addressing the comments of this reviewer and in my opinion the revised manuscript has been drastically improved as compared with the first draft. However a few points remain to be addressed. They all relate to newly added results and text changes.

1. Supplementary figure 6c Representative western blots of PPCA protein in the kidney of AAV9-Ctrl and c Representative 104 western blots of PPCA protein in the kidney of AAV9-Ctrl and AAV9-NEU1 mice. The authors demonstrate increased protein levels of PPCA in the tissues of mice treated by AAV9-NEU1. This in their opinion rescues overexpressed NEU1 from self-aggregation. The figure shows a 55 kDa band, the size of PPCA precursor. Mature PPCA is a heterodimer of 32 kDa and 20 kDa subunits which run separately on SDS gel if the protein is reduced and denatured. What kind of blot did the authors use?

2. Figure 6. To identify sites of NEU1 and ALK5 interaction in the cells the authors conducted bimolecular fluorescence complementation (BiFC) assay and in situ proximity ligation assay (PLA). Both techniques convincingly demonstrate the interaction, however the authors do not attempt to localize the intracellular compartment in which the interaction takes place. In my point of view the perinuclear fluorescent puncta in panels h, g, and j resembles the lysosomal/endosomal compartment. If this is correct the interaction could place on the surface of endosomal/lysosomal membranes. The authors can easily verify this by labeling lysosomes with LysoTracker dye or using antibodies against LAMP labeled with a fluorophore.

3. Neuraminidase activity. The authors describe (Line 483) that "The activity of Neuraminidase was determined with commercial kits according to the manufacturer's instructions (Mouse Neuraminidase ELISA kit, Mibio, catalog no. ml024431; Human Neuraminidase ELISA kit, catalog no. ml037504)."

These kits determine the amount of neuraminidase protein but not enzymatic activity of neuraminidase. How did the authors measure enzymatic activity of neuraminidase as they claim in the rebuttal? "Based on the reviewer's advices, we measured the NEU1 enzymatic activity after the cells were transfected with the NEU1 full-length plasmid. The results showed that NEU1 enzymatic activity was significantly elevated when NEU1 was overexpressed (Supplementary Fig. 12d). In the kidney

tissue, we found that NEU1 activity was decreased in Neu1CKO mice (Supplementary Fig. 3e), while the enzyme activity was increased in the mice injected with AAV9-NEU1 (Supplementary Fig. 6b).” The authors should measure enzymatic neuraminidase activity using the colorimetric or fluorogenic substrate.

Minor changes

1. References 29 and 38 are the same.
2. Line 135. “NEU1 overexpression did not form aggregation when the cells transfected with NEU1 full-length plasmid (Supplementary Fig. 2k).” Perhaps the authors meant “Overexpressed NEU1 did not form aggregates in the cells ...”
3. Line 151 “LAMP1, an indicator of exacerbated lysosomal exocytosis^{17, 29}” Please add that LAMP1 is also an indicator of lysosomal storage.
4. Lines 370-378 and Supplementary Figure 12. It is better to move this paragraph to the Results section, for example after the line 262, leaving in the Discussion only one or two sentences interpreting the results of the experiments and discussing their limitations.

Reviewer #3 (Remarks to the Author):

The authors have addressed all of my concerns. I have no additional comments.

One minor comment:

Supplementary Fig. 8g: pSmad3 is mislabeled (52 kDa, not 60 kDa).

Responses to the Reviewers' Comments

We deeply appreciate the comments from Reviewer 1# on our manuscript NCOMMS-22-29974A. The following are our responses.

Reviewer 1# (Remarks to the Author):

The revision has addressed the main issues. But, there are still some issues that need to be addressed for the sake of the rigor of the article.

Response: We thank the reviewer for the comments.

Comment 1: Some Western Blot bands are of unsatisfactory quality, i. e. Fig 8p p-Smad3, Fig 5c GAPDH.

Response: We repeated these experiments and replaced Fig. 8p p-Smad3 and Fig. 5c GAPDH in the Revised Manuscript.

Fig. 8 p Western blots of p-ALK5 (Ser165), p-SMAD2/3, and SMAD2/3 in kidney from control (Ctrl) and SaA or SaB-treated mice 10 days after UUO. $n = 3$ mice per group.

Fig. 5 c NEU1 protein levels in the kidneys of AAV9-Ctrl and AAV9-NEU1 mice. $n = 2$ mice per group.

Comment 2: In Fig 9b and 9i, some positive staining sites marked by arrows were distributed in the glomeruli. These results were also marked as positive and statistically analyzed. However, in discussion, the first sentence: "This study is the first, to our knowledge, to identify a key role for NEU1 in TEC injury and renal fibrosis based on the results of genetic, in vivo, in vitro, and pharmacological experiments." These data did not support "a key role for NEU1 in TEC injury".

Response: Indeed, in Fig 9b and 9i, some positive staining sites were distributed in the glomeruli, indicating that NEU1 knockout alleviated UUO-induced renal injury including glomeruli and renal tubules pathological changes. We agree with the reviewer's comment and have corrected the description in the Discussion section of the Revised Manuscript as follows: "This study is the first, to our knowledge, to identify a key role for TEC-located NEU1 in kidney injury and renal fibrosis based on the results of genetic, in vivo, in vitro, and pharmacological experiments." (Lines 319-320, Page 16).

Comment 3: Graphic abstract presents the distribution of ALK on the cell membrane, but in Figure 9i, some of the ALK-positive stained cells shown by arrows are distributed in the nucleus.

Response: In this work, we did not investigate the exact sub-cellular location of p-ALK5. Based on current knowledge, we assume that p-ALK5 is mainly located in cytoplasm. We cannot exclude the possibility of p-ALK5 location in the nucleus. To avoid confusion, we deleted the arrows pointing to p-ALK5 overlapped with the nucleus in Figure 9i in the Revised Manuscript.

Fig. 9 i Representative image of immunohistochemical staining with p-ALK5 (ser165) (Scale bar, 50 μ m). $n = 3$ mice per group.

Comment 4: In Fig 7c, "p-Samd2" to "p-Smad2", and in Figure 7e, "p-Samd3" to "p-Smad3".

Response: We have corrected the spelling in the Revised Manuscript.

Comment 5: In Fig 7c, d why knockdown NEU1 increased the level of p-ALK, was there a statistical difference between PBS:siNC and PBS:siNEU1?

Response: There was no statistical difference in level of p-ALK5 between PBS:siNC group and PBS:siNEU1 group. We have added the statistical results in the Revised Manuscript.

Fig.7 d

Fig. 7 d Quantitative results of p-ALK5, ALK5, p-SMAD2/3, and SMAD2/3 in HK-2 cells transfected with siNEU1 or NEU1 plasmid for 24 h and treated with TGFβ (10 ng/ml) for 24 h. *n* = 3 samples per group. Relative protein levels were shown after normalization to GAPDH. Data were presented as the mean ± SD. One-way ANOVA.

We deeply appreciate the comments from Reviewer 2# on our manuscript NCOMMS-22-29974A. The following are our responses.

Reviewer 2# (Remarks to the Author):

The authors were very responsive in addressing the comments of this reviewer and in my opinion the revised manuscript has been drastically improved as compared with the first draft.

Response: We thank the reviewer for the comments.

However, a few points remain to be addressed. They all relate to newly added results and text changes.

Comment 1: Supplementary figure 6c Representative western blots of PPCA protein in the kidney of AAV9-Ctrl and c Representative western blots of PPCA protein in the kidney of AAV9-Ctrl and AAV9-NEU1 mice. The authors demonstrate increased protein levels of PPCA in the tissues of mice treated by AAV9-NEU1. This in their opinion rescues overexpressed NEU1 from self-aggregation. The figure shows a 55 kDa band, the size of PPCA precursor. Mature PPCA is a heterodimer of 32 kDa and 20 kDa subunits which run separately on SDS gel if the protein is reduced and denatured. What kind of blot did the authors use?

Response: We used the 55 kDa band of anti-PPCA antibody. Per the reviewer's comment, we measured all bands in uncropped full-length pictures of western blotting (The EMBO J, 1998, 17, 1588-1597). The results showed that the expressions of PPCA precursor at 55 kDa were significantly upregulated in NEU1-overexpressed mice compared with the control group. We observed a slight elevation of mature PPCA at 32 kDa and no significant differences at 20 kDa between the two groups. We have added the relevant information in the Revised Manuscript (Lines 188-192, Page 10).

Supplementary Fig. 6c

Supplementary Fig. 6 c Representative western blots of PPCA protein in the kidney of AAV9-Ctrl and AAV9-NEU1 mice.

Comment 2: Figure 6. To identify sites of NEU1 and ALK5 interaction in the cells the authors conducted bimolecular fluorescence complementation (BiFC) assay and in situ proximity ligation assay (PLA). Both techniques convincingly demonstrate the interaction, however the authors do not attempt to localize the intracellular compartment in which the interaction takes place. In my point of view the perinuclear fluorescent puncta in panels h, g, and j resembles the lysosomal/endosomal compartment. If this is correct the interaction could place on the surface of endosomal/lysosomal membranes. The authors can easily verify this by labeling lysosomes with LysoTracker dye or using antibodies against LAMP labeled with a fluorophore.

Response: Per the reviewer's suggestion, we used the Lyso-Tracker to observe the localization of the interaction between NEU1 and ALK5. The results showed that fluorescent signals of NEU1 and ALK5 interaction were co-localized with lysosomes. Besides, we also used Mito-Tracker and CM-Tracker to investigate their possible localizations. The fluorescent signals of NEU1 and ALK5 interaction were also co-localized with mitochondrion, but less co-localized with cell plasma membrane (Supplementary Fig. 7c). The related information was added in the Revised Manuscript (Lines 227-230, Page 12).

Supplementary Fig. 7c

Supplementary Fig. 7c Representative fluorescence images of HK-2 cells co-expression of NEU1-VC155 and ALK5-VN173 plasmid with TGF β stimulation. Lyso-Tracker, Mito-Tracker, and CM-Tracker were used to label lysosome, mitochondrion, and cell plasma membrane respectively. Scale bar, 5 μ m.

Comment 3: Neuraminidase activity. The authors describe (Line 483) that “The activity of Neuraminidase was determined with commercial kits according to the manufacturer’s instructions (Mouse Neuraminidase ELISA kit, Mibio, catalog no. ml024431; Human Neuraminidase ELISA kit, catalog no. ml037504).” These kits determine the amount of neuraminidase protein but not enzymatic activity of neuraminidase. How did the authors measure enzymatic activity of neuraminidase as they claim in the rebuttal? “Based on the reviewer’s advices, we measured the NEU1 enzymatic activity after the cells were transfected with the NEU1 full-length plasmid. The results showed that NEU1 enzymatic activity was significantly elevated when NEU1 was overexpressed (Supplementary Fig. 12d). In the kidney tissue, we found that NEU1 activity was decreased in Neu1CKO mice (Supplementary Fig. 3e), while the enzyme activity was increased in the mice injected with AAV9-NEU1 (Supplementary Fig. 6b).” The authors should measure enzymatic neuraminidase activity using the colorimetric or fluorogenic substrate.

Response: Based on the reviewer’s advices, neuraminidase activity was measured by fluorometric assay with substrate 2’-(4-methylumbelliferyl)- α -d-N-acetylneuraminic acid (4-MU-NANA) (ab138888, abcam). The results showed that

NEU1 enzymatic activity was significantly elevated when NEU1 was overexpressed in HK-2 cells (Supplementary Fig. 9d). In the kidney tissues, we found that NEU1 activity was decreased in *Neu1*CKO mice (Supplementary Fig. 3e), while the enzyme activity was increased in the mice injected with AAV9-NEU1 (Supplementary Fig. 6b).

Supplementary Fig. 9d Supplementary Fig. 6b Supplementary Fig. 3e

Supplementary Fig. 9 d, Neuraminidase activity in HK-2 cells was measured by a fluorometric assay with substrate 2'-(4-methylumbelliferyl)- α -d-N-acetylneuraminic acid (4-MU-NANA) (ab138888, Abcam). $n = 6$ samples per group. Data were presented as mean \pm SD. One-way ANOVA.

Supplementary Fig. 6 b and Supplementary Fig. 3 e Neuraminidase activity in kidneys was measured by a fluorometric assay with substrate 2'-(4-methylumbelliferyl)- α -d-N-acetylneuraminic acid (4-MU-NANA) (ab138888, Abcam). $n = 6$ samples per group. Data were presented as mean \pm SD. Unpaired two-tailed t-test.

Minor changes:

Comment 4: References 29 and 38 are the same.

Response: We have deleted the duplicated references in the Revised Manuscript.

Comment 5: Line 135. “NEU1 overexpression did not form aggregation when the cells transfected with NEU1 full-length plasmid (Supplementary Fig. 2k).” Perhaps the authors meant “Overexpressed NEU1 did not form aggregates in the cells ...”

Response: We have corrected the description as follows: “Overexpressed NEU1 did not form aggregates when the cells transfected with NEU1 full-length plasmid (Supplementary Fig. 2k)” in the Revised Manuscript (Line 136, Page 7).

Comment 6: Line 151 “LAMP1, an indicator of exacerbated lysosomal exocytosis17, 29” Please add that LAMP1 is also an indicator of lysosomal storage.

Response: We have corrected the description as follows: “LAMP1, an indicator of lysosomal storage and exocytosis^{17, 29}” in the Revised Manuscript (Lines 150-151, Page 8).

Comment 7: Lines 370-378 and Supplementary Figure 12. It is better to move this paragraph to the Results section, for example after the line 262, leaving in the Discussion only one or two sentences interpreting the results of the experiments and discussing their limitations.

Response: We agree with the reviewer’s suggestion and have moved the relevant paragraph to the Results section in the Revised Manuscript (Lines 269-275, Page 14).

We deeply appreciate the comments from Reviewer 3# on our manuscript NCOMMS-22-29974A. The following are our responses.

Reviewer 3# (Remarks to the Author):

The authors have addressed all of my concerns. I have no additional comments.

Response: We thank the reviewer for the comments.

One minor comment:

Comment 1: Supplementary Fig. 8g: pSmad3 is mislabeled (52 kDa, not 60 kDa).

Response: We have corrected the label in the Revised Manuscript.

REVIEWERS' COMMENTS

Reviewer #1 (Remarks to the Author):

The revision has addressed the main issues, and the quality of the manuscript has been further improved.

Reviewer #2 (Remarks to the Author):

All my comment have been appropriately addressed during the second review. I do not have other suggestions or concerns.

Responses to the Reviewers' Comments

We deeply appreciate the comments from Reviewer 1# on our manuscript NCOMMS-22-29974B. The following are our responses.

Reviewer #1 (Remarks to the Author):

Comments: The revision has addressed the main issues, and the quality of the manuscript has been further improved.

Response: We thank the reviewer for the comments.

We deeply appreciate the comments from Reviewer 2# on our manuscript NCOMMS-22-29974B. The following are our responses.

Reviewer #2 (Remarks to the Author):

Comments: All my comment have been appropriately addressed during the second review. I do not have other suggestions or concerns.

Response: We thank the reviewer for the comments.